# Hemocytes facilitate interclonal cooperation-induced tumor malignancy by hijacking the innate immune system in *Drosophila*

Sihua Zhao[1,2,3,7], Yifan Guo [ID][2,3,7], Xiaoyu Kuang[2,3], Xiaoqin Li[2,3], Chenxi Wu[4], Peng Lin [ID][2,3], Qi Xie [ID][2,3], Zongzhao Zhai[5], Du Kong [ID][2,3,6✉] & Xianjue Ma [ID][2,3✉]

## Abstract

Tumor heterogeneity, a hallmark of cancer, frequently leads to treatment failure and relapse. However, the intricate communication between various cell types within the tumor microenvironment and their roles in tumor progression in vivo remain poorly understood. Here we establish a novel tumor heterogeneity model in the *Drosophila* larval eye disc epithelium and dissect the in vivo mechanisms by combining sophisticated genetics with single-cell RNA sequencing. We found that mutation of the tricellular junction protein *M6* in cells surrounding *RasV12* benign tumors promotes their malignant transformation. Mechanistically, early *RasV12//M6−/−* tumors secrete Pvf1, which activates the Pvr receptor on hemocytes, facilitating their recruitment to the tumor site. These tumor-associated hemocytes secrete the Spätzle (Spz) ligand to activate the Toll receptor within the *RasV12* tumors. This enhanced activation of the Toll pathway synergizes with *RasV12* to promote malignant transformation through the JNK-Hippo signaling cascade. In summary, our study elucidates the complex interplay between genetically distinct oncogenic cells and between tumors and hemocytes, highlighting how hemocytes exploit the ancient innate immune system to coordinate tumor heterogeneity and drive tumor progression.

**Keywords** Tumor Heterogeneity; Ras; Toll Signaling; Tricellular Junction; Hemocyte
**Subject Categories** Cancer; Immunology; Signal Transduction

## Introduction

Cancer is a complex multifactorial disease. A significant factor contributing to tumor treatment failure and resistance is the presence of tumor heterogeneity, which is defined as the coexistence of subclones carrying distinct genetic mutations within the same tumor. Through the application of the single-cell sequencing (sc-seq) technique, researchers have revealed that tumors are composed of various cell populations, and the intercellular communications between different subclones are essential for tumor progression (Gavish et al, 2023; Gonzalez-Silva et al, 2020; Stewart et al, 2020; Vitale et al, 2021). Convincing evidence supporting this concept has also been obtained from studies utilizing mouse xenograft models, indicating that the cooperation between different subclones can lead to tumor formation (Cleary et al, 2014; Hill et al, 2021; Janiszewska et al, 2019; Marusyk et al, 2014). However, current technologies may not be sufficiently advanced to fully capture and understand the in vivo molecular mechanisms of tumor heterogeneity. For instance, the widely used xenograft model fails to faithfully recapitulate the tumor microenvironment due to its foreign and immune-compromised nature (Morgan, 2012). Additionally, while sc-seq has played a crucial role in unveiling the variations among individual tumor clones, conducting in vivo validations subsequently poses a challenge (Lähnemann et al, 2020). To overcome these limitations and comprehensively unravel the mechanisms of intercellular communication during cancer progression, an effective strategy to consider is the integration of tumor heterogeneity models that are genetically traceable and editable with the sc-seq technique.

*Drosophila* has been widely employed as a model organism to address fundamental human cancer biology questions, and numerous signaling pathways initially identified in *Drosophila* were later proved to be utmost relevance in human cancers (e.g., Notch, Wnt, Hedgehog and Hippo pathway) (Mirzoyan et al, 2019; Villegas, 2019). The powerful genetic tools available in *Drosophila* make it an optimal model system for elucidating the mechanisms of oncogenic intercellular communications (Bilder et al, 2021; Enomoto et al, 2018; Fahey-Lozano et al, 2019; Katheder et al, 2017; Liu et al, 2022c). Accumulating evidence suggests that *Drosophila* genetic models faithfully replicate in vivo oncogenic cell-cell interactions, as clones harboring distinct oncogenic mutations can collaboratively promote tumor progression

[1]College of Life Sciences, Zhejiang University, 310058 Hangzhou, China. [2]School of Life Sciences, Westlake University, 310024 Hangzhou, China. [3]Westlake Laboratory of Life Sciences and Biomedicine, 310024 Hangzhou, China. [4]College of Traditional Chinese Medicine, North China University of Science and Technology, 063210 Tangshan, China. [5]College of Life Sciences, Hunan Normal University, 410081 Changsha, China. [6]Department of Hepatobiliary Surgery, The Second Hospital, Cheeloo College of Medicine, Shandong University, 250033 Jinan, China. [7]These authors contributed equally: Sihua Zhao, Yifan Guo. ✉E-mail: kongdu@email.sdu.edu.cn; maxianjue@westlake.edu.cn

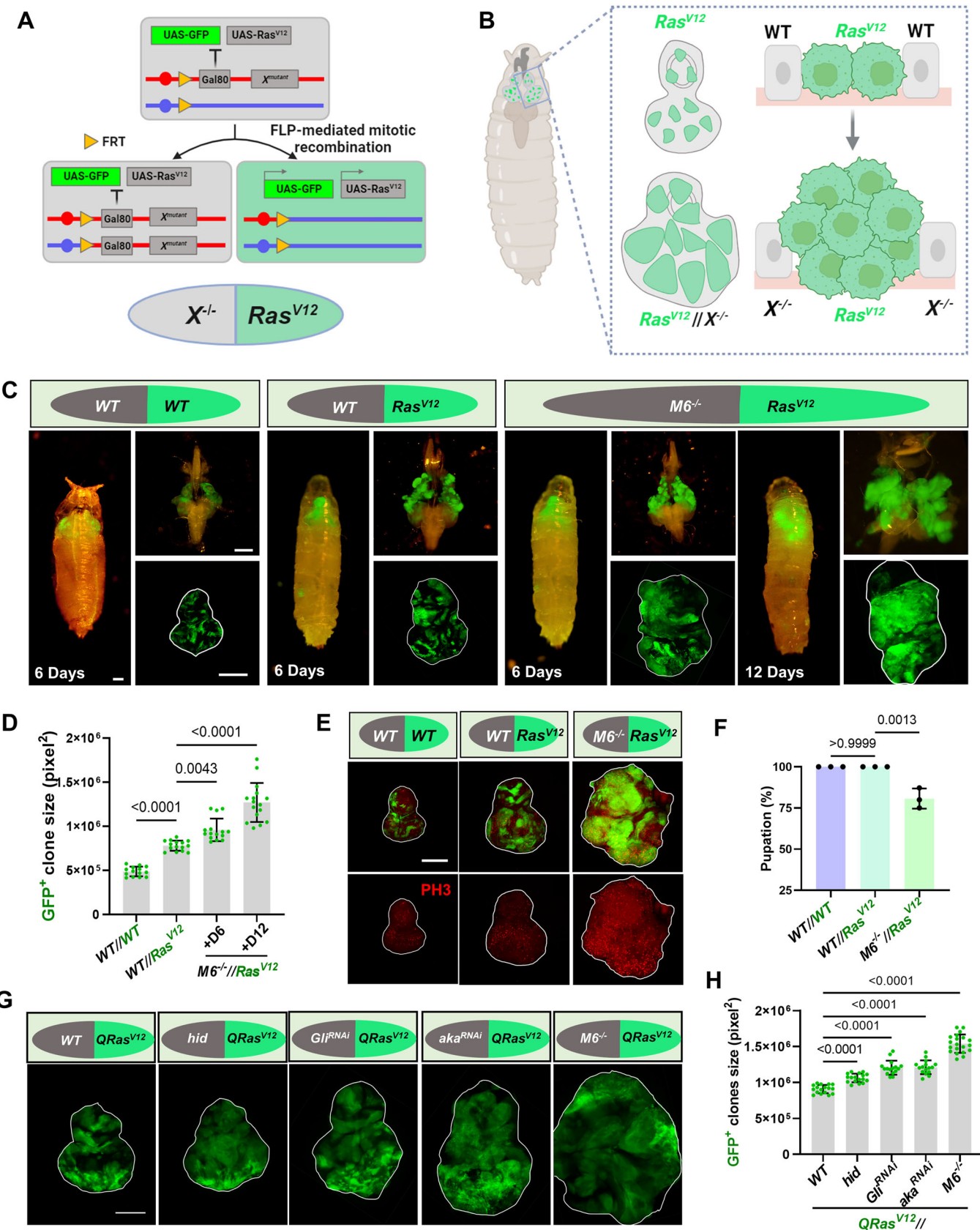

**Figure 1.   Interclonal cooperation between *Ras^V12* and *M6^−/−* promotes tumor malignancy.**

(A) Cartoon illustration of the genetic basis of the mutation of gene X (on right arm of chromosome 3, as an example) surrounding the *Ras^V12* benign tumors using the modified version of MARCM system. (B) Cartoon illustration of the strategy of generating heterogeneity tumor model in *Drosophila*. (C) Representative images of larva/pupa (left), cephalic complex (right, top) and eye-antennal disc (right, bottom) bearing GFP-labeled wild-type clones (green)//non-GFP-labeled wild-type clones (*WT//WT*), *Ras^V12*-expressing clones (green)//wild-type clones (*Ras^V12//WT*), *Ras^V12*-expressing clones (green)//M6 mutant clones (*Ras^V12//M6^−/−*). The time points "6 Days" and "12 Days" correspond to 6- and 12-days post egg-laying, respectively. White lines demarcate the borders of the eye-antennal discs. (D) Quantification of GFP⁺ clones' size with indicated genotypes (from left to right, *n* = 14, 15, 15, 16). Statistical analysis by ordinary one-way ANOVA test; mean ± SD. (E) Eye-antennal discs bearing clones of *WT//WT*, *Ras^V12//WT* and *Ras^V12//M6^−/−* were stained with anti-PH3 antibody. Eye-antennal discs were dissected from *WT//WT* and *Ras^V12//WT* at day 6 after egg laying (AEL-6), and from *Ras^V12//M6^−/−* at AEL-7 (late stage). The white lines outline the borders of the eye-antennal discs. (F) Quantification of pupation ratios of larvae with indicated genotypes: *WT//WT*, *Ras^V12//WT*, *Ras^V12//M6^−/−* (total number pooled from three independent experiments, from left to right, *n* = 212, 224, 230). Statistical analysis by ordinary one-way ANOVA test; mean ± SD. (G) Eye-antennal discs bearing *WT//QRas^V12* clones (AEL-6) or *hid//QRas^V12*, *Gli^RNAi//QRas^V12*, *aka^RNAi//QRas^V12* and *M6^−/−//QRas^V12* clones (all at AEL-7, late stage). White lines demarcate the borders of the eye-antennal discs. (H) Quantification of GFP⁺ clone sizes for the indicated genotypes (from left to right, *n* = 17, 17, 16, 16, 20). Statistical analysis by ordinary one-way ANOVA test; mean ± SD. Exact *P* values are shown in the corresponding panels. Scale bars: 200 μm (C, E, G). Source data are available online for this figure.

(Chatterjee et al, 2023; Enomoto and Igaki, 2022). For example, oncogenic *Ras* (*Ras^V12*) clones with defects in the mitochondrial respiratory complex could lead to the malignancy of neighboring benign *Ras^V12* tumors through the induction of Unpaired (Upd, an IL-6 homolog) and Wingless (Wg, a Wnt homolog) (Ohsawa et al, 2012). Likewise, benign *Ras^V12* tumors exhibit a shift towards malignancy upon being encompassed by clones featuring over-expression of the oncoprotein *Src* or clones that are deficient in polarity due to *scribble* (*scrib*) mutations (Enomoto et al, 2021; Wu et al, 2010).

Tricellular junction (TCJ) proteins are specialized proteins that are located at the points where three epithelial or endothelial cells meet. They are crucial for maintaining the integrity and function of tissues, and dysregulation of these proteins can lead to various pathological conditions, including deafness and certain types of cancer (Bosveld et al, 2018; Higashi and Chiba, 2020). In *Drosophila*, the key TCJ proteins include Anakonda/bark beetle (Aka/bark), Gliotactin (Gli), sidekick (sdk), and M6. These proteins regulate a range of biological activities, such as disengagement of daughter cells during cytokinesis, TCJ assembly, positioning of photoreceptor neurons, and cell contraction (de Bournonville and Le Borgne, 2020; Malin et al, 2022; Wang et al, 2018; Wittek et al, 2020). Interestingly, mutations in *M6* could synergize with *Ras^V12* to drive cell-autonomous overgrowth and apical cell delamination (Dunn et al, 2018). While prior studies demonstrated that clonal *scrib* mutations (a polarity/bicellular junction gene) surrounding *Ras^V12*-expressing clones promote tumorigenesis (Wu et al, 2010), it remains unknown whether TCJ-specific perturbations, including M6, could affect heterogeneity-driven tumor progression.

In this study, using the *Drosophila* eye-antennal imaginal epithelium, we discovered that clones bearing *M6* mutations could promote the tumor progression of neighboring benign *Ras^V12* tumors. Our bulk RNA-seq and single-cell RNA sequencing (scRNA-seq) analyses revealed that subclonal cooperation-induced malignancy was due to the activation of the innate immune-responding Toll pathway, which in turn activates the JNK-Hippo signaling cascade. Notably, we revealed the specific upregulation of the Toll ligand, Spz, in tumor-associated hemocytes. Moreover, we demonstrated that hemocytes derived Spz is both necessary and sufficient for the malignant transformation of *Ras^V12* tumors. In summary, our findings shed light on a previously unrecognized role of the Toll pathway in promoting tumorigenesis, while providing a mechanistic understanding of how hemocytes regulate epithelial tumor heterogeneity by hijacking the innate immune system.

# Results

## Interclonal cooperation between *Ras^V12* and *M6^−/−* promotes tumor malignancy

To establish a tumor heterogeneity model in the eye-antennal imaginal discs of *Drosophila* larvae, we modified the widely used genetic recombinase system known as MARCM (mosaic analysis with a repressible cell marker), which was originally designed to generate positively labeled homozygous mutant clones (Lee and Luo, 1999). We adapted this system by recombining the mutations of interest (*X*) with the corresponding Gal80 onto the same chromosome arm (Fig. 1A). This design facilitates the generation of two neighboring daughter cells with different genotypes after mitosis: one GFP negative clone carrying the homozygous mutant of *X*, and one GFP positive clone with ectopic *Ras^V12* expression (Fig. 1A,B). *M6* mutant clones exhibited cell-autonomous apoptosis and were at least partially eliminated through a caspase-dependent mechanism (Fig. EV1A,B). In contrast, *Ras^V12* clonal expression alone induced hyperproliferation but retained benign tumor characteristics (Fig. 1C). Remarkably, we observed that *Ras^V12*-expressing clones underwent malignant transformation when *M6* was specifically mutated in neighboring cells (hereafter referred to as *Ras^V12//M6^−/−*) (Fig. 1C). Compared to *Ras^V12* alone, *Ras^V12//M6^−/−* tumors exhibited significant tumor overgrowth (Fig. 1D), increased cell-autonomous proliferation (Fig. 1E), and reduced pupation rates (Fig. 1F). This malignant progression was contingent upon spatial interaction between GFP⁺ *Ras^V12* clones and adjacent *M6^−/−* tissue, as *M6* overexpression specifically in *M6* mutant clones dramatically rescued the overgrowth of neighboring GFP-labeled *QRas^V12* tumors (Fig. EV1C,D), confirming that the mutation of *M6* is indispensable for the tumorous growth of surrounding *Ras^V12* clones.

Given that *M6* is a tricellular junction (TCJ) component and its depletion induces apoptosis, we explored two potential mechanisms underlying the phenotypic differences between *Ras^V12//WT* and *Ras^V12//M6^−/−* mosaic tissue: (1) TCJ dysfunction and (2) apoptosis-induced compensatory proliferation. Strikingly, using the Gal4-UAS/Q-MARCM dual system to knock down two other TCJ genes, *Gli* and *aka*, in cells neighboring *Ras^V12* clones (*QRas^V12//Gli^RNAi*, *QRas^V12//aka^RNAi*) phenocopied the *M6* loss-of-function phenotype, driving overgrowth of adjacent *QRas^V12* clones (Fig. 1G,H). Conversely, inducing apoptosis by over-expressing the pro-apoptotic gene *hid* (*head involution defective*) in cells surrounding *Ras^V12* clones triggered only mild

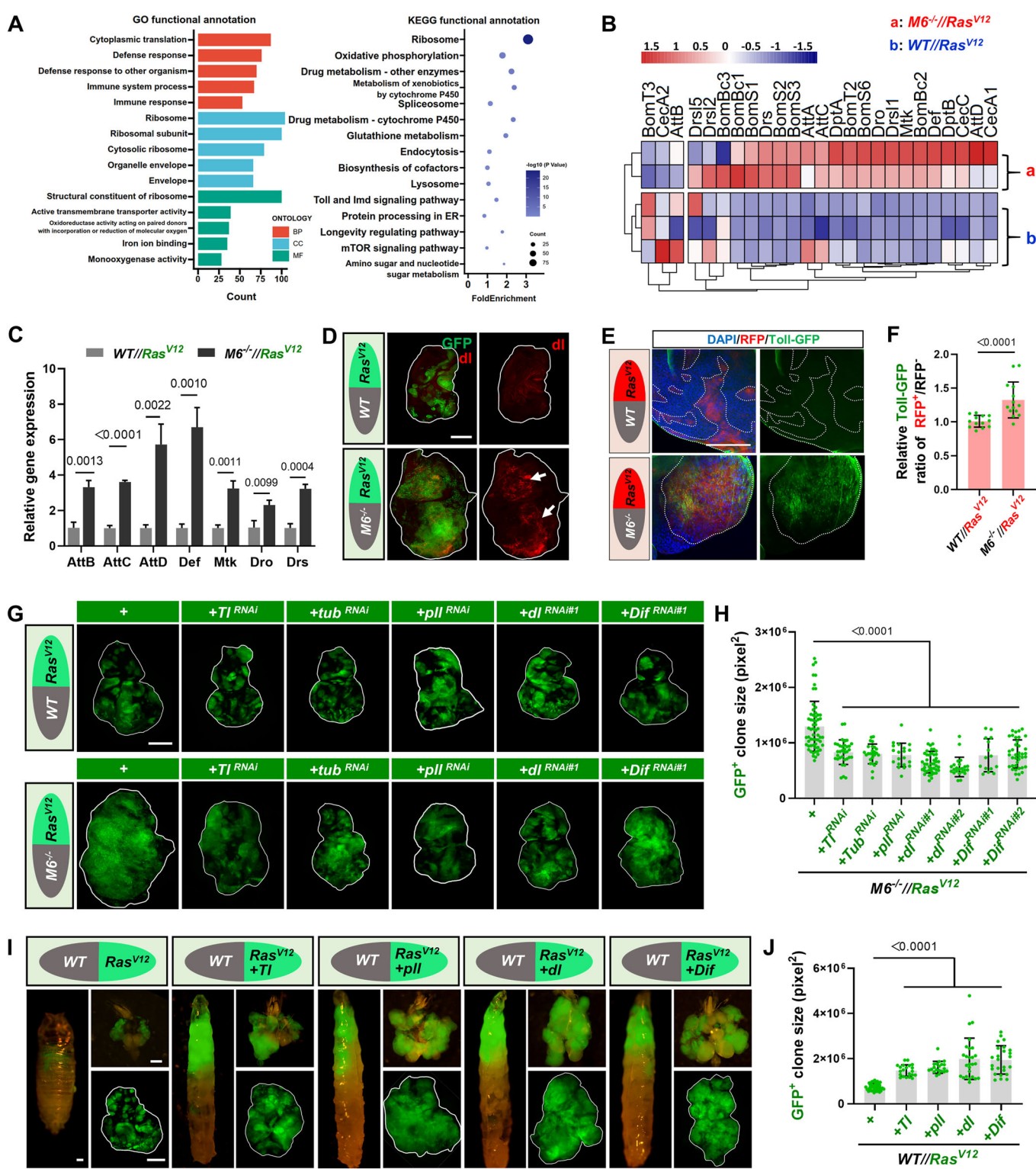

compensatory proliferation, with no significant change in overall eye-disc size, indicating that heterogeneous apoptosis induction alone does not robustly promote oncogenesis (Figs. 1G,H and EV1E). Collectively, these findings highlight TCJ architectural disruption as a key factor enabling interclonal oncogenic cooperation with $Ras^{V12}$.

### $Ras$ synergizes with Toll signaling to promote malignancy in $Ras^{V12}//M6^{-/-}$ tumors

Next, to dissect the molecular mechanism that regulates the malignant transformation of $Ras^{V12}$ tumors, we performed bulk RNA-sequencing analysis on both late-stage $Ras^{V12}$ tumors and

◄ **Figure 2. $Ras^{V12}//M6^{-/-}$ promotes tumor malignancy via activating the Toll signaling pathway.**

(A) The top five enriched GO regulon terms and the top 15 enriched KEGG terms derived from 1,504 up-regulated genes by comparing $Ras^{V12}//M6^{-/-}$ with $Ras^{V12}//WT$ samples. The hypergeometric distribution examinations were performed, with $P < 0.05$ serving as the filtration threshold. Three GO categories are indicated by different colors. The $-log10$ transformed $P$ value of each KEGG term is indicated by scaled color, and the dot size reflects the gene count of each KEGG term. (B) Heatmap showing the normalized expression of target genes associated with Toll and Imd signaling pathways in $Ras^{V12}//M6^{-/-}$ and $Ras^{V12}//WT$ samples. Normalized gene expression value for the heatmap was calculated as $log_2(CPM + 1)$ and scaled by column. (C) qPCR to determine relative AMPs expression level of $Ras^{V12}//WT$ and $Ras^{V12}//M6^{-/-}$ tumors ($n = 3$ independent experiments). Statistical analysis by unpaired two-tailed Student's $t$ test; mean ± SD. (D) Eye-antennal discs bearing clones of $Ras^{V12}//WT$ (AEL-6) and $Ras^{V12}//M6^{-/-}$ (AEL-7, late stage) were stained with dl antibody. The arrows indicate increased fluorescence signals. White lines demarcate the borders of the eye-antennal discs. (E) Eye-antennal discs contain RFP-labeled clones expressing $Ras^{V12}$ (red) surrounding wild-type ($Ras^{V12}//WT$) clones at AEL-6, or M6 mutant clones ($Ras^{V12}//M6^{-/-}$) at AEL-7 (late stage). GFP signals represent the expression of endogenous Toll. White dashed lines mark the boundaries of GFP-positive clones. (F) Quantification of relative Toll-GFP fluorescence intensity (from left to right, $n = 13, 14$). Statistical analysis by unpaired non-parametric Mann–Whitney test; mean ± SD. (G) Representative immunofluorescent images of eye-antennal discs with the following genotypes: $Ras^{V12}//WT$ (AEL-6), $Ras^{V12}$ combined with RNAi against $Tl$, $Tub$, $pll$, $dl$, or $Dif$ in WT background (all AEL-6), and their corresponding $M6^{-/-}$ background counterparts (all AEL-7, late stage). White outlines denote the borders of the eye-antennal discs. (H) Quantification of GFP$^+$ clone sizes across different genotypes (from left to right, $n = 60, 34, 26, 20, 44, 23, 15, 39$). Statistical analysis by ordinary one-way ANOVA test; mean ± SD. (I) Representative images of late-stage larva/pupa, cephalic complex, and eye-antennal discs with the following genotypes: $Ras^{V12}//WT$ (AEL-6), $Ras^{V12} + Tl//WT$ (AEL-8), $Ras^{V12} + pll//WT$ (AEL-8), $Ras^{V12} + dl//WT$ (AEL-8), $Ras^{V12} + Dif//WT$ (AEL-8). White outlines denote the borders of the eye-antennal discs. (J) Quantification of GFP-positive clone sizes across different genotypes (from left to right, $n = 42, 21, 20, 26, 25$). Statistical analysis by ordinary one-way ANOVA test; mean ± SD. Exact $P$ values are shown in the corresponding panels. Scale bars: 50 μm (E) and 200 μm (D, G, I). Source data are available online for this figure.

$Ras^{V12}//M6^{-/-}$ tumors. The Gene Ontology (GO) and Kyoto Encyclopedia of Genes and Genomes (KEGG) analyses against 1504 upregulated genes revealed a significant enrichment of defense response, immune response and immune related signaling pathways (Fig. 2A). In *Drosophila*, there are two major innate immunoregulatory pathways known as the Toll pathway and the immune deficiency (Imd) pathway, they collectively control the systemic production of antimicrobial peptides (AMPs) to combat microbial infection (Lemaitre and Hoffmann, 2007; Lemaitre et al, 1996; Lemaitre et al, 1997). We observed a marked increase in multiple AMPs in the $Ras^{V12}//M6^{-/-}$ tumors (Fig. 2B), which was subsequently confirmed through qRT-PCR analysis (Fig. 2C). Interestingly, a strong upregulation of dorsal (Spannl et al, 2020), the downstream transcription factor of the Toll pathway (Fig. 2D), and increased expression of endogenous Toll (Fig. 2E,F) were observed in $Ras^{V12}//M6^{-/-}$ tumors, while there was no change in the expression of Relish (Rel), the downstream transcription factor of the Imd pathway (Fig. EV1F), indicating that Toll pathway might be more important in $Ras^{V12}//M6^{-/-}$ tumors.

The Toll pathway can be canonically activated by Gram-positive bacterial infections and fungal infections (De Gregorio et al, 2002; Lemaitre et al, 1996). To explore whether the activation of Toll signaling induced by $Ras^{V12}//M6^{-/-}$ is a consequence of infection, we utilized axenic cultures and antibiotic cocktail treatment to deplete the bacterial microbiome to below detectable levels (Fig. EV1G). Interestingly, the elimination of bacteria further enhanced the overgrowth of $Ras^{V12}//M6^{-/-}$ tumor (Fig. EV1H,I), suggesting a potential growth-inhibition role of bacteria in $Ras^{V12}//M6^{-/-}$ tumor-bearing flies. Additionally, we observed that despite a significant reduction in *Drs* expression, which is a target AMP-coding gene of the Toll signaling pathway, following antibiotic treatment, $Ras^{V12}//M6^{-/-}$ tumor still exhibited higher *Drs* expression levels compared to $Ras^{V12}$ tumor (Fig. EV1J), implying that the interclonal cooperation between $Ras^{V12}$ and $M6^{-/-}$ can induce Toll activation independently of bacterial infection.

To further dissect the potential role of AMPs in $Ras^{V12}//M6^{-/-}$-induced tumorigenesis, we deleted one copy of *Defensin* (*Def*), an AMP previous implicated in fly tumor progression (Parvy et al, 2019). Additionally, we utilized a mutant strain deficient in multiple key AMPs, where one copy of each AMP gene (*Def*, *AttC*, *Dro*, *AttA*, *AttB*, and *Dpt*) was deleted in

$Ras^{V12}//M6^{-/-}$ tumors. However, neither heterozygous mutations in *Def* nor the simultaneous knockout of multiple AMPs rescued the overgrowth phenotype of $Ras^{V12}//M6^{-/-}$ tumors (Fig. EV1K,L). These findings suggest that AMP induction in $Ras^{V12}//M6^{-/-}$ tumors is associated with non-specific immune activation rather than a direct role in oncogenesis. While AMPs are transcriptionally induced in $Ras^{V12}//M6^{-/-}$ tumors, their functional role in this context requires further investigation. Given the complexity of AMP-immune crosstalk, a further comprehensive mechanistic analysis is required in future studies.

To further explore the role of the Toll pathway in $Ras^{V12}$ tumor malignancy, we inhibited several key components of Toll signaling, including *Toll* (*Tl*), *tube* (*tub*), *pelle* (*pll*), *dorsal* (*dl*), and *Dorsal-related immunity factor* (*Dif*). These interventions had minimal impact on benign $Ras^{V12}$ tumor growth (Figs. 2G and EV2A–D) but significantly suppressed the progression of $Ras^{V12}//M6^{-/-}$ tumors (Figs. 2G,H and EV2E,F). Remarkably, genetic activation of Toll signaling by co-expressing an activated form of *Tl*, the protein kinase *pll*, the transcription factors *dl* or *Dif* phenocopied the effects of $Ras^{V12}//M6^{-/-}$ tumors, transforming $Ras^{V12}$ tumors into highly proliferative malignant tumors (Figs. 2I,J and EV2G,H). Notably, clonal overexpression of *pll*, *dl*, or *Dif* significantly increased clone area compared to wild-type controls (Fig. EV2C,D), indicating that Toll pathway activation promotes proliferation even in the absence of oncogenic $Ras^{V12}$. These findings expand our understanding of Toll signaling beyond its canonical immune functions and highlight its potential interplay with oncogenic processes in tumor-immune crosstalk. However, inhibition of the Imd pathway by knocking down downstream transcription factor *Rel* failed to suppress the overgrowth of $Ras^{V12}//M6^{-/-}$ tumors (Fig. EV2I,J). Collectively, these results suggest that Toll pathway activation is both necessary and sufficient for the malignant transformation of $Ras^{V12}$ tumors, highlighting its critical role in tumor progression.

## Hippo signaling pathway inactivation promotes $Ras^{V12}//M6^{-/-}$ tumor proliferation

To further elucidate the mechanisms underlying tumor malignancy driven by interclonal communications in $Ras^{V12}//M6^{-/-}$ tumors, and to specifically dissect the changes occurring within different subclones during the early stages of tumorigenesis, we performed

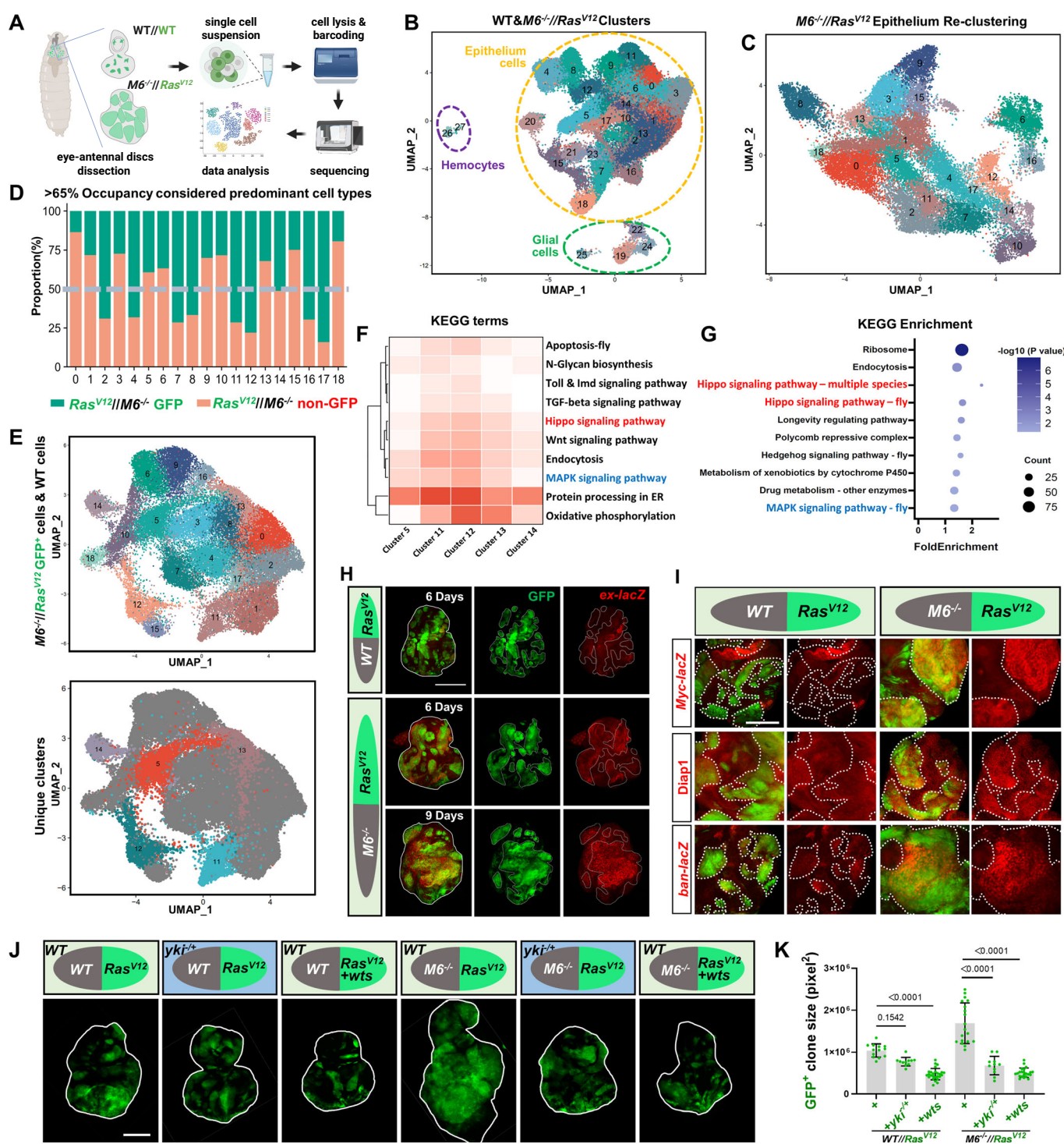

scRNA-seq analysis on eye-antennal discs dissected from wild-type (WT) and early-stage $Ras^{V12}//M6^{-/-}$ larvae (Fig. 3A). Based on the similarities in the gene expression profiles of high-quality cells, 32,225 cells from WT and 43,992 cells from $Ras^{V12}//M6^{-/-}$ tumors were collected and assembled into 27 clusters that were categorized into three groups: epithelium cells, hemocytes, and glial cells (Figs. 3B and EV3A). Next, we extracted $Ras^{V12}//M6^{-/-}$ tumor derived epithelial cells and re-clustered them into 19 clusters for

further analysis (Fig. 3C). Using the expression profiles of exogenously introduced *GFP* and *Ras85D* markers (Fig. EV3B), we applied a previously established methodology (Chatterjee et al, 2022; Lee et al, 2020) to define a threshold of 65% occupancy, enabling the identification of clusters predominantly composed of GFP-labeled tumor cells within the $Ras^{V12}//M6^{-/-}$ scRNA-seq datasets (cluster 2, 4, 7, 8, 11, 12, 16 and 17) (Fig. 3D). To identify differences between $Ras^{V12}//M6^{-/-}$ GFP cells and WT cells,

◄

**Figure 3. Hippo signaling pathway is inactivated in *Ras^V12*//*M6^−/−* tumors.**

(A) Scheme illustration of the experimental workflow for preparing *Drosophila* tumor samples and performing single-cell RNA-seq analysis. (B) Uniform manifold approximation and projection (UMAP) plot showing integrated WT and *Ras^V12*//*M6^−/−* single-cell datasets. A total of 32,225 cells from WT datasets and 43,992 cells from *Ras^V12*//*M6^−/−* datasets were integrated using 2000 anchor genes. Cell types were annotated using known markers. Clusters 26 and 27 correspond to *Drosophila* hemocytes, clusters 19, 22, 24, and 25 represent glial cells (perineurial glia, wrapping glia, and sub-perineural glia), while the remaining clusters consist of a mix of photoreceptors, ommatidial and interommatidial cells, undifferentiated cells, and other cell types that collectively form the pseudostratified columnar epithelium of the eye-antennal disc. (C) UMAP plot showing cellular heterogeneity among re-clustered *Ras^V12*//*M6^−/−* epithelial cells. Each dot represents an individual cell, with distinct colors indicating different cell populations. A total of 19 clusters were generated (resolution = 1.3). (D) Bar plot showing the fraction of GFP-positive cells and non-GFP cells in each cluster. Clusters having at least 330 GFP⁺ cells and over 65% GFP⁺ cell proportion were classified as GFP⁺-dominant clusters. (E) Upper panel: UMAP plot illustrating the integration of *Ras^V12*//*M6^−/−* GFP-positive epithelial cells and WT epithelial cells. A total of 19 clusters were generated (resolution = 1.2). Bottom panel: UMAP plot showing clusters unique to the *Ras^V12*//*M6^−/−* GFP⁺ cells compared to WT cells, including cluster 5, 11, 12, 13, and 14. (F) Ten overlapped enriched KEGG terms between clusters 5, 11, 12, 13, and 14. (G) KEGG enrichment of all DEGs from bulk RNA-seq comparing *Ras^V12*//*M6^−/−* and *Ras^V12*//*WT*. The hypergeometric distribution examinations were performed, with *P* < 0.05 serving as the filtration threshold. The −log10 transformed *P* value of each KEGG term is indicated by scaled color, and the dot size reflects the gene count of each KEGG term. (H) Representative immunofluorescent images showing *ex-lacZ* staining in eye-antennal discs bearing *Ras^V12*//*WT* and *Ras^V12*//*M6^−/−* clones at 6 days and 9 days after egg laying. White lines outline the borders of the eye-antennal discs, while white dashed lines indicate the boundaries of GFP-positive clones. (I) Representative immunofluorescent images showing *Myc-lacZ*, Diap1, and *ban-lacZ* staining in eye-antennal discs bearing *Ras^V12*//*WT* and *Ras^V12*//*M6^−/−* clones at AEL-6. White dashed lines mark the boundaries of the GFP-positive clones. (J) Eye-antennal discs with *Ras^V12*//*WT*, *Ras^V12*+*yki−/+*//*yki−/+*, and *Ras^V12*+*wts*//*WT* clones (AEL-6), and corresponding *M6^−/−* background genotypes: *Ras^V12*//*M6^−/−*, *Ras^V12*+*yki−/+*//*M6^−/−*+*yki−/+*, *Ras^V12*+*wts*//*M6^−/−* (AEL-7, late stage). White lines outline the borders of the eye-antennal discs. (K) Quantification of GFP⁺ clone sizes for the indicated genotypes (from left to right, *n* = 13, 11, 26, 19, 11, 22). Statistical analysis by ordinary one-way ANOVA test; mean ± SD. Exact *P* values are shown in the corresponding panels. Scale bars: 100 μm (I), 200 μm (H, J). Source data are available online for this figure.

GFP-positive cells from these clusters were isolated. Subsequently, these GFP-positive cells were co-integrated along with WT epithelial cells and subjected to re-clustering, resulting in 19 distinct cell subpopulations (Fig. 3E). Among these, five clusters, including 5, 11, 12, 13, and 14, were identified as unique to *Ras^V12*//*M6^−/−* tumors (Fig. 3E). Differentially expressed genes (DEGs) within each cluster were subjected to KEGG pathway analysis, revealing several overlapping regulons across different clusters (Fig. 3F). Notably, aside from enrichment in Toll and Imd signaling pathways, we detected significant enrichment of the Hippo signaling pathway (Fig. 3F), a master regulator of tissue growth and tumorigenesis (Kulkarni et al, 2020; Ma et al, 2019; Ma et al, 2020; Ma et al, 2018; Song and Ma, 2023). Further analysis of DEGs derived from bulk RNA-seq data of *Ras^V12*//*M6^−/−* tumors corroborated significant enrichment of Hippo signaling compared to *Ras^V12* tumors (Fig. 3G).

Next, we investigated the in vivo role of Hippo signaling in *Ras^V12*//*M6^−/−*-induced tumor progression. *Ras^V12*//*M6^−/−* tumors displayed significant upregulation of multiple Hippo pathway target genes, including the specific target *expanded* (*ex*) (Fig. 3H), and other targets including *Myc*, *Death-associated inhibitor of apoptosis 1* (*Diap1*), *bantam* (*ban*), *Cyclin E* (*CycE*), *wingless* (*wg*), and *four-joint* (*fj*) (Figs. 3I and EV3D), as confirmed by qRT-PCR analysis (Fig. EV3E). Notably, we observed an apparent discrepancy between Diap1 protein and mRNA levels. While *Diap1* mRNA is known to be directly induced by the Yki/Sd transcriptional complex(Wu et al, 2008), the protein level of Diap1 can be regulated post-transcriptionally, including pro-apoptotic gene-mediated ubiquitination (Goyal et al, 2000; Yoo et al, 2002). Indeed, despite the overgrowth of *Ras^V12* tumor clones in *Ras^V12*//*M6^−/−* tissues, the transcription of *reaper* (*rpr*), a pro-apoptotic gene, was robustly induced specifically in GFP⁺ tumor clones (Fig. EV3F). Therefore, the modest increase in Diap1 protein despite robust *Diap1* mRNA upregulation reflects a dynamic equilibrium between Yki-mediated transcriptional activation and Rpr/Rpr-like-mediated protein degradation. The key components of Hippo pathway consist of *hippo* (*hpo*), *warts* (*wts*), and *yorkie* (*yki*) [reviewed in (Zheng and Pan, 2019)]. Hpo phosphorylates

Wts, which subsequently phosphorylates and inactivates Yki. We observed that inhibiting Yki activity through either the removal of one copy of endogenous *yki* or ectopic expression of *wts* significantly suppressed the overgrowth of *Ras^V12*//*M6^−/−* tumors (Fig. 3J,K). These findings suggest that Hippo pathway is inactivated in *Ras^V12*//*M6^−/−* tumors, and activation of Hippo signaling can effectively suppress *Ras^V12*//*M6^−/−*-induced tumor progression. In addition, scRNA-seq data revealed elevated expression of several Toll receptors, including *18w* (*Toll-2*), *Toll-7*, and *Tollo*, as well as the ligand *spz3*, in GFP-positive *Ras^V12* tumor cells, alongside a mild upregulation of *Tl* (Fig. EV3G).

## *Ras^V12*//*M6^−/−* promotes tumor malignancy through Toll-JNK-Hippo signaling cascade

Next, we investigated the genetic interactions between Toll and Hippo pathways in *Ras^V12*//*M6^−/−* tumors. Antibody staining and qPCR analysis revealed that *Ras^V12* tumors with increased Toll pathway activity significantly increased the transcription of Hippo pathway target genes, including Diap1, *ban-lacZ*, and Wg (Figs. 4A,B and EV4A). Consistently, inhibiting Yki activity by co-expressing *wts* or *hpo* significantly impeded the synergistic tumor-promoting effect between *Ras^V12* and Toll activation (Fig. 4C,D), and restored the pupation defects (Fig. 4E). On the contrary, inhibiting Toll activity did not affect the overgrowth phenotype induced by co-expressing *Ras^V12* and *yki^S168A* (Fig. 4F,G). Together, these genetic data demonstrate that Toll pathway activation drives tumor malignancy by inhibiting the Hippo pathway within the *Ras^V12* clones of the *Ras^V12*//*M6^−/−* heterogeneous tumors.

To further dissect the mechanisms by which the Toll pathway modulates Hippo and Yki activity, we performed clonal over-expression of *dl* or *Dif* under physiological conditions and analyzed the expression of Hippo target genes. Surprisingly, while *dl*/*Dif* overexpression significantly induced tissue overgrowth, it paradoxically reduced the protein levels of Hippo target genes Diap1 and CycE (Fig. EV4B,C). This apparent contradiction highlights context-dependent modulation of Hippo/Yki activity by Toll

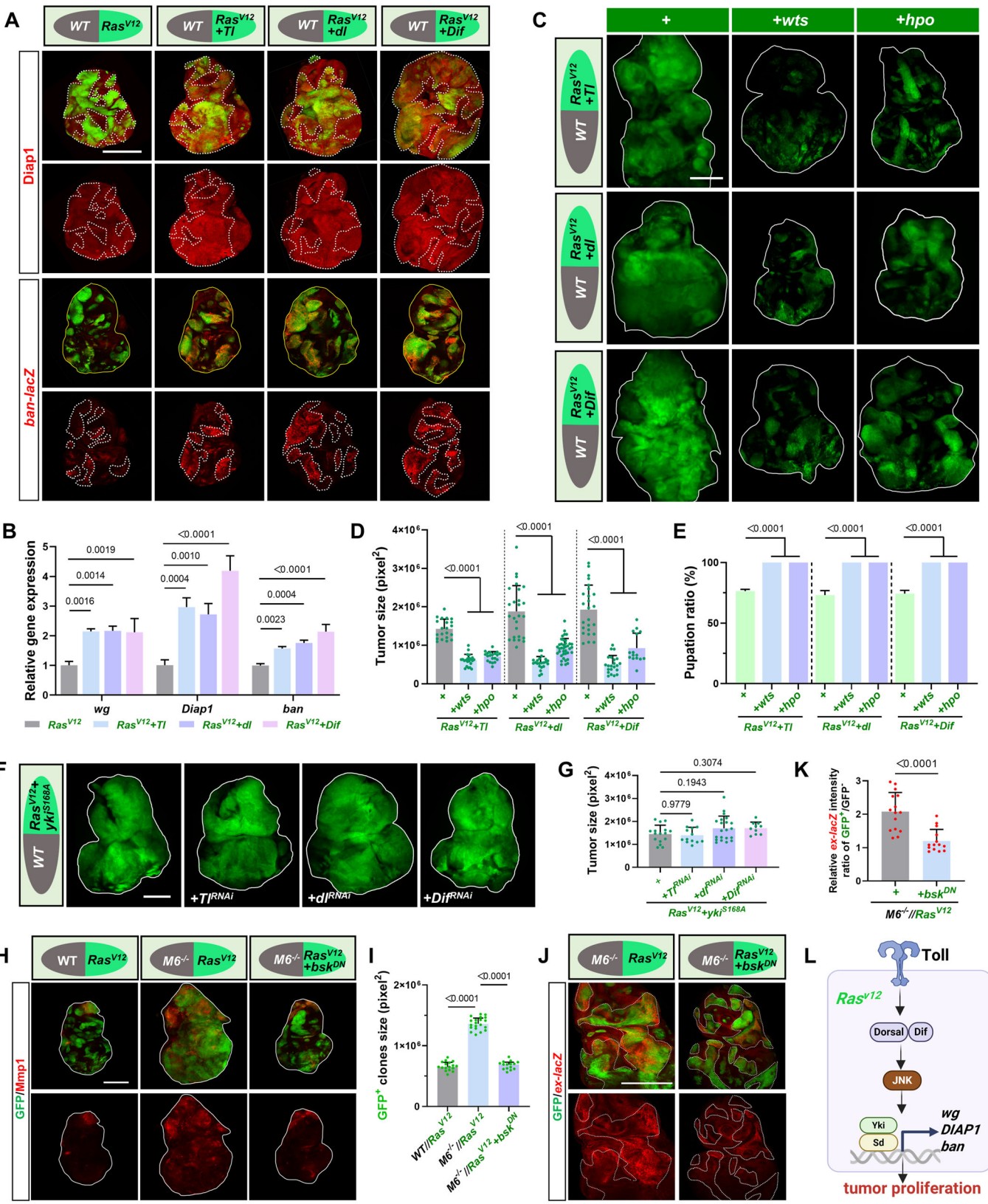

**Figure 4. Activation of Toll signaling promotes the overgrowth of Ras^V12 clones through JNK-mediated Hippo pathway inactivation.**

(A) Eye-antennal discs bearing clones of *Ras^V12*//*WT* (AEL-6), or *Ras^V12* combined with *Tl*, *dl*, or *Dif* in *WT* background (AEL-7), were stained with anti-Diap1 antibody (upper panels). The same genotypes (all AEL-6) were stained with anti-β-galactosidase antibody to visualize the *ban-lacZ* reporter (lower panels). The yellow lines outline the borders of the eye-antennal discs, the white dashed lines mark the boundaries of the GFP-positive clones. (B) qPCR analysis to determine the mRNA levels of *yki* target genes (*wg*, *Diap1*, and *ban*) in eye-antennal discs of indicated flies ($n = 3$ independent experiments). Statistical analysis by ordinary one-way ANOVA test; mean ± SD. (C) Eye-antennal discs bearing clones of *Ras^V12* + *Tl*//*WT*, *Ras^V12* + *Tl*+ *wts*//*WT*, *Ras^V12* + *Tl*+ *hpo*//*WT*, *Ras^V12*+ *dl*//*WT*, *Ras^V12*+ *dl*+ *wts*//*WT*, *Ras^V12*+ *dl*+ *hpo*//*WT*, *Ras^V12*+ *Dif*//*WT*, *Ras^V12*+ *Dif*+ *wts*//*WT*, *Ras^V12*+ *Dif*+ *hpo*//*WT* (all AEL-7, late stage). The white lines outline the borders of the eye-antennal discs. (D) Quantification of GFP$^+$ clone sizes for indicated genotypes (from left to right, $n = 21$, 22, 20, 26, 22, 38, 25, 22, 15). Statistical analysis by ordinary one-way ANOVA test; mean ± SD. (E) Quantification of pupation ratios for flies with indicated genotypes (total number from three independent experiments, from left to right, $n = 229$, 208, 214, 219, 226, 241, 229, 219, 206). Statistical analysis by ordinary one-way ANOVA test; mean ± SD. (F) Eye-antennal discs bearing clones of *Ras^V12*+*yki^{S168A}*//*WT*, *Ras^V12*+*yki^{S168A}* + *Tl^{RNAi}*//*WT*, *Ras^V12*+*yki^{S168A}*+*dl^{RNAi}*//*WT*, *Ras^V12*+*yki^{S168A}*+*Dif^{RNAi}*//*WT* (all AEL-7, late stage). White lines outline the borders of the eye-antennal discs. (G) Quantification of tumor size with indicated genotypes (from left to right, $n = 16$, 13, 21, 10). Statistical analysis by ordinary one-way ANOVA test; mean ± SD. (H) Eye-antennal discs bearing clones of *Ras^V12*//*WT* (AEL-6), *Ras^V12*//*M6^{-/-}* (AEL-7), *Ras^V12*+*bsk^{DN}*//*M6^{-/-}* (AEL-7) were stained with anti-Mmp1 antibody. White lines outline the borders of the eye-antennal discs. (I) Quantification of GFP$^+$ clone sizes with indicated genotypes (from left to right, $n = 17$, 20, 15). Statistical analysis by ordinary one-way ANOVA test; mean ± SD. (J) Eye-antennal discs bearing clones of *Ras^V12*//*M6^{-/-}* and *Ras^V12*+*bsk^{DN}*//*M6^{-/-}* were stained with anti-β-galactosidase antibody at AEL-6 to visualize the *ex-lacZ* reporter. White dashed lines mark the boundaries of the GFP-positive clones. (K) Quantification of *ex-lacZ* staining intensity in disc with indicated genotypes (from left to right, $n = 15$, 14). Statistical analysis by unpaired two-tailed Student's *t* test; mean ± SD. (L) A model illustrating Toll pathway activation in *Ras^V12* clones promotes malignancy through JNK-dependent transcriptional upregulation of Yki/Sd target genes. Exact *P* values are shown in the corresponding panels. Scale bars: 200 μm (A, C, F, H, J). Source data are available online for this figure.

signaling, aligning with previous findings that Toll suppresses Yki activity in the fat body during microbial infection (Liu et al, 2016), yet promotes Yki-dependent growth via JNK signaling in polarity-deficient cells (Katsukawa et al, 2018). However, the regulatory mechanisms of the Toll pathway on the Hippo pathway under physiological conditions as well as other environmental conditions remain to be more thoroughly and systematically investigated. Specifically, in *Ras^V12*//*M6^{-/-}* tumors, we speculate that the metabolic stress induced by oncogenic *Ras^V12* may override the typical crosstalk between Toll and Yki. These data also underscore the importance of caution when extrapolating the role of Toll signaling from immune contexts to oncogenic settings. Supporting this, KEGG pathway analysis of GFP$^+$ *Ras^V12*//*M6^{-/-}* tumor cells also revealed significant enrichment of MAPK (mitogen-activated protein kinase) signaling components (Fig. 3F,G), while functional validation confirmed JNK activation through Mmp1 upregulation (Fig. 4H). Importantly, genetic inhibition of JNK signaling by expressing a dominant-negative form of *bsk* (*bsk^{DN}*) suppressed *Ras^V12*//*M6^{-/-}* tumorigenesis and abolished the upregulation of Mmp1 and *ex* transcription (Fig. 4H–K). These results indicate that *Ras^V12*//*M6^{-/-}* tumors activate JNK signaling to modulate Hippo pathway activity.

Next, we investigated whether activation of Toll pathway alone is sufficient to activate JNK signaling. Under physiological conditions, ectopic expression of *dl* or *Dif* activates JNK signaling, as evidenced by upregulation of *puc-lacZ* and Mmp1. Under oncogenic stress, Toll pathway activation synergizes with *Ras^V12* to robustly activate JNK signaling (Fig. EV4D). These findings collectively support a model in which *Ras^V12*//*M6^{-/-}* tumors drive tumorigenesis through the Toll-JNK-Hippo signaling cascade (Fig. 4L).

## Spz activates Toll pathway to facilitate malignant transformation of *Ras^V12*//*M6^{-/-}* tumors

Next, we investigated the mechanism by which Toll pathway is activated. Spz is a secreted cytokine and acts as the major ligand for the Toll receptor in *Drosophila*, it plays a crucial role in the immune response and development (De Gregorio et al, 2002;

Lemaitre et al, 1996). Interestingly, we observed that cells with high Spz expression specifically accumulated around *Ras^V12* tumors (Fig. 5A). In contrast, no significant changes in Spz expression were detected in the eye-antennae disc containing only *Ras^V12* clones, *M6^{-/-}* clones, and wild-type clones (Figs. 5A and EV5A). To further examine the role of Spz in *Ras^V12*//*M6^{-/-}* tumor progression, we removed *spz* entirely from the larvae. Remarkably, this resulted in a significant reduction in *Ras^V12*//*M6^{-/-}* tumor size (Fig. 5B,C), accompanied by the suppression of dl upregulation and of Hippo target genes, including *ex-lacZ*, CycE, and Wg (Figs. 5D and EV5B). Importantly, re-activation of the Toll pathway via overexpression of *Tl*, *dl*, or *Dif* in the *spz*-deficient genetic background effectively reversed the tumor suppression phenotype (Fig. 5B,C). These findings imply that Spz is a key regulator of Toll pathway activation within *Ras^V12* clones in *Ras^V12*//*M6^{-/-}* tumors, underscoring its essential role in driving malignant transformation.

## Hemocytes-derived Spz activates the Toll pathway in *Ras^V12* clones within the *Ras^V12*//*M6^{-/-}* tumors

One of the primary sites for the production and activation of the Spz protein in *Drosophila* is the hemocytes (blood cells) (Irving et al, 2005; Shia et al, 2009). In addition to immature hemocytes, mature hemocytes can be categorized into three major types: plasmatocytes, crystal cells, and lamellocytes. Plasmatocytes constitute over 90–95% of the population and are equivalent to mammalian macrophages (Williams, 2007). Indeed, our scRNA-seq data analysis revealed the presence of hemocyte clusters (Fig. 3B, cluster 26 and 27). Compared to the WT group, *Ras^V12*//*M6^{-/-}* tumors exhibited a dramatic increase in hemocytes adhering to tumor sites (Figs. 3B and EV3A). Upon further clustering, four hemocyte subclusters were identified in *Ras^V12*//*M6^{-/-}* tumor samples, with elevated *spz* expression observed across multiple hemocyte clusters, but not within epithelial cells (Fig. EV5C,D). This contrasts with hemocytes attached to *Ras^V12*/*scrib^{-/-}*, *dlg^{-/-}* and *scrib^{-/-}* tumors (Cordero et al, 2010; Parisi et al, 2014), where *egr* expression levels remain unchanged in *Ras^V12*//*M6^{-/-}* tumor-attached hemocytes (Fig. EV5C). This difference indicates that hemocytes recruited by tumors with distinct genetic backgrounds

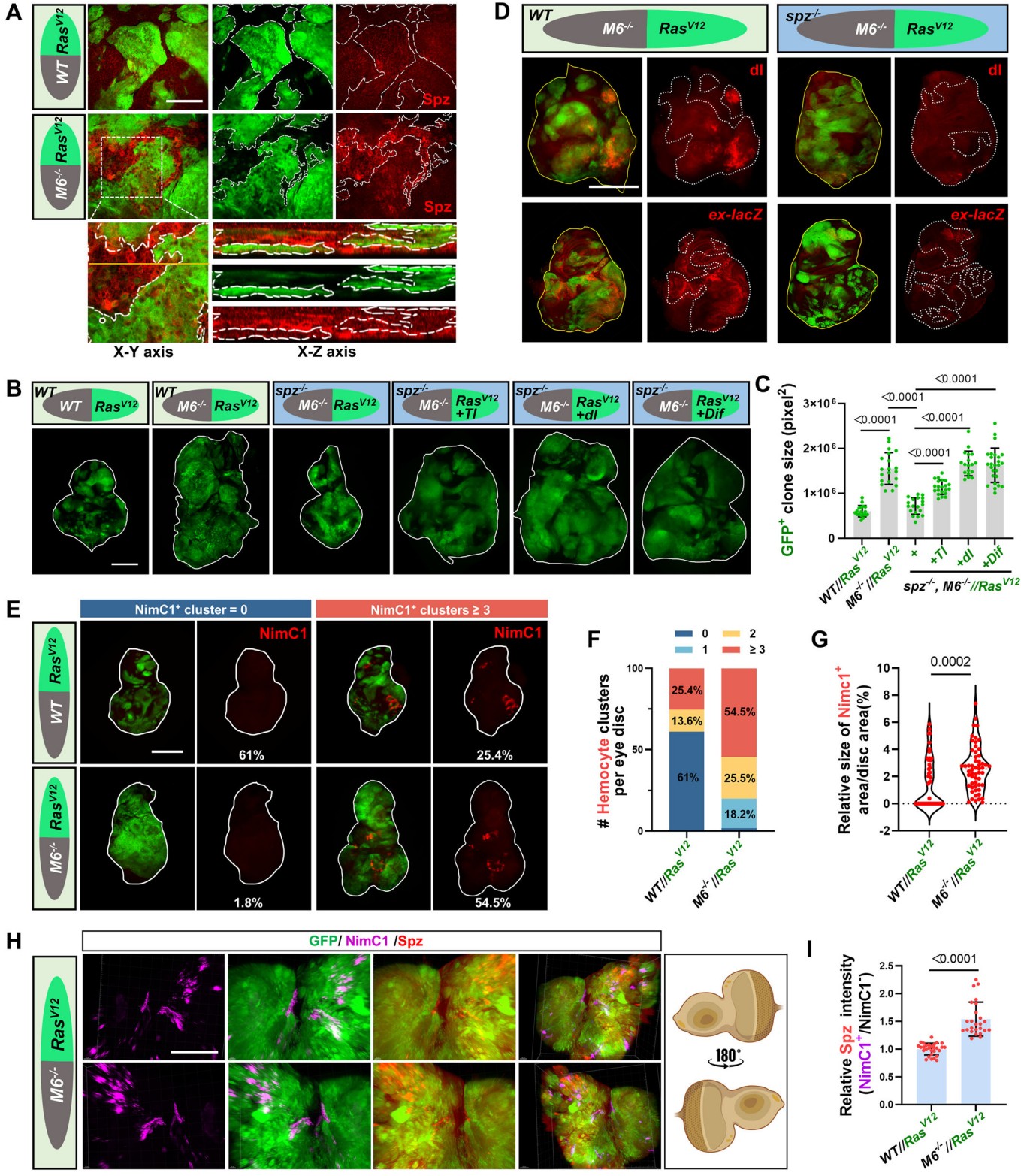

**Figure 5.  *spz* upregulation in hemocytes is essential for Toll pathway activation and malignant transformation of *Ras^V12//M6^−/−* tumors.**

(A) Eye-antennal discs bearing clones of *Ras^V12//WT* and *Ras^V12//M6^−/−* were stained with anti-Spz antibody. Bottom panels show xy and xz across-section views. The yellow line represents the position of lateral section images. (B) Eye-antennal discs bearing clones of *Ras^V12//WT* (AEL-6), and *Ras^V12//M6^−/−*, *Ras^V12+spz^−/−//M6^−/−+spz^−/−*, *Ras^V12+ Tl+ spz^−/−//M6^−/−+spz^−/−*, *Ras^V12+dl+ spz^−/−//M6^−/−+spz^−/−*, *Ras^V12+Dif+ spz^−/−//M6^−/−+spz^−/−* (AEL-7). White lines demarcate the boundaries of the eye-antennal discs. (C) Quantification of GFP^+ clone sizes across different genotypes (from left to right, n = 18, 20, 19, 21, 18, 26). Statistical analysis by ordinary one-way ANOVA test; mean ± SD. (D) Eye-antennal discs bearing clones of *Ras^V12//M6^−/−* and *Ras^V12+spz^−/−//M6^−/−* *spz^−* were stained with anti-dl antibody (right) or anti-β-galactosidase antibody (left) at AEL-6. The β-gal signal marks the *ex-lacZ* reporter. The yellow lines outline the borders of the eye-antennal discs, while the white dashed lines mark the boundaries of the GFP-positive clones. (E) Eye-antennal discs bearing clones of *Ras^V12//WT* and *Ras^V12//M6^−/−* were stained with anti-NimC1 antibody at AEL-6. Phenotype penetrance percentages were observed as 61%, 25.4%, 1.8%, and 54.5%. White lines outline the borders of the eye-antennal discs. (F) Quantification of hemocyte cluster proportions across different genotypes, categorized by the number of hemocyte clusters. Percentages indicate the proportion of hemocyte clusters with corresponding numbers. (G) Quantification of relative total NimC1^+ area normalized to the disc size area across different genotypes (from left to right, n = 50, 56). Violin plots represent kernel density estimation of data distribution, with the width proportional to the number of data points at each Y-value. Scattered red dots depict individual data points, and statistical analysis was performed using an unpaired non-parametric Mann–Whitney test. (H) Eye-antennal disc bearing clones of *Ras^V12//M6^−/−* were stained with both anti-Spz and anti-NimC1 antibodies at AEL-7. Refer to Movie EV1 for additional visualization. (I) Quantification of relative Spz signal intensity across indicated genotypes (from left to right, n = 29, 25). Statistical analysis by unpaired non-parametric Mann–Whitney test; mean ± SD. Exact P values are shown in the corresponding panels. Scale bars: 50 μm (A), 200 μm (D, B, E, H). Source data are available online for this figure.

exhibit differential characteristics. Additionally, analysis of Toll ligands and receptors in *Ras^V12//M6^−/−*-attached hemocytes revealed significant upregulation of *Toll-4* in cluster 0 and cluster 3 (Fig. EV5E). However, the specific role of *Toll-4* in this context remains uncharacterized and warrants further investigation.

The evidence presented above implies that tumor-attached hemocytes might be the source of Spz. To verify this, we examined the abundance of hemocytes by conducting antibody staining against NimC1, a commonly used marker to identify hemocytes adhering to tissue in *Drosophila* (Kurucz et al, 2007). Consistent with our scRNA-seq analysis, we observed increased abundance of hemocytes in *Ras^V12//M6^−/−* tumors (Fig. 5E–G). Additionally, these tumor-attached hemocytes exhibited increased expression of Spz (Fig. 5H,I; Movie EV1), corroborating the upregulation of *spz* observed in our scRNA-seq dataset (Fig. EV5C). Based on these findings, we hypothesize that hemocytes are the primary source of Spz in *Ras^V12//M6^−/−* tumors.

Previous studies have reported that ROS-dependent apoptosis can recruit hemocytes (Perez et al, 2017). While *M6*-mutant clones alone induced apoptosis (Fig. EV1A,B), no increase in hemocyte recruitment was observed under this condition (Fig. EV5F,G). Interestingly, Dcp-1 staining of *Ras^V12//M6^−/−* tumors revealed apoptosis occurring within GFP^− *M6* mutant clones, with significantly more apoptotic cells detected in GFP^+ *Ras^V12* tumor cells (Fig. EV5H). This observation aligns with the upregulation of *rpr-lacZ* in *Ras^V12//M6^−/−* tumors (Fig. EV3F). Given that hemocyte recruitment persists even when apoptosis is blocked during cell competition (Zhu et al, 2025), it would be intriguing to explore the role of apoptosis in hemocyte recruitment across different tumorigenic contexts.

Next, to investigate whether hemocyte-derived Spz promotes *Ras^V12//M6^−/−* tumor transformation in vivo, we combined the QMARCM and Gal4/UAS system. We generated *QRas^V12//M6^−/−* tumors on *Drosophila* eye-antennal discs and knocked down *spz* in hemocytes via *Hemese-Gal4* (*He-Gal4*), a pan-hemocyte-specific driver. This resulted in significant suppression of *QRas^V12//M6^−/−* tumor growth (Fig. 6A–C). Conversely, ectopic expression of an activated form of *spz* (*spz^ACT*) under the control of *Cg-Gal4* (which labels both hemocytes and fat bodies) remotely accelerated *QRas^V12* tumor overgrowth (Fig. EV6A–C), increased the attachment of Spz-expressing hemocytes (Fig. EV6D), and upregulated the expression of dl, *yki* targets Diap1 and CycE (Fig. EV6E), as well as JNK target

genes Mmp1 and P-JNK (Fig. EV6F). Expression of *spz^ACT* via *He-Gal4* similarly recapitulated *QRas^V12* tumor overgrowth, hemocyte recruitment, and the upregulation of dl expression (Figs. 6D,E and EV6G–J), suggesting that hemocyte secreted Spz drives the progression of *Ras^V12* tumors.

To further substantiate these findings, we performed allograft tumor experiments by transplanting either *Ras^V12* or *Ras^V12//M6^−/−* eye-antennal disc tumors into the abdomens of female hosts and measuring tumor size 8 days post-transplantation (Fig. 6F). Consistent with prior observations, the transplanted *Ras^V12//M6^−/−* tumor grafts exhibited significantly larger sizes compared to *Ras^V12* grafts (Fig. 6G,H). Hemocyte-specific knockdown of *spz* or ablation of hemocyte by overexpressing pro-apoptotic genes such as *rpr* or *hid*, both driven by *Hml-gal4*, significantly inhibited tumor growth in the *Ras^V12//M6^−/−* allograft model (Fig. 6G, H). Collectively, these data suggest that hemocyte-derived Spz activates the Toll pathway, thereby transforming *Ras^V12* clones into malignant tumors (Fig. 6I).

## Pvf/Pvr signaling mediates crosstalk between *Ras^V12* clones and hemocytes in *Ras^V12//M6^−/−* heterogeneous tumors

To explore the mechanisms underlying the upregulation of *spz* expression by hemocytes, we performed integrative analyses of hemocytes attached to both *WT* and *Ras^V12//M6^−/−* tumor-bearing discs. This analysis revealed five distinct hemocyte clusters unique to *Ras^V12//M6^−/−* tumors (cluster 0, 2, 5, 6, 7) (Fig. 7A). The DEGs between the WT hemocytes and *Ras^V12//M6^−/−* hemocytes were extracted for further functional annotation, and a total of 29 overlapping enriched regulons were identified from KEGG database, including two positively correlated pathways, the immune pathway and MAPK pathway (Figs. 7B–D and EV7A). *Drosophila* encompasses three main MAPK pathways: the ERK (extracellular-signal-regulated kinase), p38, and JNK pathways. Interestingly, previous studies have reported that JNK pathway activation can upregulate *spz* expression in *Drosophila* (Wu et al, 2015). In line with this, we observed a positive correlation between the expression of *spz* and the activation of JNK specifically in *Ras^V12//M6^−/−* tumor-attached hemocytes (Fig. EV7A), suggesting that JNK activation in hemocytes may contribute to the upregulation of *spz*.

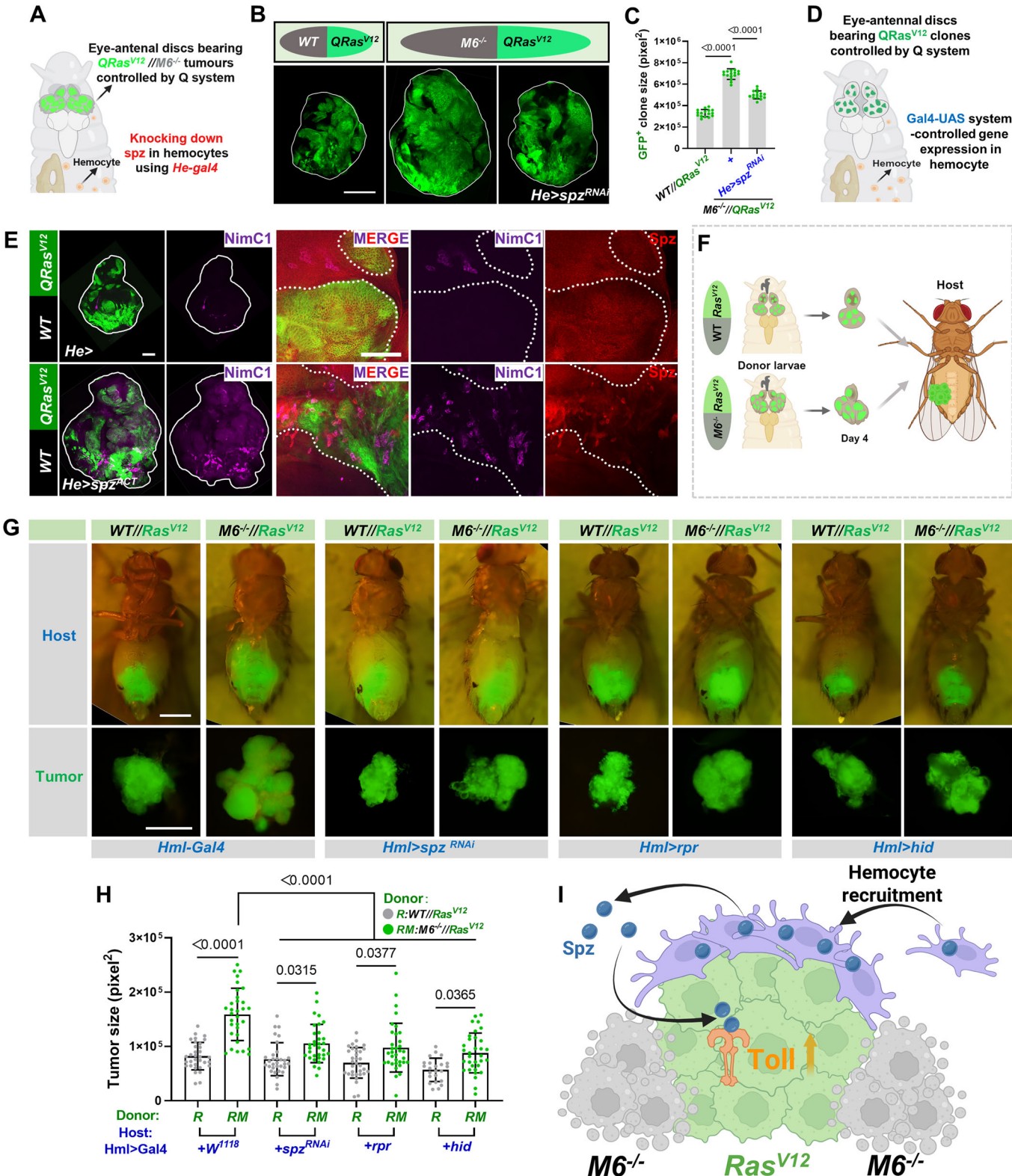

**Figure 6. Hemocyte-derived Spz promotes $Ras^{V12}//M6^{-/-}$ tumor overgrowth.**

(A) Cartoon illustrating the dual-expression system designed to simultaneously knock down *spz* in hemocytes while inducing GFP-labeled $QRas^{V12}//M6^{-/-}$ tumors in the eye-antennal disc under the control of the Q system. (B) Eye-antennal discs bearing GFP-labeled $QRas^{V12}$ clones (AEL-6), and $QRas^{V12}//M6^{-/-}$ tumors without or with $spz^{RNAi}$ expression driven by the *He* promoter (AEL-7). White lines demarcate the boundaries of the eye-antennal discs. (C) Quantification of GFP$^+$ clone sizes across the indicated genotypes (from left to right, $n = 16, 16, 15$). Statistical analysis by ordinary one-way ANOVA test; mean ± SD. (D) Cartoon illustrating the dual-expression system to overexpress activated *spz* ($spz^{ACT}$) in hemocytes while simultaneously inducing GFP-labeled $QRas^{V12}$ clones in the eye-antennal disc, controlled by the Q system. (E) Eye-antennal discs bearing GFP-labeled $QRas^{V12}$ clones without (AEL-6, upper panels) or with (AEL-7, lower panels) $spz^{ACT}$ expression by the *He* promoter were stained with anti-NimC1 and anti-Spz antibodies. White lines demarcate the boundaries of the eye-antennal discs, while white dashed lines mark the boundaries of the GFP-positive clones. (F) Cartoon illustration of the allograft tumor model: Four days post hatching, eye-antennal discs bearing either $Ras^{V12}$ tumor or $Ras^{V12}//M6^{-/-}$ tumor were transplanted into the abdomens of adult female hosts with indicated genotypes. Transplanted tumor sizes were measured eight days post-transplantation. (G) $Ras^{V12}//WT$ and $Ras^{V12}//M6^{-/-}$ eye-antennal discs were transplanted into female adults with the following genotypes: *Hml-Gal4, Hml > spz$^{RNAi}$, Hml > rpr* and *Hml > hid*, and transplants were dissected 8 days later for quantification. (H) Quantification of GFP-positive transplanted tumor sizes across indicated genotypes (from left to right, $n = 33, 32, 32, 33, 33, 34, 23, 33$). Statistical analysis by ordinary one-way ANOVA test; mean ± SD. (I) Cartoon illustration depicting tumor-associated hemocytes secreting Spz to activate the Toll pathway, leading to the overgrowth of $Ras^{V12}$ clones within $Ras^{V12}//M6^{-/-}$ tumors. Exact *P* values are shown in the corresponding panels. Scale bars: 100 μm (E), 200 μm (B), 500 μm (G). Source data are available online for this figure.

To further explore tumor-hemocyte interactions, we integrated both GFP-positive and GFP-negative epithelium clusters from $Ras^{V12}//M6^{-/-}$ tumors with hemocytes profiles (Figs. 7E and EV7B). Using FlyPhoneDB, a web-based tool for predicting cell-cell communications in *Drosophila* (Liu et al, 2022d), we analyzed ligand-receptor expression and identified three pathways with significant interaction scores ($P < 0.05$) between GFP-positive tumor cells and hemocytes: the epidermal growth factor receptor (*EGFR*), fibroblast growth factor receptor (*FGFR*), and platelet-derived growth factor receptor (*PVR*) pathways. Among these, the PVR pathway exhibited exceptionally active tumor-hemocyte crosstalk (Figs. 7F and EV7C,D), whereas no significant connectivity was observed between GFP-negative clusters and hemocytes. GFP-positive $Ras^{V12}//M6^{-/-}$ tumor subclusters displayed elevated expression of the PVR ligands (*Pvf1*, *Pvf2*, and *Pvf3*), while hemocytes exhibited high transcriptional levels of the receptor *Pvr* (Fig. EV7E). Consistently, bulk RNA-seq analysis further confirmed significant upregulation of PVR ligands in $Ras^{V12}//M6^{-/-}$ tumor samples (Fig. 7G). Notably, we observed upregulated expression of *Ets21C*, a transcription factor known to directly induce *Pvf1* expression (Mundorf et al, 2019), in both scRNA-seq and bulk RNA-seq datasets of $Ras^{V12}//M6^{-/-}$ tumors (Figs. 7G and EV7E).

Based on these findings, we propose that elevated *Ets21C* levels in $Ras^{V12}//M6^{-/-}$ tumor drive Pvf1 secretion, thereby promoting hemocyte recruitment. Supporting this hypothesis, knockdown of either *Ets21C* or *Pvf1* in $Ras^{V12}$ clones within $Ras^{V12}//M6^{-/-}$ tumors significantly reduced tumor growth and reduced hemocyte recruitment (Figs. 7H and EV7F–K). In contrast, knockdown of *Pvf2* in the $Ras^{V12}$ clones of the $Ras^{V12}//M6^{-/-}$ tumors had no effect on tumor growth or hemocyte recruitment (Figs. 7H and EV7F–L). Next, to elucidate the in vivo role of Pvf1-Pvr signaling in regulating hemocyte phenotype during $Ras^{V12}//M6^{-/-}$ tumorigenesis, we used the dual system mentioned above to inhibit Pvr activity in hemocytes. Notably, Pvr inhibition by overexpressing a dominant negative form ($Pvr^{DN}$) via *He-Gal4* significantly attenuated tumor growth and reduced hemocyte recruitment to $QRas^{V12}//M6^{-/-}$ tumors (Figs. 7I and EV8A,B). Furthermore, using the allograft tumor model, we found that hemocyte-specific inhibition of *Pvr* ($Pvr^{DN}$) significantly suppressed the overgrowth of transplanted $Ras^{V12}//M6^{-/-}$ tumors (Fig. EV8C,D). These findings indicate that Pvf/Pvr signaling mediates crosstalk between $Ras^{V12}$ clones and hemocytes in $Ras^{V12}//M6^{-/-}$ heterogeneous tumors

(Fig. 7J). Notably, inhibiting *Pvr* activity in hemocytes did not completely block the recruitment of hemocytes to the tumor (Figs. 7I and EV8B), suggesting that additional signals may also contribute to hemocyte recruitment. Finally, qPCR analysis revealed that JNK pathway inhibition in $Ras^{V12}//M6^{-/-}$ tumors significantly reduced the expression of *Ets21C* and the JNK target gene *Mmp1* (Fig. EV8E), and knockdown of *Ets21C* abolished the transcriptional upregulation of *Pvf1* induced by $Ras^{V12}//M6^{-/-}$ tumors (Fig. EV8F). Collectively, these results demonstrate that JNK pathway activation in $Ras^{V12}//M6^{-/-}$ tumors induces *Ets21C* expression, which enhances *Pvf1* production, facilitating hemocyte recruitment and ultimately driving $Ras^{V12}//M6^{-/-}$ tumor proliferation (Fig. EV8G).

## Discussion

Tumor heterogeneity is well-established feature across diverse cancer types, with profound implications for tumorigenesis and therapeutic resistance (Gavish et al, 2023; Gonzalez-Silva et al, 2020; Vitale et al, 2021). While intercellular communication between tumors and their microenvironment is recognized as a critical driver of malignancy, the in vivo mechanisms governing these interactions during tumor progression remain poorly understood, largely due to a lack of tractable experimental models. To address this gap, we developed a *Drosophila* tumor heterogeneity model by selectively disrupting the tricellular junction protein M6 in cells adjacent to benign $Ras^{V12}$ tumors. Our findings revealed that loss of *M6* promotes tumor malignancy in neighboring $Ras^{V12}$ tumors by inducing secretion of Pvf1. Pvf1 acts as a chemotactic factor, recruiting hemocytes via the Pvr receptor. These hemocytes, in turn, activate a paracrine Spz-Toll signaling axis within $Ras^{V12}$ clones, establishing a complex interplay between the tumor and its microenvironment. Eventually, the activation of the Toll pathway synergizes with oncogenic *Ras* to induce tumor progression through a JNK-Hippo signaling cascade (Fig. EV8G). This intricate signaling network highlights the cooperative interactions between clones bearing distinct mutations and the immune microenvironment, providing mechanistic insights into how tumor heterogeneity and intercellular communication contribute to malignancy. Our study underscores the importance of understanding tumor-microenvironment dynamics and presents the *Drosophila* model as a powerful platform for dissecting these processes.

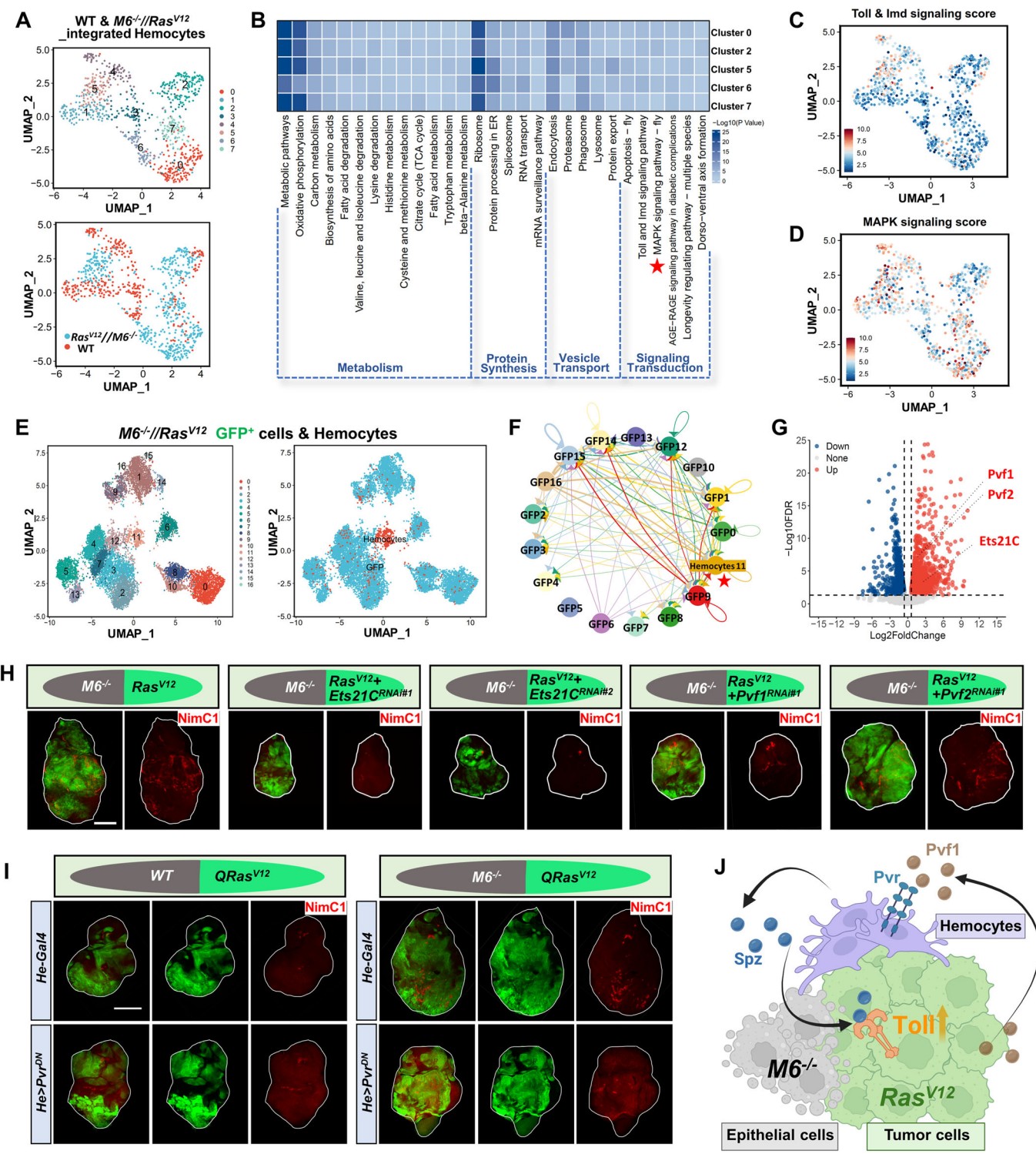

Interestingly, the cell polarity protein Scrib, which localizes to bicellular septate junctions (Byri et al, 2015), promotes $Ras^{V12}$ clone malignancy when mutated in neighboring cells (Wu et al, 2010), further highlighting the critical role of junctional integrity in tumor initiation and progression. Both $Ras^{V12}//scrib^{-/-}$ and $Ras^{V12}//M6^{-/-}$ models illustrate cooperative oncogenesis in *Drosophila*, where $Ras^{V12}$ alone creates benign tumors but requires additional

mutations in surrounding cells to drive malignancy. Both models exploit the genetic tractability of *Drosophila* to uncover fundamental principles of intercellular communication in tumor progression, demonstrating how distinct genetic alterations in neighboring cells can synergize to promote malignant transformation. Notably, the $Ras^{V12}//scrib^{-/-}$ model primarily underscores the activation of JNK pathway, whereas our $Ras^{V12}//M6^{-/-}$ model

◄

**Figure 7. Pvf/Pvr signaling guides the crosstalk between *Ras^V12* clones and hemocytes, regulating *Ras^V12*//*M6^−/−* tumorigenesis.**

(A) UMAP plots showing integrated WT and *Ras^V12*//*M6^−/−* tumor-attached hemocytes, revealing origin-specific clusters. (B) Heatmap showing the enriched 29 overlapping KEGG terms in the 5 clusters (0, 2, 5, 6, 7) of *Ras^V12*//*M6^−/−* tumor-attached hemocytes. (C, D) UMAP plots showing Toll & lmd signaling score (C), and MAPK signaling score (D) of integrated WT and *Ras^V12*//*M6^−/−* hemocytes. (E) UMAP plots of integrated *Ras^V12*//*M6^−/−* tumor-attached hemocytes and *Ras^V12*//*M6^−/−* GFP^+ tumor cells. A total of 17 clusters were identified (left, resolution = 1.2). (F) Circle plot showing a significant enrichment in the PVR signaling pathway between multiple *Ras^V12*//*M6^−/−* GFP^+ tumor cell clusters and *Ras^V12*//*M6^−/−* tumor-associated hemocytes (cluster 11, marked with a red asterisk), revealing a Pvf/Pvr signaling-based cell-cell communication between *Ras^V12*//*M6^−/−* GFP^+ tumor cells and hemocytes. (G) Volcano plot of bulk RNA-seq data by comparing *Ras^V12*//*M6^−/−* malignant tumors (two biological replicates) with *Ras^V12*//*WT* benign tumors (three biological replicates), Pvf1, Pvf2, and Ets21C are shown. DEGs are identified with the criteria of FDR > 0.05 and |Fold-Change| >1.5. (H) Eye-antennal discs bearing clones of *Ras^V12*//*M6^−/−*, *Ras^V12*+*Ets21C^RNAi#1*//*M6^−/−*, *Ras^V12*+*Ets21C^RNAi#2*//*M6^−/−*, *Ras^V12*+*Pvf1^RNAi#1*// *M6^−/−*, and *Ras^V12*+*Pvf2^RNAi#1*//*M6^−/−* were stained with anti-NimC1 antibody (AEL-7, late stage). White lines outline the borders of the eye-antennal discs. (I) Eye-antennal discs bearing GFP-labeled *QRas^V12* clones (AEL-6) and *QRas^V12*//*M6^−/−* tumors (AEL-7), in the absence or presence of *Pvr^DN* expression driven by the *He* promoter. White lines outline the borders of the eye-antennal discs. (J) Proposed working model for *Ras^V12*//*M6^−/−* tumor progression: *Ras^V12* tumors secrete Pvf1, which activates Pvr on hemocytes and promotes their recruitment. The recruited hemocytes, in turn, secrete Spz and activate the Toll pathway in *Ras^V12* tumor clones, thereby driving proliferation and contributing to the progression of *Ras^V12*//*M6^−/−* heterogeneous tumors. Scale bars: 200 μm (H, I). Source data are available online for this figure.

reveals a more complex signaling cascade involving multiple pathways and cell types. Further investigation into the mechanistic similarities and differences between these two different heterogeneous tumor models would be of great interest.

## Dynamic regulation of hemocyte-tissue adhesion

Previous studies and recent research have demonstrated that ROS induced by apoptotic caspase promotes hemocytes recruitment in *Drosophila* eye epithelium, facilitating malignant tumor progression (Hirooka et al, 2025; Perez et al, 2017). Similarly, Toll activation in "undead cells" recruits hemocytes through caspase-dependent ROS signaling, driving JNK-mediated proliferation (Shields et al, 2022). In this study, we reveal that *M6* mutant clones alone undergoing apoptosis fail to recruit hemocytes. These findings align with recent evidence showing that apoptosis induction is dispensable for hemocyte recruitment during classical cell competition (Zhu et al, 2025). This suggests that hemocyte recruitment to apoptotic sites in *Drosophila* is context-dependent and may require simultaneous ROS generation, or cytokine secretion such as Pvf1. Crucially, the functional outcome of hemocyte recruitment—whether promoting cell death or proliferation—appears to depend on the specific cellular context. For instance, when only the polarity gene *scrib* is mutated, hemocyte-secreted Eiger facilitates the clearance of *scrib*-mutant cells (Parisi et al, 2014). In contrast, in *Ras/scrib^−/−* tumors, activation of the *Ras* oncogene subverts this hemocyte-mediated clearance process, hijacking it to promote tumor proliferation (Cordero et al, 2010; Hirooka et al, 2025; Perez et al, 2017). Furthermore, hemocytes can detect cell competition within tissues and actively participate by promoting the competitive interactions between "winner" and "loser" cells (Zhu et al, 2025). In our *Ras^V12*//*M6^−/−* tumor model, hemocytes again demonstrate specificity by homing to tumor cells, guided by Pvf1 secretion, and contributing to tumor progression.

These collective findings indicate that hemocytes possess a sophisticated ability to precisely sense genetic heterogeneity within tissues. Based on this discrimination, they can activate specific downstream signaling pathways that ultimately modulate the outcome of cellular interactions, promoting the clearance or proliferation of specific genotypic clones. Interestingly, both the "undead cell" model and the tumor heterogeneity model described here rely on hemocyte recruitment to activate Toll signaling, which further activates the JNK pathway to drive tumor progression. In the "undead cell" model, Toll/

Toll-9 activation by *hid/rpr*-initiated caspase activity (which is subsequently inhibited), leading to hemocyte infiltration. In contrast, our model uniquely integrates Pvf1/Pvr-mediated adhesion to reprogram the tumor microenvironment.

Collectively, these findings underscore the complexity of hemocyte-mediated immune surveillance in *Drosophila*. However, a limitation of our study is the lack of mechanistic exploration into the interplay between apoptosis and ROS production or Pvf1 secretion in *Ras^V12*//*M6^−/−* tumors. While extensive apoptosis was observed in these tumors, the interplay between ROS-mediated oxidative stress and Pvf1-driven hemocyte recruitment remains uncharacterized. It is also worth noting that bulk RNA-seq data inherently include contributions from infiltrating hemocytes, which play a role in tumor microenvironments by influencing the local transcriptome. These cells are recognized as a potential source of immune-related transcriptional signals. To address this, we performed scRNA-seq analyses, which not only validated our bulk findings (e.g., Pvf1 differential expression) but also revealed previously unappreciated cellular heterogeneity within tumor-associated hemocytes. Hemocytes demonstrated remarkable functional plasticity, integrating spatial gradients, metabolic cues, and genetic alterations to execute context-dependent responses, participating in the competition and cooperation among different clones. Future investigations should focus on elucidating how tumor heterogeneity and dynamic microenvironmental signals regulate hemocyte functional states, particularly their dual roles in both anti-tumor defense and immunosuppressive microenvironment formation.

## Pvf1/Pvr signaling as a multifunctional axis in systemic dysregulation

Pvf1/Pvr signaling has emerged as a critical mediator of tumor-host organ interactions, playing a pivotal role in systemic dysregulation. Recent studies in *Drosophila* have demonstrated that Pvf1, secreted by intestinal tumors, can activate the Pvr receptor in distant organs, including the Malpighian tubules and AKH-producing cells (APCs). This activation leads to pathological outcomes such as renal dysfunction (Xu et al, 2024) and systemic host wasting (Ding et al, 2025). Despite these findings, the mechanisms by which Pvf1 selectively triggers distinct downstream Pvr signaling pathways, such as JNK and ERK, in different target organs remain incompletely understood. For instance, Pvr activation in the

Malpighian tubules primarily regulates uric acid metabolism through the JNK/Jra signaling axis (Xu et al, 2024). In contrast, Pvr activation in APCs appears to drive AKH release via mechanisms involving extracellular matrix (ECM) remodeling mediated by Mmp2 and enhanced innervation by excitatory cholinergic neurons (Ding et al, 2025). These distinct signaling outcomes emphasize the complexity and organ-specificity of Pvf1/Pvr interactions. Adding to this complexity, although we cannot exclude the involvement of additional factors contributing to hemocyte recruitment, we discovered that Pvf1 secreted from $Ras^{V12}//M6^{-/-}$ tumors facilitate the recruitment and the activation of hemocytes. Activated hemocytes, in turn, upregulate the Toll ligand Spz via a mechanism that remains unidentified. Further investigation is needed to clarify the spatiotemporal dynamics of Pvf1-Pvr complexes and their interplay with the tumor microenvironment, which might provide deeper insights into the systemic effects of tumor-derived signals and their contributions to host dysfunction.

## Toll receptor diversity: a double-edged sword in tumor immunity

Our data reveal spatially resolved crosstalk between epithelial clones with distinct driver mutations ($Ras^{V12}$ and $M6$) and between tumors and hemocytes, highlighting complex signaling networks that shape tumor progression. Our study highlights the tumor-promoting role of Toll pathway activation in the presence of oncogenic $Ras$ mutation. Beyond the canonical Toll receptor, the *Drosophila* genome encodes eight additional Toll family receptors (TLRs), whose roles in tumorigenesis are context-dependent, exerting both pro- and anti-tumorigenic effects across flies and human (Ji and Hoffmann, 2024; Pradere et al, 2014).

For instance, Toll signaling activation has been shown to impede the elimination of polarity-deficient cells, thereby fostering their tumorigenic growth (Katsukawa et al, 2018). Similarly, Toll-7 overexpression drives tumor overgrowth, while Toll-9 induction promotes undead apoptosis-induced proliferation (Ding et al, 2022; Shields et al, 2022). Additionally, in tumor-associated hemocytes, we observed elevated expression of the ligand *spz* and the receptor *Toll-4*. In $Ras^{V12}//M6$ tumors, we detected upregulation of receptors *18w* (*Toll-2*), *Toll-7*, and *Tollo*, along with the ligand *spz3* in GFP$^+$ $Ras^{V12}$ tumor cells. Intriguingly, prior studies demonstrate that Spz ligands can bind multiple Toll receptors, including Toll-7, a receptor previously implicated in tumor promotion (Chowdhury et al, 2019; Ding et al, 2022). Our findings align with these reports and highlight the need to further dissect the functional roles of these upregulated ligands/receptors in tumorigenesis.

Conversely, Toll activation in the fat body upregulates AMP expression, leading to the remote triggering of tumor cell death (Parisi et al, 2014; Parvy et al, 2019). Moreover, we have previously demonstrated that Toll-6 activation facilitates the elimination of precancerous *scrib* clones by inducing mechanical tension-mediated Hippo activation (Kong et al, 2022). Collectively, these results illustrate the multifaceted roles of Toll-like receptors in tumor immunity: they act as both tumor promoters and suppressors, with their functional outcomes dictated by specific receptor identity, genetic contest, and tissue microenvironment.

## *Drosophila* as a paradigm for conserved tumor-immune dynamics

The fruit fly *Drosophila* has emerged as a powerful model organism for investigating fundamental questions in human cancer biology. This is largely attributed to its suite of sophisticated genetic tools, the high level of genetic conservation, the short life cycle, and its potential for in vivo testing of anti-cancer drugs (Bangi et al, 2019; Bilder et al, 2021; Dar et al, 2012; Hsi et al, 2023; Liu et al, 2022c). In this study, we underscore the critical value of *Drosophila* in dissecting tumor heterogeneity and unraveling the intricate intercellular communication networks within the tumor microenvironment. The success of *Drosophila* models in elucidating conserved molecular mechanisms underlying tumor progression highlights their potential to offer insights into human tumor heterogeneity. Moving forward, translating these insights gained from *Drosophila* studies may enable the development of novel therapeutic strategies, either by disrupting tumor-hemocyte (immune cell) interactions or by leveraging innate immune pathways. Such advancements will further strengthen the connection between *Drosophila* models and human cancer biology, bridging the gap between basic research and clinical applications.

# Methods

**Reagents and tools table**

| Reagent/resource | Reference or source | Identifier or catalog number |
| --- | --- | --- |
| **Experimental models** | | |
| *Drosophila* strains | This study | Table EV1 |
| **Antibodies** | | |
| Mouse monoclonal anti-Mmp1 | DSHB | Cat# 3A6B4, RRID: AB_579780 Cat# 3B8D12, RRID: AB_579781 Cat# 5H7B11, RRID: AB_579779 |
| Dorsal mouse monoclonal | DSHB | Cat# anti-dorsal 7A4; RRID: AB_528204 |
| Relish mouse monoclonal | DSHB | Cat# anti-Relish-C 21F3; RRID: AB_1553772 |
| Cyclin E rat monoclonal | Gift from Helena Richardson (La Trobe University) | Brumby et al, 2002 |
| Mouse monoclonal anti-β-gal | DSHB | Cat# 40-1a, RRID: AB_528100 |
| Mouse monoclonal anti-wg | DSHB | Cat# 4D4, RRID: AB_528512 |
| Rabbit polyclonal anti-cleaved *Drosophila* Dcp-1 | CST | Cat# 9578, RRID: AB_2721060 |
| rabbit polyclonal anti-phospho-histone H3 | CST | Cat# 9701, RRID: AB_331535 |
| Rabbit polyclonal anti-active JNK | Promega | Cat# v2973; RRID: AB_3391716 |

| Reagent/resource | Reference or source | Identifier or catalog number |
|---|---|---|
| Diap1 mouse monoclonal | Gift from Bruce Hay (California Institute of Technology) | N/A |
| Spz rabbit polyclonal | This paper | N/A |
| NimC1 mouse monoclonal | Gift from Istvan Andó (Biological Research Center of the Hungarian Academy of Sciences) | Kurucz, Markus et al, 2007 |
| Goat anti-Rabbit IgG (H + L) Alexa Fluor™ Plus 555 | Invitrogen | Cat# A32732, RRID: AB_2633281 |
| Goat anti-Mouse IgG (H + L) Alexa Fluor™ Plus 555 | Invitrogen | Cat# A32727; RRID: AB_2633276 |
| Goat anti-Rat IgG (H + L) Alexa Fluor™ Plus 555 | Invitrogen | Cat# A21434, RRID: AB_141733 |
| Goat anti-Rabbit IgG (H + L) Alexa Fluor™ Plus 647 | Invitrogen | Cat# A32733, RRID: AB_2633282 |
| Goat anti-Mouse IgG (H + L) Alexa Fluor™ Plus 647 | Invitrogen | Cat#A21237; RRID: AB_2535806 |
| **Oligonucleotides and other sequence-based reagents** | | |
| Oligonucleotides | This study | Table EV1 |
| **Chemicals, enzymes and other reagents** | | |
| Trizol | Thermo Fisher Scientific | Cat# 15596026 |
| Alexa Fluor™ 546 Phalloidin | Invitrogen | Cat# A12381 |
| G418 Sulfate | Macklin | Cat# 10854254 |
| Ampicillin | Macklin | Cat# 10868057 |
| Kanamycin | Macklin | Cat# 13921196 |
| Chloramphenicol | Macklin | Cat# 14454961 |
| pen/strep | Thermo Fisher Scientific | Cat# 15140122 |
| MRS Agar | Solarbio | Cat# M8330 |
| FBS | Gibco | Cat# C11875500CP |
| Highscript III reverse transcriptase | Vazyme | Cat# R212-02 |
| ChamQ Universal SYBR qPCR Master Mix | Vazyme | Cat# Q711-02 |
| DPBS basic (1×) | Gibco | Cat# 8123518 |
| Buffer EB Elution buffer | QIAGEN | Cat# 19086 |
| AMPure XP | Beckman | Cat# A63881 |
| **Software** | | |
| GraphPad Prism 9.0 | http://www.graphpad.com | RRID: SCR_002798 |
| ImageJ | https://imagej.net | RRID: SCR_003070 |
| Adobe Photoshop | https://www.adobe.com/products/photoshop.html | RRID: SCR_014199 |

## Methods and protocols

### Fly husbandry and genetics

*Drosophila* stocks and crosses are typically maintained on a standard cornmeal-yeast food medium at 25 °C, unless otherwise specified. The composition of the standard food medium is as follows: 50 g cornmeal, 9 g agar, 24.5 g dry yeast, 7.25 g white sugar, 30 g brown sugar, 4.4 ml propionic acid, 12.5 ml ethanol, and 1.25 g nipagin per liter of medium. For a complete list of *Drosophila* lines used, please refer to Table EV1. Additionally, detailed genotypes for each figure panel can be found in Table EV2.

### Production of anti-Spz antibody

Polyclonal rabbit antibodies targeting *Drosophila* Spz were generated through ABclonal (Wuhan, China). To facilitate prokaryotic expression of the immunogen, the amino acids 187-307 of Spz (NP_524526.1) were integrated into pET-28a-SUMO. Two experiment-grade Japanese white rabbits were immunized, and the resulting antibodies were purified via affinity purification of the pET-28a-SUMO-Spz protein using anti-serum.

### Immunostaining and imaging

Third-instar larvae imaginal discs were dissected in cold PBS and fixed with 4% paraformaldehyde for 15 min at room temperature (RT) with shaking. Subsequently, the discs were washed three times with PBS containing 0.1% Triton X-100 solution (PBST). To block nonspecific binding, the samples were incubated in PBS with 5% normal goat serum for 1 h at RT. Following this, the samples were incubated with primary antibodies (1:200) overnight at 4 °C. After three washes with PBST, the samples were incubated with secondary antibodies (1:400) for 1 h at RT, protected from light. Following PBST washes (3×), the samples were mounted with a DAPI-containing medium and imaged using an Olympus FV3000-BX63 confocal microscope (inverted) equipped with ×20 objectives, Nikon A1R confocal microscope (inverted) using 60× oil objectives or a Zeiss Axio Observer with ApoTome2. The acquired images were processed using Image J and Photoshop CS5 (Adobe) for image merging and resizing. For the measurement of immunofluorescence staining intensity, more than 10 eye-antennal discs were randomly chosen for quantification. The Fiji/ImageJ software was used for automated quantification of fluorescent intensity.

### Clone size measurement

The fluorescent intensity in the images was determined using the ImageJ (Fiji) software, either through manual or automatic calculations. Additionally, the sizes of GFP clones in the eye-antennal disc were determined using ImageJ.

### RNA isolation and RT-qPCR

The third instar larvae of *Drosophila* were washed twice with sterile dPBS and dissected in pre-cooled dPBS. The collected eye-antennal discs were quickly frozen with liquid nitrogen and placed in −80 °C refrigerator for later use. Total RNA was extracted from the eye-antennal discs using TRIzol (Ambion), and cDNA was obtained by reverse transcription according to the manufacturer's instructions.

Quantitative PCR was performed with SYBR Green Real-time PCR Master Mix to determine the relative abundance of mRNA. The enrichment of DNA sequences was quantified using qPCR with the primers listed in Table EV1.

### Antibiotic food formula

The original standard food was boiled twice, allowed to cool down, and then the following concentrations of drugs were added: 50 μg/ml of ampicillin, 25 μg/ml of kanamycin, 25 μg/ml of chloramphenicol, and 1.7 ml/100 ml of penicillin/streptomycin.

### Culture of germ-free Drosophila

Germ-free *Drosophila* were produced by collecting freshly laid eggs, incubating them in 10% bleach for 5 min, washing them in sterile water three times for 5 min each time, and transferring the washed eggs to food containing antibiotics. The eggs were then placed in a sterile incubator for culture, and the third instar larvae were dissected at a later stage.

### Drosophila bacteria culture

*Drosophila* fed on normal food, and flies fed on antibiotic-containing food were collected separately at the third stage of larvae. The collected larvae were washed once with ddH$_2$O, then washed with 75% alcohol, and finally washed again with ddH$_2$O. Ten cleaned larvae were randomly selected and were placed in a 1.5-ml centrifuge tube for grinding with a grinding rod. The homogenate after grinding was diluted 1000 times and evenly smeared on an MRS (de Man, Rogosa and Sharpe) agar plate. The plate was then cultured in an incubator at 25 °C. After 48 h, observations were made, and photographs were taken.

### Drosophila allograft model

*Drosophila* larvae bearing $Ras^{V12}//WT$ and $Ras^{V12}//M6^{-/-}$ tumors on the eye discs were cultured in incubators at 25 °C. On the fourth day after hatching, eye discs of roughly the same size were selected for injection. The host flies had the following genotypes: *Hml > Gal4, Hml > spz^{RNAi}, Hml > rpr, Hml > hid, Hml > Pvr^{DN}*. These host flies were cultured in a 25 °C incubator, and female flies were chosen for the tumor transplantation experiment on the third day after eclosion. The $Ras^{V12}//WT$ and $Ras^{V12}//M6^{-/-}$ tumor-bearing discs were aspirated into capillaries and injected into the lower abdomen of the host flies using a pipette gun. The hosts that received the transplants were then cultured in a 29 °C incubator, and their food was changed once a day to ensure the cleanliness of the injected flies. On the eighth day after transplantation, the hosts were photographed using fluorescent stereomicroscope (Olympus, SZX16), and the transplanted tumors in the abdomen were dissected and their size was measured.

### Bulk RNA-seq

Bulk-seq analysis is primarily focused on characterizing the molecular changes occurring in late-stage $Ras^{V12}//M6^{-/-}$ tumors. To this end, we collected samples on the 10th day post-egg-laying. At this developmental stage, most flies had already entered pupation, whereas in non-pupated $Ras^{V12}//M6^{-/-}$ flies, GFP-positive tumor cells had extensively infiltrated and occupied the majority of the eye-antennal disc tissue.

For downstream molecular profiling, A total of 100 eye-antenna-disc tumors from $Ras^{V12}//WT$ (control group, 3 replicates) and $Ras^{V12}//M6^{-/-}$ (case group, 2 replicates) were harvested for total RNA extraction by using Trizol (Invitrogen, Carlsbad, CA) according to manual instruction. The mRNA library construction was performed as previously described (Liu et al, 2022a). Briefly, the mRNA libraries were amplified with phi29 to make DNA nanoballs (DNBs), which were then loaded into the patterned nanoarray, and single end 50 bases reads were generated on BGIseq500 platform (BGI-Shenzhen, China).

### Bulk RNA-seq data analysis

The transcriptomic data was processed through a standardized pipeline. Data quality control and reads statistics were performed by using FastQC v0.11.8 software (https://github.com/s-andrews/FastQC/releases/tag/v0.11.8). The maintained high-quality reads were mapped to the *Drosophila melanogaster* reference genome (*Drosophila_melanogaster*.BDGP6.32.108) by using Hisat2 (Kim et al, 2019). Gene feature counting was conducted by using HTSeq (Anders et al, 2015) with default settings. Differential gene expression analysis and further data exploration were processed in R (version 4.2.0). Differentially expressed genes (DEGs) were identified by using the "RLE" method of edgeR (Robinson et al, 2010) meeting the criterion of |Fold-Change| >1.5 and false discovery rate (FDR) < 0.05. Over-representation analyses (ORA) were performed by using clusterProfiler package (Yu et al, 2012), DEGs were annotated against terms in Gene Ontology (GO) consortium and Kyoto Encyclopedia of Genes and Genomes (KEGG) database. Relevant results were visualized by using pheatmap (v1.0.12), RColorBrewer (v1.1-3), and ggplot2 (v3.4.4) packages.

### Single-cell RNA-seq

scRNA-seq analysis is designed to characterize early-stage molecular changes in $Ras^{V12}//M6^{-/-}$ tumors and aim to enrich GFP-negative $M6^{-/-}$ cells. To capture these dynamic early events, we collected samples on the 6th day post-egg-laying, a developmental stage when $Ras^{V12}//M6^{-/-}$ tumors still retain a detectable population of $M6^{-/-}$ cells, facilitating the study of their early tumorigenic progression.

For scRNA-seq library preparation, we harvested 150 eye-antennal discs from wild-type (WT) late third-instar larvae (control group, 3 biological replicates) and 50 tumors from $Ras^{V12}//M6^{-/-}$ larvae (case group, 3 biological replicates). Tissues were digested by using sCelLive™ Tissue Dissociation Mix (Singleron). Cell viability was microscopically evaluated by using Trypan Blue staining (Cattenoz et al, 2020, C0040, Solarbio) as well as Acridine Orange/Propidium Iodide (AO/PI) staining (#RE010212, Countstar). Samples with cell viability greater than 90% were used for subsequent single cell library construction strictly following the instructions of GEXSCOPE® Single Cell RNA Library Kits (Dura et al, 2019) (Singleron). Briefly, single-cell suspensions (4.0 ~ 5.0 ×105 cells/mL) were loaded on the microwell chip using the Singleron Matrix® Single Cell Processing System (Singleron), in which single cells were lysed and their RNA were captured by barcoding beads. After collecting barcoding beads from the microwell chip, the reverse transcription of captured mRNA and cDNA amplification were followed. The amplified cDNA was further fragmented and ligated with sequencing adapters, and libraries that passed quality control were then sequenced on Illumina NovaSeq 6000 with 150 bp paired end reads.

### Single-cell RNA-seq data analysis

scRNA-seq data were mapped and quantified by using Celescope pipelines (https://github.com/singleron-RD/CeleScope) v1.10.0 against *Drosophila melanogaster* reference genome (*Drosophila_melanogaster*.BDGP6.32.108) with exogenous gene sequence (GFP and Gal80) inserted. Different scRNA-seq datasets were loaded into R (version 4.2.0) by using the *Read10X* function of Seurat v4.2 (Butler et al, 2018; Stuart et al, 2019). Sample-specific thresholds for unique molecular identifiers (UMIs), gene counts, mitochondrial gene expression, and ribosomal gene expression were set to filter low-quality cells. In wild-type group, cells passed the following standards "$500 < nCount\_RNA < 8000$; $250 < nFeature\_RNA < 1500$; the proportion of mitochondrial gene counts and ribosomal gene counts: <30%" were maintained as high-quality cells. The criterion for $Ras^{V12}//M6^{-/-}$ group cells is "$200 < nCount\_RNA < 12{,}000$; $250 < nFeature\_RNA < 2500$; mitochondrial gene percentage <35%; ribosomal gene percentage < 40%". The doublets were identified by applying DoubletFinder (McGinnis et al, 2019) with 5% putative doublets. As a result, a total of 32,225 cells were obtained for wild-type control, and 43,992 cells for $Ras^{V12}//M6^{-/-}$ tumor group. Normalized counts and scaled gene expression values were obtained using the *NormalizeData* and *ScaleData* functions. For each sample, 2500 top variable genes identified by *FindVariableFeatures* function were selected for dimension reduction of principal component analysis (PCA), and cells were finally projected in uniform manifold approximation and projection (UMAP) space using 40 principal components (PCs), and primary cell clustering was performed using the Louvain algorithm (resolution 0.3). *FindIntegrationAnchors* and *IntegrateData* functions were used to integrate different datasets, where 2000 genes were used for anchor finding, and 50 dimensions were set for the canonical correlation analysis (CCA). Cell-type annotations were performed based on known gene markers and previous studies (Ariss et al, 2018; Bravo González-Blas et al, 2020). The top-expressing genes and DEGs between clusters were identified by *FindAllMarkers* and *FindMarkers* functions with a threshold of expressing in 25% cluster cells. The visualization and functional annotation of genes of interest were dependent on various R packages, including ComplexHeatmap (Gu et al, 2016), clusterProfiler (Yu et al, 2012), scRNAtoolVis (https://github.com/junjunlab/scRNAtoolVis), etc. The clusters and cells of interest were extracted by using *subset* function, and re-integrated and re-clustered with higher resolution (0.7–1.3) under different conditions to reveal the heterogeneity between subclusters. The eye-antennal disc tissue-resident hemocytes were isolated from previously reported scRNA-seq dataset (Bravo Gonzalez-Blas et al, 2020) (GSE141589). Intercellular communication networks across different signaling pathways were constructed by using FlyPhoneDB (Liu et al, 2022b) locally.

### Graphics

Panels in Figs. 1A,B, 4L, 5H, 6A,D,F,I, 7J, EV1G, EV6A, and EV8G and synopsis were created with BioRender.com.

### Quantification and statistical analysis

ImageJ was used to quantify the area of GFP clones, and OLYMPUS cellSens Dimension was used to measure the size of the transplanted tumor. The size of GFP clones was determined by analyzing all the clones in at least 10 randomly selected eye-antennal discs of each genotype. To assess the signal strength of GFP clones and adjacent wild-type cells within the clones, at least ten eye discs were analyzed using ImageJ.

All statistical analyses were performed using GraphPad Prism 9.0 software. The investigators analyzing the data were not blind to the identity of the samples. The specific statistical tests used for each analysis are indicated in the corresponding figures. Statistical significance was determined by performing an unpaired two-tailed Student's *t* test to compare two groups or by conducting ordinary one-way ANOVA with Bonferroni's, Dunnett's, or Tukey's correction for multiple comparisons when comparing three or more groups for parametric data. Non-parametric data were analyzed with unpaired non-parametric Mann–Whitney tests. The data are presented as mean ± standard deviation (s.d.). $P$ value < 0.05 was considered statistically significant.

## Data availability

All sequencing data of this study are deposited in the National Center for Biotechnology Information Sequence Read Archive (Sun et al, 2025) with the accession number BioProject: PRJNA1064441.

The source data of this paper are collected in the following database record: biostudies:S-SCDT-10_1038-S44318-025-00547-5.

## Peer review information

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

## Acknowledgements

We thank Tian Xu, Tatsushi Igaki, Duojia Pan, Lei Xue, Shian Wu, Istvan Andó, Helena Richardson, Bruce Hay, Bloomington *Drosophila* Stock Center, Vienna *Drosophila* Resource Center, Tsinghua Fly Center and Developmental Studies Hybridoma Bank for providing fly stocks and reagents; the Microscopy Core

Facility and the High-Performance Computing Center of Westlake University for the facility support and technical assistance; Wenhan Liu for fly stock maintenance. This project is supported by startup funds from Westlake University, National Natural Science Foundation of China (32170824, 32322027), HRHI program (1011103360222B1) of Westlake Laboratory of Life Sciences and Biomedicine, "Pioneer" and "Leading Goose" R&D Program of Zhejiang (2024SSYS0034).

## Author contributions

**Sihua Zhao**: Investigation; Writing—original draft. **Yifan Guo**: Investigation; Writing—original draft. **Xiaoyu Kuang**: Investigation. **Xiaoqin Li**: Investigation. **Chenxi Wu**: Resources. **Peng Lin**: Investigation. **Qi Xie**: Supervision. **Zongzhao Zhai**: Resources; Investigation. **Du Kong**: Investigation; Writing—original draft; Writing—review and editing. **Xianjue Ma**: Conceptualization; Supervision; Funding acquisition; Writing—original draft; Writing—review and editing.

Source data underlying figure panels in this paper may have individual authorship assigned. Where available, figure panel/source data authorship is listed in the following database record: biostudies:S-SCDT-10_1038-S44318-025-00547-5.

## Disclosure and competing interests statement

The authors declare no competing interests.

# Expanded View Figures

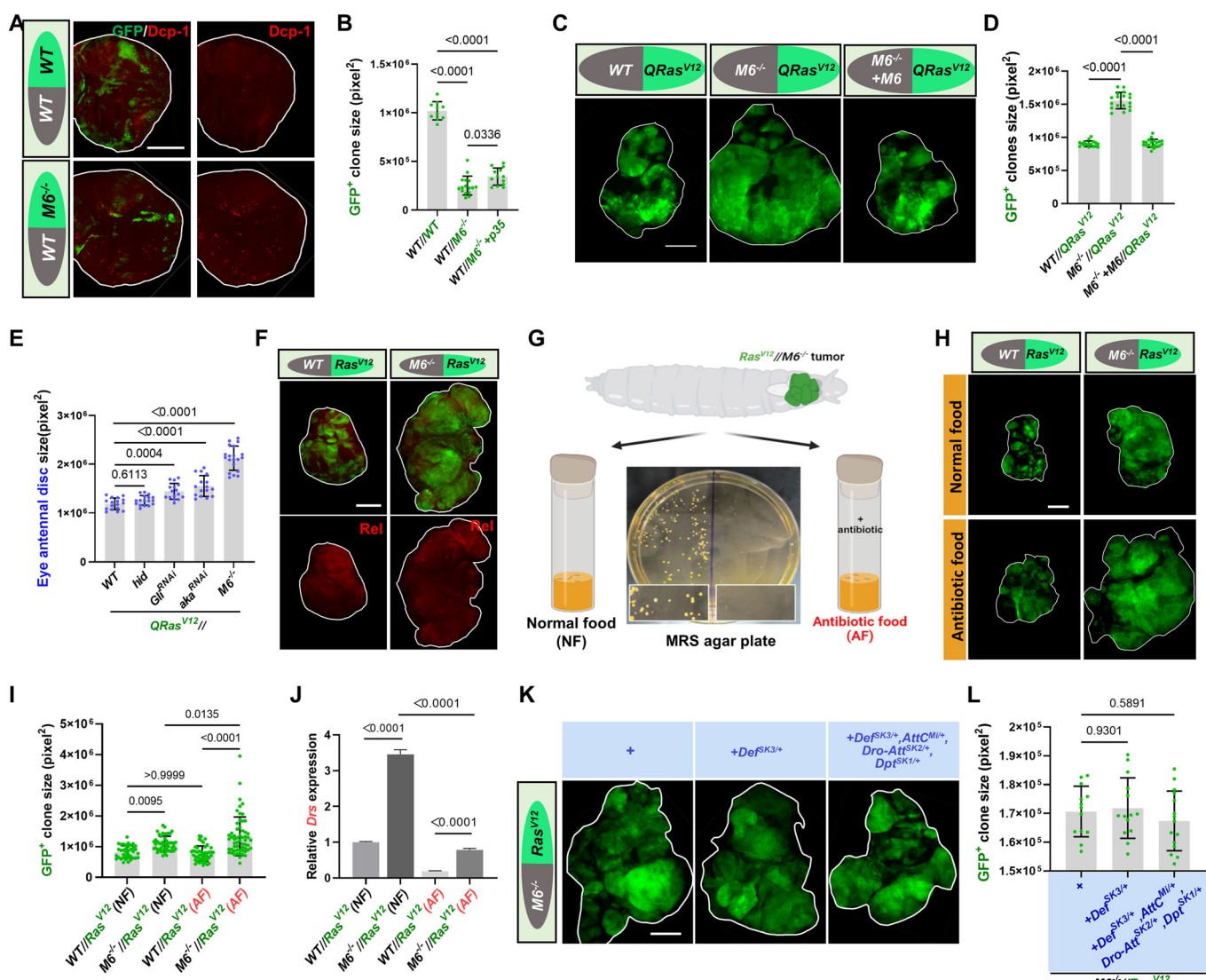

**Figure EV1.** *Ras^V12^//M6^−/−* tumor-induced Toll activation is independent of microbial infection.

(A) Eye discs bearing GFP-labeled clones of WT and *M6^−/−* were stained with anti-Dcp-1 antibody at AEL-6. The white lines outline the borders of the eye discs. (B) Quantification of GFP+ clones' size (from left to right, *n* = 10, 16, 13) with indicated genotypes. Statistical analysis by ordinary one-way ANOVA test; mean ± SD. (C) Eye-antennal discs bearing clones of *WT//QRas^V12^* (AEL-6), and *M6^−/−^//QRas^V12^*, M6 + *M6^−/−^//QRas^V12^* (AEL-7). The white lines outline the borders of the eye-antennal discs. (D) Quantification of GFP+ clones' size (from left to right, *n* = 17, 18, 20) with indicated genotypes. Statistical analysis by ordinary one-way ANOVA test; mean ± SD. (E) Quantification of eye-antennal discs' size (from left to right, *n* = 17, 17, 16, 16, 20) with indicated genotypes. Statistical analysis by ordinary one-way ANOVA test; mean ± SD. (F) Eye-antennal discs bearing clones of *Ras^V12^//WT* (AEL-6) and *Ras^V12^//M6^−/−* (AEL-7) were stained with Relish antibody. The white lines outline the borders of the eye-antennal discs. (G) *Ras^V12^//M6^−/−* tumor-bearing larvae were cultured on normal food (NF) and food supplemented with antibiotics (AF), respectively. Representative bacterial culture images were shown with larvae grown either on normal food or antibiotic food on MRS agar plates. (H) Eye-antennal discs bearing clones of *Ras^V12^//WT* (AEL-6) or *Ras^V12^//M6^−/−* (AEL-7) cultured with or without antibiotic treatment. The white lines outline the borders of the eye-antennal discs. (I) Quantification of GFP+ clones' size with the indicated genotypes (from left to right, *n* = 38, 50, 47, 58). Statistical analysis by ordinary one-way ANOVA test; mean ± SD. (J) qPCR analysis to determine relative *Drs* mRNA levels of eye-antennal disc dissected from *Ras^V12^//WT* (AF and NF) or *Ras^V12^//M6^−/−* (AF and NF) (*n* = 4 independent experiments). Statistical analysis by ordinary one-way ANOVA test; mean ± SD. (K) Eye-antennal discs bearing clones of *Ras^V12^//WT*, *Ras^V12^+Def^SK3/+^//M6^−/−^+Def^SK3/+^*, *Ras^V12^+Def^SK3/+^+AttC^MI/+^+Dro-Att^SK2/+^+Dpt^SK1/+^//M6^−/−^+Def^SK3/+^+AttC^MI/+^+Dro-Att^SK2/+^+Dpt^SK1/+^* (AEL-7). The white lines outline the borders of the eye-antennal discs. (L) Quantification of GFP+ clones' size with the indicated genotypes (from left to right, *n* = 13, 14, 15). Statistical analysis by ordinary one-way ANOVA test; mean ± SD. Exact *P* values are shown in the corresponding panels. Scale bars: 100 μm (A), 200 μm (C, F, H, K).

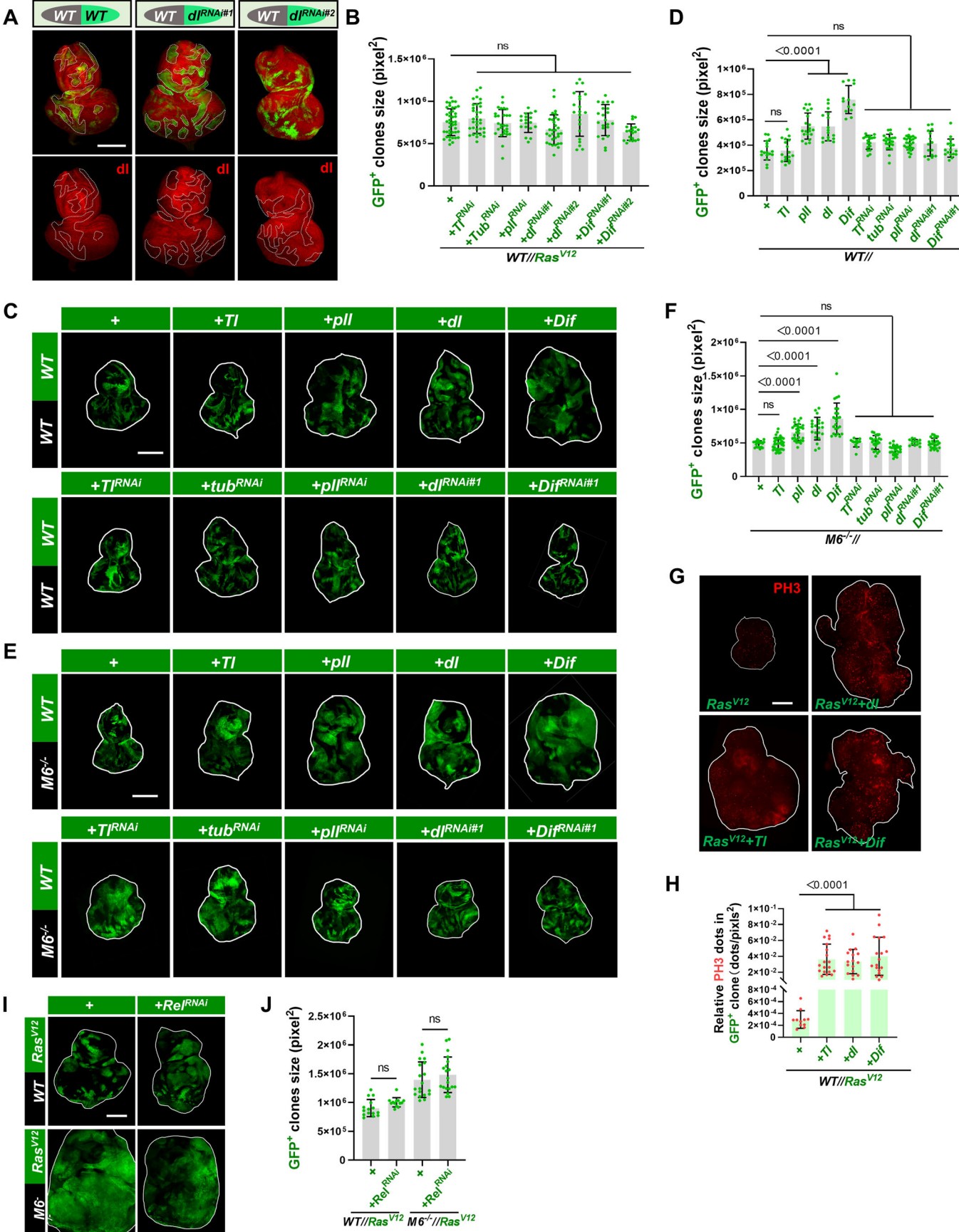

◀  **Figure EV2.  Inhibition of the Imd pathway does not affect *Ras^V12^//M6^−/−^* tumors.**

(**A**) Eye-antennal discs bearing clones of *WT//WT*, *dl^RNAi#1^//WT*, *dl^RNAi#2^//WT* were stained with anti-dl antibody (AEL-6). The white lines outline the borders of the eye-antennal discs. (**B**) Quantification of GFP⁺ clones' size with indicated genotypes (from left to right, $n = 42, 33, 28, 18, 31, 21, 22, 20$). Statistical analysis by ordinary one-way ANOVA test; mean ± SD; ns., not significant. (**C**) Eye-antennal discs bearing *WT//WT*, *Tl//WT*, *pll//WT*, *dl//WT*, *Dif//WT*, *Tl^RNAi^//WT*, *tub^RNAi^//WT*, *pll^RNAi^//WT*, *dl^RNAi#1^//WT*, *Dif^RNAi#1^//WT* (AEL-6). The white lines outline the borders of the eye-antennal discs. (**D**) Quantification of GFP⁺ clones' size for the indicated genotypes (from left to right, $n = 17, 17, 20, 16, 13, 18, 22, 25, 15, 17$). Statistical analysis by ordinary one-way ANOVA test; mean ± SD; ns., not significant. (**E**) Eye-antennal discs bearing *WT//M6^−/−^*, *Tl//M6^−/−^*, *pll//M6^−/−^*, *dl//M6^−/−^*, *Dif//M6^−/−^*, *Tl^RNAi^//M6^−/−^*, *tub^RNAi^//M6^−/−^*, *pll^RNAi^//M6^−/−^*, *dl^RNAi#1^//M6^−/−^*, *Dif^RNAi#1^//M6^−/−^* (AEL-6). The white lines outline the borders of the eye-antennal discs. (**F**) Quantification of GFP⁺ clones' size for the indicated genotypes (from left to right, $n = 18, 34, 29, 21, 24, 13, 24, 25, 13, 25$). Statistical analysis by ordinary one-way ANOVA test; mean ± SD; ns., not significant. (**G**) Eye-antennal discs with *Ras^V12^//WT* clones (AEL-6), and with *Ras^V12^ + Tl//WT*, *Ras^V12^+dl//WT*, *Ras^V12^+ Dif//WT* clones (AEL-8), were stained with anti-PH3 antibody. The white lines outline the borders of the eye-antennal discs. (**H**) Quantification of PH3 dots per GFP⁺ clones' area (from left to right, $n = 11, 19, 16, 18$). Statistical analysis by ordinary one-way ANOVA test; mean ± SD. (**I**) Eye-antennal discs with *Ras^V12^//WT*, *Ras^V12^+Rel^RNAi^//WT* clones (AEL-6), and with *Ras^V12^//M6^−/−^*, *Ras^V12^+Rel^RNAi^// M6^−/−^* clones (AEL-7). The white lines outline the borders of the eye-antennal discs. (**J**) Quantification of GFP⁺ clones' size for the indicated genotypes (from left to right, $n = 15, 13, 19, 20$). Statistical analysis by ordinary one-way ANOVA test; mean ± SD. Exact *P* values are shown in the corresponding panels. Scale bars: 200 µm (**A, C, E, G, I**).

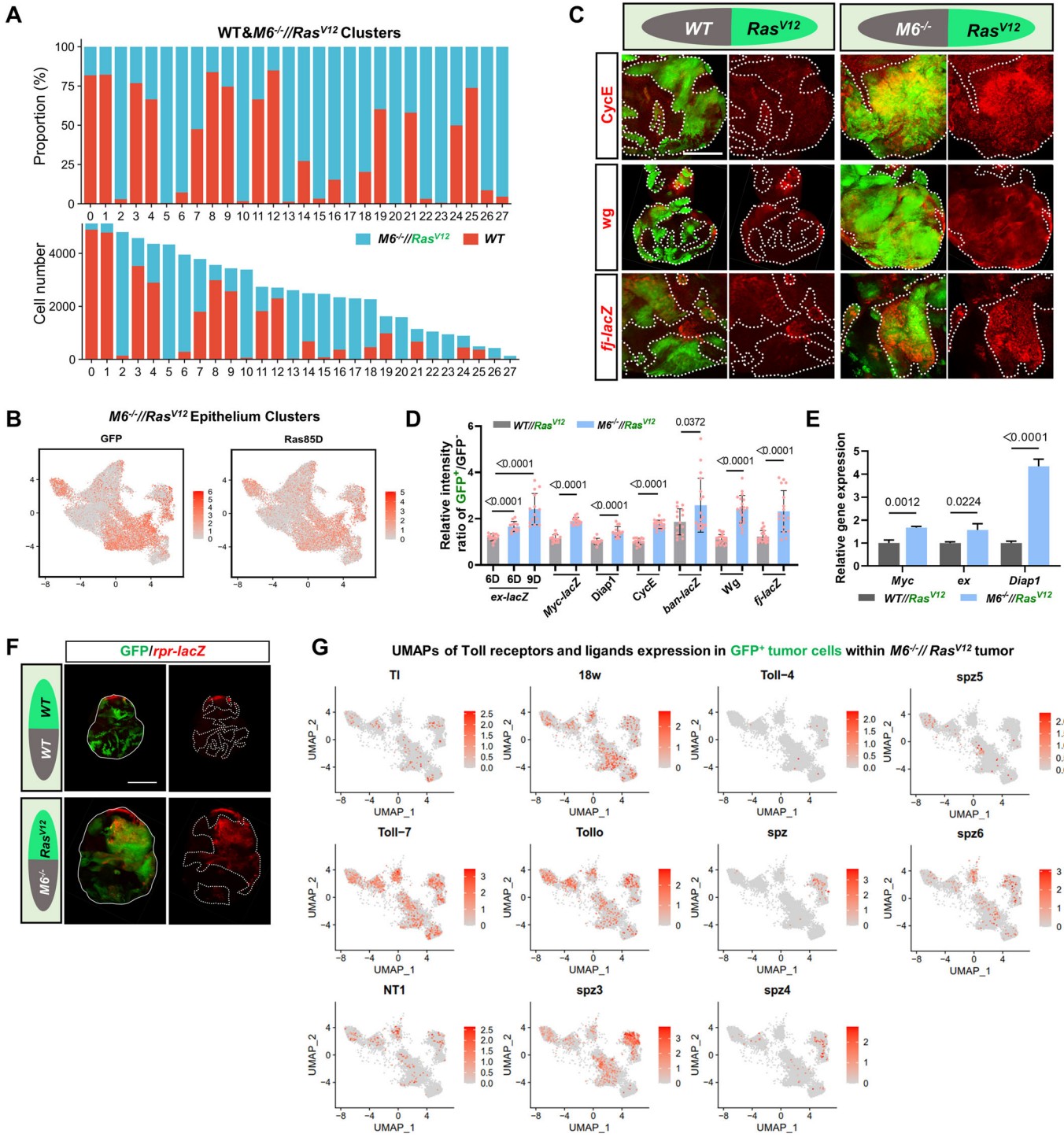

◀  **Figure EV3.  Both *Ras^{V12}//M6^{−/−}* and *Ras^{V12}*+Toll activation-induced tumors display enhanced expression of *yki* target genes.**

(A) The proportion (upper) and number (bottom) of cells of each cluster from *WT* or *Ras^{V12}//M6^{−/−}* discs are shown (data from Fig. 3B). (B) UMAP plots showing the marker genes expression of *GFP* and *Ras85D* for the *Ras^{V12}//M6^{−/−}* epithelial cells. (C) Eye-antennal discs harboring *Ras^{V12}//WT* (AEL-6) and *Ras^{V12}//M6^{−/−}* (AEL-7) clones were stained for CycE, Wg, and *fj-lacZ*. The white dashed lines demarcate the boundaries of GFP-positive clones. (D) Quantification of relative staining intensity across different experimental groups (from left to right, $n = 15, 12, 14, 14, 14, 13, 15, 16, 17, 15, 18, 20, 20, 18, 17$). Statistical analysis by unpaired two-tailed Student's t-test; mean ± SD. (E) qPCR analysis of mRNA levels for *yki* target genes (*Myc*, *ex*, and *Diap1*) in eye-antennal discs from the indicated genotypes ($n = 3$ independent experiments). Statistical analysis by unpaired two-tailed Student's *t* test; mean ± SD. (F) Eye-antennal discs bearing clones of *WT//WT* (AEL-6) and *Ras^{V12}//M6^{−/−}* (AEL-7) were stained with anti-β-galactosidase antibodies to visualize the *rpr-lacZ* reporter. White solid lines outline the borders of the eye-antennal discs, while white dashed lines indicate the boundaries of GFP-positive clones. (G) UMAP plots showing the expression of Toll receptors and ligands in the GFP+ epithelial cells of *Ras^{V12}//M6^{−/−}* tumors. Exact *P* values are shown in the corresponding panels. Scale bars: 100 μm (C) and 200 μm (F).

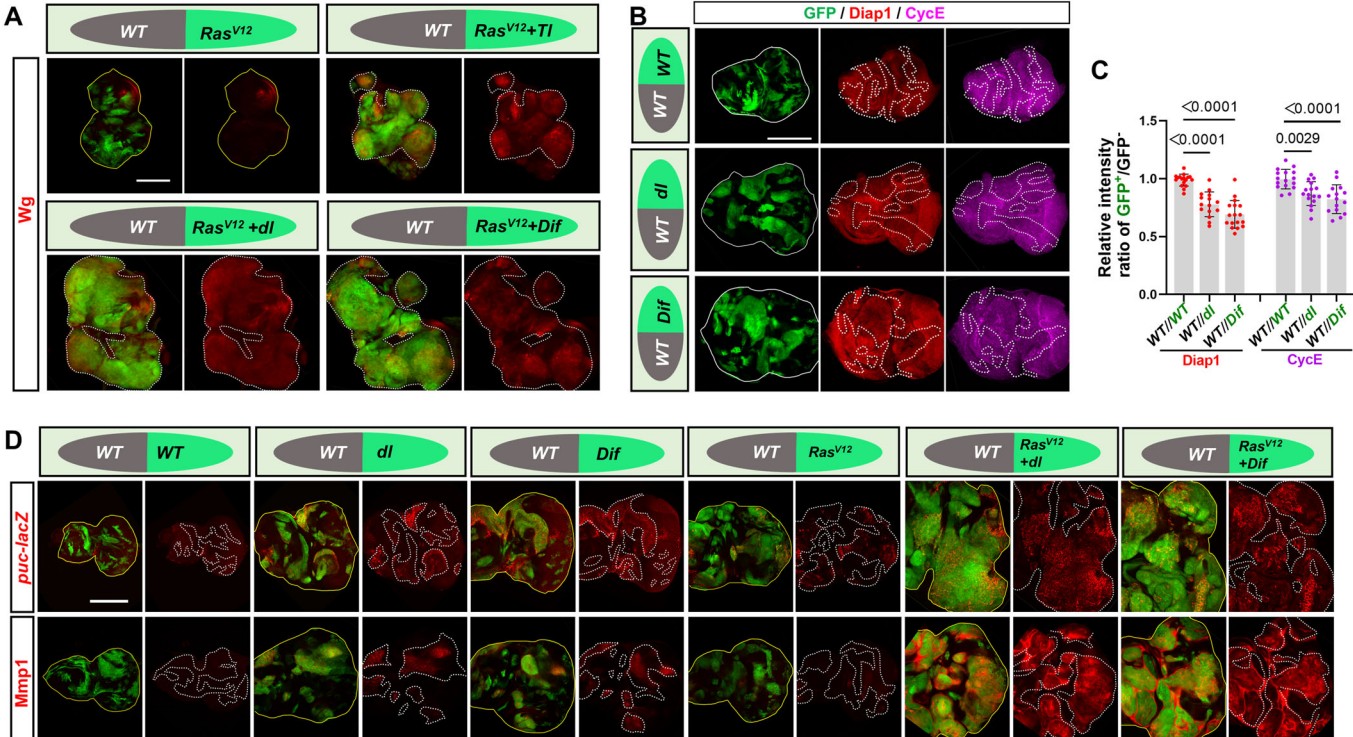

**Figure EV4. Toll pathway genetically acts upstream of JNK pathway.**

(A) Eye-antennal discs with $Ras^{V12}//WT$ clones (AEL-6), and $Ras^{V12} + Tl//WT$, $Ras^{V12}+ dl//WT$, $Ras^{V12} + Dif//WT$ clones (AEL-7) were stained with anti-Wg antibody. The yellow lines outline the borders of the eye-antennal discs, the white dashed lines mark the boundaries of the GFP-positive clones. (B) Eye-antennal discs bearing clones of $WT//WT$, $dl//WT$, $Dif//WT$ were stained with anti-Diap1 and anti-CycE antibodies (AEL-6). The white lines outline the borders of the eye-antennal discs, the white dashed lines mark the boundaries of the GFP-positive clones. (C) Quantification of relative staining intensity (from left to right, $n = 17, 15, 17, 16, 16, 15$). Statistical analysis by ordinary one-way ANOVA test; mean ± SD. (D) Eye-antennal discs with $WT//WT$, $dl//WT$, $Dif//WT$, $Ras^{V12}//WT$ clones (AEL-6), as well as $Ras^{V12}+ dl//WT$ and $Ras^{V12}+ Dif //WT$ clones (AEL-7), were stained with antibodies to detect $puc-lacZ$ and Mmp1 expression. Yellow lines outline the borders of the eye-antennal discs, while white dashed lines indicate the boundaries of GFP-positive clones. Exact $P$ values are shown in the corresponding panels. Scale bars: 200 μm (A, B, D).

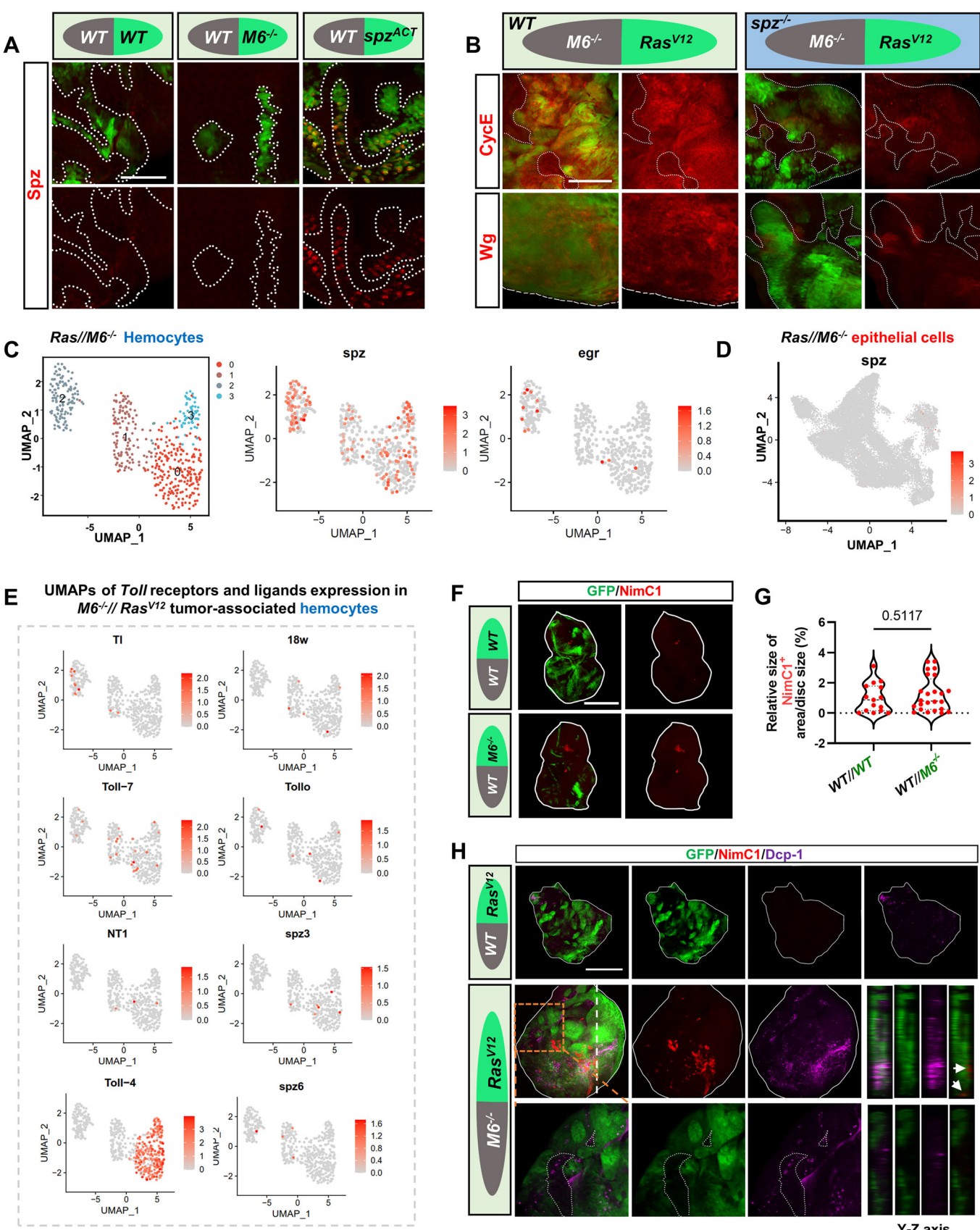

**Figure EV5. scRNA-seq analysis reveals the specific upregulation of *spz* in *Ras^V12^//M6^−/−^* attached hemocytes, rather than epithelial cells.**

(A) Eye-antennal discs bearing clones of *WT//WT*, *M6^−/−^//WT* and *spz^ACT^//WT* were stained with anti-Spz antibody (AEL-6). White dashed lines mark the boundaries of the GFP-positive clones. (B) Eye-antennal discs bearing clones of *Ras^V12^//M6^−/−^*, *Ras^V12^+spz^−/−^//M6^−/−^+spz^−/−^* were stained with anti-cycE and anti-Wg antibodies (AEL-7). White dashed lines mark the boundaries of the GFP-positive clones. (C) UMAP plot showing the re-clustered *Ras^V12^//M6^−/−^*-attached hemocytes. A total of 516 hemocytes were classified into four subclusters (left). UMAP plot showing the *spz* and *egr* expression in *Ras^V12^//M6^−/−^*-attached hemocytes. (D) UMAP plot showing the *spz* expression in *Ras^V12^//M6^−/−^* epithelial cells. (E) UMAP plot showing the expression of Toll receptors and ligands in *Ras^V12^//M6^−/−^*-attached hemocytes. (F) Eye-antennal discs bearing clones of *WT//WT*, *M6^−/−^//WT* were stained with anti-NimC1 antibody (AEL-6). White lines outline the borders of the eye-antennal discs. (G) Quantification of relative total NimC1^+^ area compared to the area of disc size for the indicated genotypes (from left to right, $n = 15, 24$). Violin plots represent kernel density estimation of data distribution, with the width proportionate to the number of points at each Y value. Scattered red dots represent individual data points. Statistical analysis by unpaired two-tailed Student's t-test. (H) Eye-antennal discs bearing clones of *Ras^V12^//WT* (upper row, AEL-6) and *Ras^V12^//M6^−/−^* (lower two rows, AEL-7) were stained with anti-NimC1 and anti-Dcp-1 antibodies. In *Ras^V12^//M6^−/−^* disc, bottom images show xy cross-section, right images show yz across-section. Straight white dashed line indicates the position of vertical section images, the white lines outline the borders of the eye-antennal discs, while the white dashed lines mark the boundaries of the GFP-positive clones. Arrows highlight NimC1-positive hemocytes adhering to GFP-positive cells. Exact *P* values are shown in the corresponding panels. Scale bars: 50 μm (A), 100 μm (B), and 200 μm (F, H).

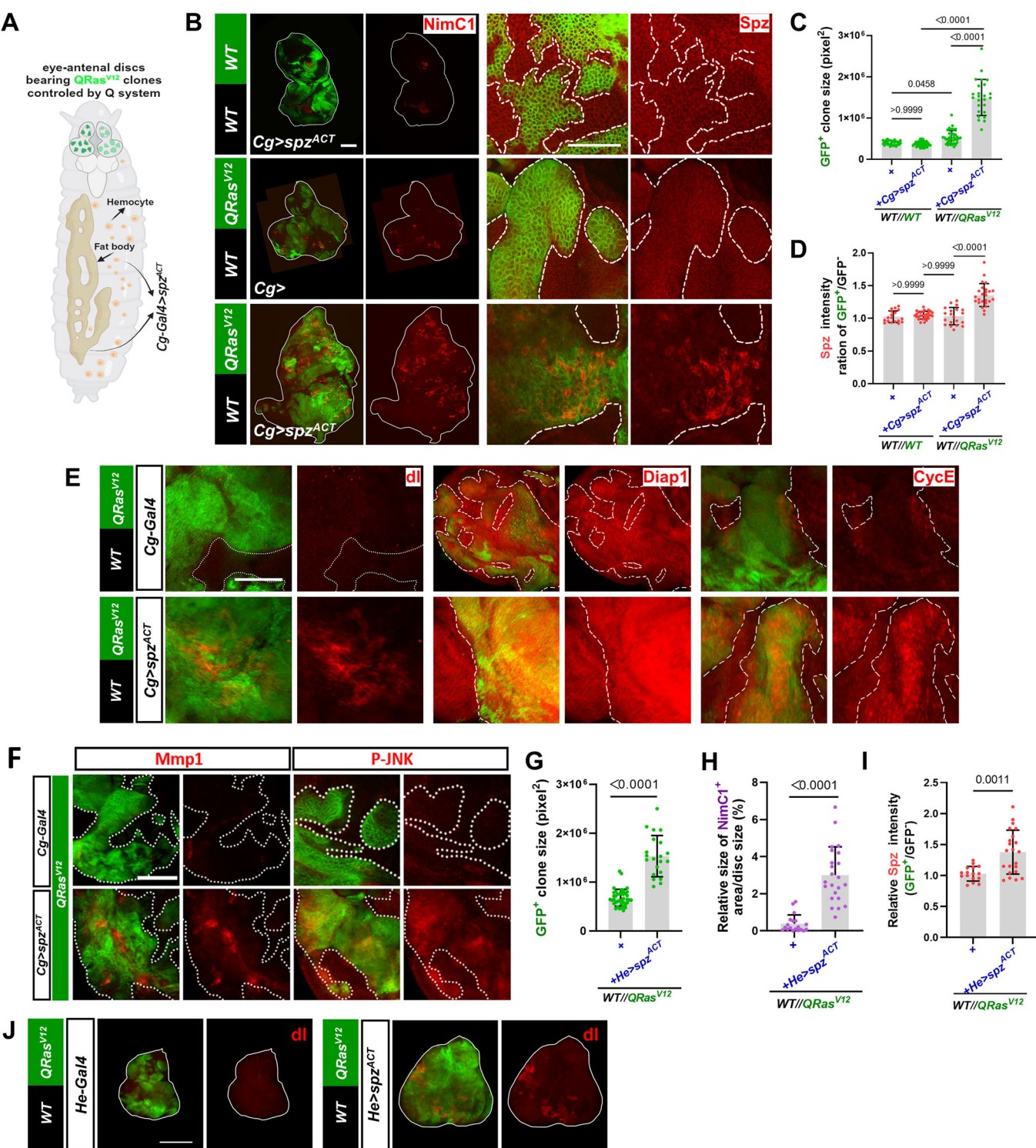

◄ **Figure EV6. Overexpression of activated *spz* driven by *Cg-gal4* and *He-gal4* promotes the overgrowth of *QRas^VI2* clones in the distal eye-antennal disc.**

(A) Schematic representation of the dual-expression system designed to overexpress activated *spz* (*spz^ACT*) in hemocytes and fat bodies under the control of *Cg* promoter and simultaneously inducing GFP-labeled *QRas^VI2* clones in the distal eye-antennal disc controlled by the Q system. (B) Eye-antennal discs bearing GFP-labeled clones of wild-type cells (upper row, AEL-6), *QRas^VI2*-overexpression clones without (middle row, AEL-6), or with *spz^ACT* overexpression driven by the *Cg* promoter (lower row, AEL-7) were stained with anti-NimC1 and anti-Spz antibodies. White solid lines mark the borders of the eye-antennal discs, while white dashed lines delineate the boundaries of the GFP-positive clones. (C) Quantification of GFP⁺ clones' size of indicated flies (from left to right, $n = 29, 48, 39, 25$). Statistical analysis by ordinary one-way ANOVA test; mean ± SD. (D) Quantification of relative Spz signal intensity in the indicated genotypes (from left to right, $n = 18, 29, 20, 25$). Statistical analysis by ordinary one-way ANOVA test; mean ± SD. (E) Eye-antennal discs harboring GFP-labeled *QRas^VI2* clones, either without (upper row, AEL-6), or with *spz^ACT* expression under the control of the *Cg* promoter (lower row, AEL-7) were stained with anti-dl, anti-Diap1, and anti-CycE antibodies. White dashed lines mark the boundaries of the GFP-positive clones. (F) Eye-antennal discs harboring GFP-labeled *QRas^VI2* clones, either without (upper row, AEL-6), or with *spz^ACT* expression under the control of the *Cg* promoter (lower row, AEL-7) were stained with anti-Mmp1, and anti-P-JNK antibodies. White dashed lines mark the boundaries of the GFP-positive clones. (G) Quantification of GFP-labeled *QRas^VI2* clones without or with *spz^ACT* expression under the *He* promoter (from left to right, $n = 40, 21$). Statistical analysis by unpaired non-parametric Mann-Whitney test; mean ± SD. (H) Quantification of relative total NimC1⁺ area compared to the area of disc size with indicated genotypes (from left to right, $n = 19, 23$). Statistical analysis by unpaired non-parametric Mann–Whitney test; mean ± SD. (I) Quantification of relative Spz signal intensity of indicated flies (from left to right, $n = 16, 22$). Statistical analysis by unpaired non-parametric Mann–Whitney test; mean ± SD. (J) Eye-antennal discs harboring GFP-labeled *QRas^VI2* clones, either without (AEL-6), or with *spz^ACT* expression under the control of *He* promoter (AEL-7) were stained with anti-dl antibody. White lines outline the borders of the eye-antennal discs. Exact *P* values are shown in the corresponding panels. Scale bars: 100 μm (**B**, **E**, **F**), 200 μm (**J**).

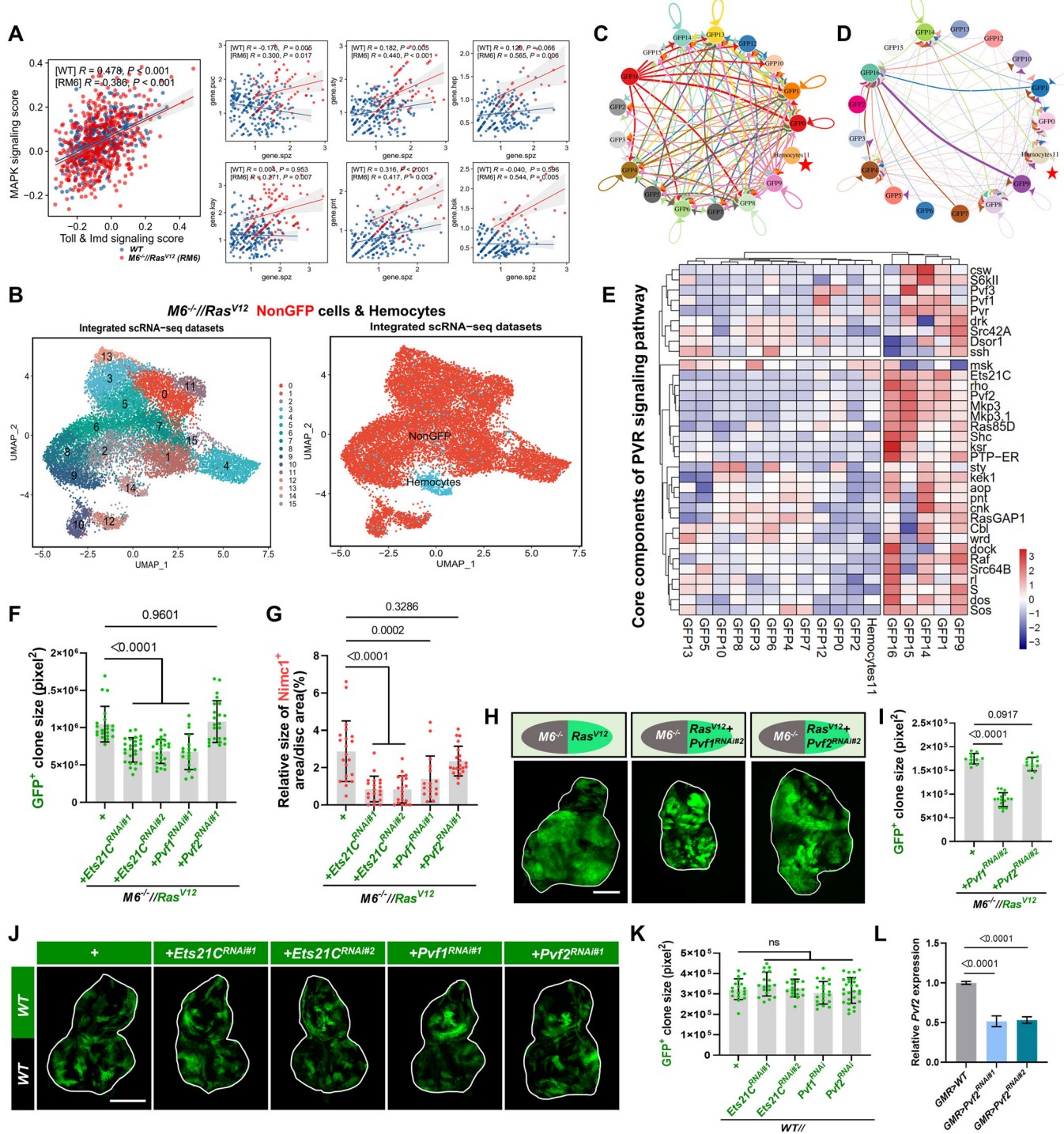

◀ **Figure EV7.** In *Ras^V12^//M6^−/−^ tumor-associated hemocytes, the expression level of spz is positively correlated with the expression of key genes in the JNK pathway.*

(A) Expression correlations analyses of WT hemocytes and *Ras^V12^//M6^−/−^* tumor-associated hemocytes. Toll & Imd signaling score and MAPK signaling are positively correlated (left), the expression of *spz* showed pan-positive correlations with MAPK signaling critical members, including *puc, sty, hep, kay, pnt,* and *bsk* (right). These correlations were determined by using Spearman's correlation analysis, with coefficients R and *P*-values reported, where $P < 0.05$ indicates statistical significance. (B) UMAP plots showing integrated hemocytes and non-GFP cells derived from *Ras^V12^//M6^−/−^* samples. A total of 16 clusters were identified (left, resolution $= 1.2$). (C) Circular plot showing a significant enrichment in the EGFR signaling pathway between multiple *Ras^V12^//M6^−/−^* GFP^+^ tumor cell clusters and *Ras^V12^//M6^−/−^* tumor-associated hemocytes (cluster 11, marked with a red asterisk). Notably, *Ras^V12^//M6^−/−^* GFP^+^ tumor cells do not communicate with hemocytes via the EGFR pathway. (D) Circular plot showing a significant enrichment in the FGFR signaling pathway between multiple *Ras^V12^//M6^−/−^* GFP^+^ tumor cell clusters and *Ras^V12^//M6^−/^* tumor-associated hemocytes (cluster 11, marked with a red asterisk). The analysis reveals weak intercellular communication between *Ras^V12^//M6^−/−^* GFP^+^ tumor cells and hemocytes via the FGFR pathway. (E) Heatmap displaying the expression patterns of PVR signaling pathway members across different clusters (as shown in Fig. 7F). Each row represents the average expression level of a gene, normalized by z-score. Clusters 1, 9, 14, and 15 from GFP^+^ tumors exhibit high expression levels of ligands (*Pvf1, Pvf2,* and *Pvf3*), whereas hemocytes show elevated expression of the receptor *Pvr*. (F) Quantification of GFP^+^ clone sizes for the indicated genotypes (from left to right, $n = 21, 26, 24, 18, 23$). Statistical analysis by ordinary one-way ANOVA test; mean ± SD. (G) Quantification of the relative NimC1^+^ area for the indicated genotypes (from left to right, $n = 21, 20, 20, 17, 21$). Statistical analysis by ordinary one-way ANOVA test; mean ± SD. (H) Eye-antennal discs bearing clones of *Ras^V12^//M6^−/−^, Ras^V12^+Pvf1^RNAi#2^//M6^−/−^,* and *Ras^V12^+Pvf2^RNAi#2^//M6^−/−^* were stained with anti-NimC1 antibody (AEL-7). White lines delineate the borders of the eye-antennal discs. (I) Quantification of GFP^+^ clone sizes for the indicated genotypes (from left to right, $n = 11, 16, 13$). Statistical analysis by ordinary one-way ANOVA test; mean ± SD. (J) Fluorescent eye-antennal discs bearing the following genotypes: *WT//WT, Ets21C^RNAi#1^//WT, Ets21C^RNAi#2^//WT, Pvf1^RNAi#1^//WT, Pvf2^RNAi#1^//WT* (AEL-6). White lines delineate the borders of the eye-antennal discs. (K) Quantification of GFP^+^ clone sizes for the indicated genotypes (from left to right, $n = 18, 18, 18, 20, 27$). Statistical analysis by ordinary one-way ANOVA test; mean ± SD. (L) qPCR analysis of *Pvf2* mRNA levels in the adult heads with the indicated genotypes ($n = 3$ independent experiments). Statistical analysis by ordinary one-way ANOVA test; mean ± SD. Exact *P* values are shown in the corresponding panels. Scale bars: 200 μm (H, J).

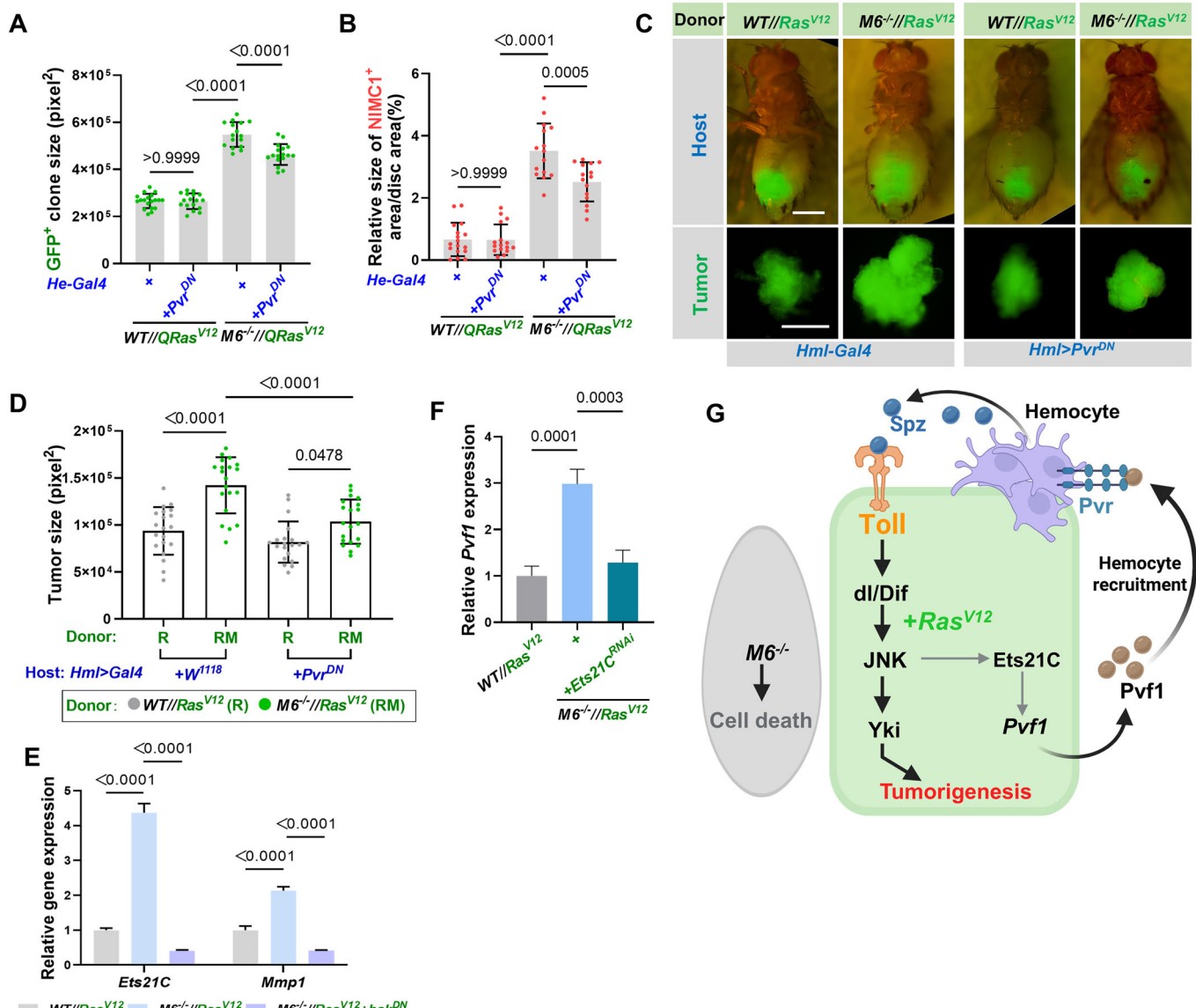

**Figure EV8. Pvr in hemocytes promotes the proliferation of *Ras^V12^//M6^−/−^* tumors and hemocyte adhesion.**

(A) Quantification of GFP+ clone sizes for indicated genotypes (from left to right, n = 19, 17, 16, 17). Statistical analysis by ordinary one-way ANOVA test; mean ± SD. (B) Quantification of the relative NimC1+ area for the indicated genotypes (from left to right, n = 17, 16, 15, 15). Statistical analysis by ordinary one-way ANOVA test; mean ± SD. (C) *Ras^V12^//WT* and *Ras^V12^//M6^−/−^* eye discs were transplanted into female adults with the following genotypes: *Hml-Gal4* and *Hml>Pvr^DN^*. Transplants were dissected 8 days post-transplantation for quantification. (D) Quantification of GFP+ transplanted tumors with indicated genotypes (from left to right, n = 20, 20, 20, 20). Statistical analysis by ordinary one-way ANOVA test; mean ± SD. (E) qPCR analysis of mRNA levels for *Ets21C* and *Mmp1* in eye-antennal discs of indicated flies (n = 3 independent experiments). Statistical analysis by ordinary one-way ANOVA test; mean ± SD. (F) qPCR analysis of mRNA levels of *Pvf1* in eye-antennal discs of indicated flies (n = 3 independent experiments). Statistical analysis by ordinary one-way ANOVA test; mean ± SD. (G) Proposed working model illustrating the progression of *Ras^V12^//M6^−/−^* tumors: Loss of the tricellular junction protein M6 promotes tumor malignancy in neighboring *Ras^V12^* clones by inducing secretion of Pvf1. Pvf1 acts as a chemoattractant, recruiting hemocytes through the Pvr receptor. These recruited hemocytes, in turn, activate a paracrine Spz-Toll signaling axis within *Ras^V12^* clones. Activation of the Toll pathway synergizes with oncogenic *Ras* to promote tumorigenesis through a JNK-Hippo signaling cascade. Exact P values are shown in the corresponding panels. Scale bar: 500 μm (C).

