## [Peer Review File · The EMBO Journal]

Hemocytes facilitate interclonal cooperation-induced tumor malignancy by hijacking the innate immune system in *Drosophila*

Sihua Zhao, Yifan Guo, Xiaoyu Kuang, Xiaoqin Li, Chenxi Wu, Peng Lin, Qi Xie, Zongzhao Zhai, Du Kong, and Xianjue Ma

Corresponding authors: Xianjue Ma (maxianjue@westlake.edu.cn) , Du Kong (kongdu@email.sdu.edu.cn)

Review Timeline:

Submission Date:	16th Feb 25
Editorial Decision:	17th Mar 25
Revision Received:	14th Jun 25
Editorial Decision:	16th Jul 25
Revision Received:	24th Jul 25
Accepted:	7th Aug 25

Editor: Ieva Gailite

Transaction Report:

Dear Xianjue,

Thank you for submitting your manuscript for consideration by the EMBO Journal. We have now received comments from a full set of reviewers, which are included below for your information.

As you will see, all reviewers are generally positive in their assessment and appreciate the contribution of the study to the research field. At the same time, they indicate several concerns that would be important to address in the revised study, in particular regarding the involvement of the Hippo pathway downstream of Toll signalling and the mechanisms mediating hemocyte recruitment. They also indicate several instances in which the findings contradict the existing literature, which should be discussed and clarified in the manuscript. From my side, I find these points generally reasonable. Therefore, I invite you to address these comments in a revised manuscript. I think that it would be useful to discuss the revision in more detail via email or phone/videoconferencing - please let me know which option you prefer.

We generally allow three months as standard revision time, which can be extended to six months in the case of major revisions. Should you foresee a problem in meeting this deadline, please let us know in advance to discuss an extension. As a matter of policy, competing manuscripts published during this period will not negatively impact on our assessment of the conceptual advance presented by your study. However, please contact me as soon as possible upon publication of any related work to discuss the appropriate course of action.

When preparing your letter of response to the referees' comments, please bear in mind that this will form part of the Review Process File and will therefore be available online to the community. For more details on our Transparent Editorial Process, please visit our website: <https://www.embopress.org/page/journal/14602075/authorguide#transparentprocess>. Please also see the attached instructions for further guidelines on preparation of the revised manuscript.

Please feel free to contact me if you have any further questions regarding the revision. Thank you for the opportunity to consider your work for publication. I look forward to discussing your revision.

With best regards,

Ieva

- a point-by-point response to the referees' comments, with a detailed description of the changes made (as a word file).
- a word file of the manuscript text.
- individual production quality figure files (one file per figure)
- a complete author checklist, which you can download from our author guidelines

(<https://www.embopress.org/page/journal/14602075/authorguide>).

- Expanded View files (replacing Supplementary Information)

- a Reagents and Tools Table as part of the Methods section, which can be downloaded from our author guidelines

(<https://www.embopress.org/page/journal/14602075/authorguide#structuredmethods>)

We realize that it is difficult to revise to a specific deadline. In the interest of protecting the conceptual advance provided by the work, we recommend a revision within 3 months (15th Jun 2025). Please discuss the revision progress ahead of this time with the editor if you require more time to complete the revisions.

Referee #1:

This paper from the group of Dr. Ma and Dr. Kong is an extremely interesting and detailed dissection of the intercellular dynamics that can influence epithelial tumor growth in *Drosophila*. The authors discover that a mutation linked to the tricellular junction protein M2 in epithelial cells leads to the overgrowth of flanking neighbors carrying a tumorigenic RasV12 mutation. The authors conduct RNA seq, both bulk and single cell, bioinformatic analysis, and mostly elegant and rigorous genetics to identify the signaling pathways involved. They find that the M6-/RasV12 juxtaposition leads to activation of Ets21C and the expression of Pvf1-3 in the RasV12 cells. This attracts macrophages, oddly only to the neighboring M6- cells, not the RasV12 cells that express the attractant. Hemocytes secrete spaetzle, leading to activation of the Toll and Hippo pathway in the RasV12 cells to induce their overgrowth.

The well conducted and rigorously quantified experiments that undergird the above conclusions are the following: Through RNAi knockdown they show that the Toll pathway is necessary in the transformed cells for the influence of the M2 mutant neighbors, and even show that activating the Toll pathway in transformed cells is sufficient to induce overgrowth when they are flanked by normal neighbors. They find that the Hippo pathway and Yorkie lie downstream of the Toll pathway in transformed cells for overgrowth. They demonstrate that hemocytes are attracted to the M6- mutant tissue flanking RasV12 epithelial cells, and that hemocyte production of Spaetzle is what induces overgrowth of RasV12 tissue by inducing TI signaling. They show that activated spaetzle production by hemocytes is sufficient to induce their own recruitment to wt tissue flanking RasV12 clones and RasV12 clone overgrowth when flanked by wt neighbors. They elucidate that ablation of hemocytes is sufficient to prevent the overgrowth of the M6-/ RasV12 heterogeneous clones. They show that RasV12 clones flanked by M6- ones express higher levels of Ets21C and Pvf1-3, and hemocytes in these genetic backgrounds express higher levels of PVR. Knocking down Pvf1 expression in the RasV12 mutants or PVR in the hemocytes

This paper is broadly interesting for the fly field, particularly those in epithelial biology, tumor growth, and immune action, as well as those studying Hippo signalling. I also believe vertebrate cancer biologists will find it interesting. I have a few questions which I believe can be addressed potentially by some experiments, but probably only by examination of their extant data and some textual additions.

Major questions:

1) What is happening with the AMPs?

a) Can the authors clarify what the role actually is for the AMPs? While the authors provide evidence that these are upregulated in the M6-/ RasV12 context, the importance of this is not examined. Have the authors knocked out one or a set of the AMP targets they identify and seen an effect on tumor overgrowth? If they have and there is no phenotype, they should state that, and say that there may be redundancy or other downstream targets of DI and Df which mediate the proliferative effect.

b) The data on DIAP1 in particular seems inconsistent from assay to assay. The Diap1 antibody stain in 3H, I which shows an extremely high basal level of protein expression and only a small increase in the M6- condition does not match the gene expression analysis in 3J which shows a very high induction of Diap1.

Is this an error in their figure assembly?

c) The authors data runs counter to previous studies showing a roles for AMPs in inducing death of dlG tumors in flies (Parvy et al, 2019, PMID 31358113) and a role for AMPs in killing vertebrate tumors (PMID 28422728). In this paper the induction of AMPs is associated with tumor growth. It can be that different tumor genetics might lead to different sensitivities. However, another explanation could be that the current authors' recruited hemocytes do not express TNFs. Analysis of their RNA seq data should clarify this. Some comment in the discussion as to the difference in AMP sensitivity between theirs and the previously published findings would be beneficial for the reader.

d) Alternatively the authors could just remove all the AMP data. I think the paper is strong enough without it and the authors could clarify the precise role in later papers.

2) RasV12 independent effect of the TI pathway on proliferation.

The authors bury the data in S2C which shows that activation of the TI pathway in fact leads to cellular overgrowth even without the RasV12 mutation. The size increase is around 2x less than in the presence of the RasV12 mutation, but nonetheless is interesting and deserves a mention in the main text.

3) TI and Hippo pathway intersection

The authors' findings run counter to a previous paper PMID: 26824654 in which the Toll pathway leads to Yki inactivation in the fat body to enable the expression of innate immunity including AMP expression.

Again this could be due to distinctions in the two tissues' responses, but could the authors please comment on this in the discussion?

Minor points:

3) Genetics

a) The authors use a well known M6W186* mutation but do not show that this is the cause of the overgrowth phenotype of the RasV12 neighbors by rescuing the phenotype through expression of M6. Doing the rescue is time consuming and is not key for their story since this paper does not focus on the mechanisms by which the "M6" mutant tissue is signaling to the RasV12 neighbors to lead to the Pvf1 expression which recruits hemocytes and starts the TI signaling. I don't ask for the experiment, but just for language stating that the evidence supports but does not prove that loss of a tricellular junction protein is the initiating cause.

b) While the authors use two RNAis to knockdown dl, Df, and Ets21c in some of their experiments they don't do it in all of them. They do not use two TI RNAis, nor two Tub RNAis, nor two PI RNAis. The fact that they are knocking down multiple members of the same pathway and show the same phenotype, and show the opposite phenotype upon overexpression of members of that pathway convinces me not to request the usual level of rigor for RNAi use for each gene that is standard in the field (ie two independent RNAis per gene). However it would have been cleaner data if they had. In particular the absence of an effect from the Pvf2 RNAi is not convincing if only one.

Textual issue:

3l y axis, I think they mean ratio not ration

5G is this showing the size of the sum of all the clusters for an eye disc or the size of each individual cluster?

Referee #2:

In this study, Zhao, Guo and collaborators use a modified version of the MARCM technique to generate mosaic eye disc with neighbouring clones either expressing a constitutively active form of the oncogene Ras (RasV12), or mutant for the tricellular component M6. They show that RasV12 clones grow massively only when confronted with M6 mutant clones (RasV12//M6) and not when confronted to wild-type tissues (RasV12). Using scRNA-Seq of these RasV12//M6 tumours they identify the transcriptomes of their constitutive cell types identifying different clusters for the eye disc epithelial cells, glial cells, and attached haemocytes. Mechanistically, they show that the Toll/tube/pelle/dorsal/Dif pathway is activated and required in the RasV12 overgrowing tissues. They propose that the Toll pathway promotes Yki activity thus promoting growth. Using dual expression systems and transplantation assays, they show that spz producing haemocytes are required for RasV12//M6 tissue growth and dl activation. Finally, they identify Pvf1 secreted by the RasV12 cells as the activator of haemocytes.

This is a huge amount of work, and the majority of experiments supports the claims which are original. Indeed, the main originality lies in the difference of behaviour between RasV12 and RasV12//M6 mosaic tissues in terms of growth potential. The second original finding is that the Toll pathway is required for tumour growth, while it had been suggested to promote the death of dlg or scrib mutant tissues. There are however some points that need to be clarified.

Main comments.

1. The authors need to state exactly at what age the different images and measurements of the discs have been performed and ensure as much as possible that they compare discs of similar age. Indeed, one of the obvious observation from Figure 1C is that RasV12//M6 animals have an extended larval life (even though most will pupate Fig 1F), which would translate into increased time for proliferation. Strict timing of the different conditions is critical, and should be indicated.

2. Many IF panels have a high magnification which makes interpretation complicated. This is especially true for Fig 3H, 4A, 5D, S3E. In these figures, the point is to compare RasV12 GFP positive tissues to GFP negative tissues (M6 mutant or WT), and compare the levels of staining. At the magnification shown, it is impossible to find enough GFP negative tissues to compare.

3. The Hippo pathway down-regulation is hard to interpret. Indeed, removing yki or overexpressing wts would severely reduce the growth of any imaginal disc tissue. The fact that they are epistatic to RasV12//M6 and to Toll pathway is thus not that strong of an argument. We are thus left with the yki targets, many of which are not specific of the Hippo pathway (myc, Diap1, CycE, ban, wg, fj are also targets of Notch for instance). The pictures shown for DIAP1 are not convincing given point 2. Ex-LacZ is more specific. A more detailed analysis should be performed at earlier time points (smaller clones) to better document Ex-LacZ activation.

The authors should also discuss models as to how Toll/pll/dl/Dif affects Hippo and Yki activity.

4. The description of the scRNA-Seq analysis is hard to follow, in particular why the different re-clusterings and fusions are made. I am not sure I understood the >65% occupancy and the way it is documented in Fig. 3D. Which clusters are GFP? Do they correspond mainly to RasV12 tumoral specific tissues?

5. The spz expressing haemocytes appear to be concentrated on M6 mutant tissues which will be eliminated (Fig. 5A&H). Could authors show Dcp1 staining? Is there increased cell death at M6 tissues (like in WT discs)? Is there also increased haemocytes expressing spz attracted to M6 mutant cells in WT discs (Fig S1)? This is important, since similar haemocytes recruitment to M6 mutant clones would thus indicate that one signal attracting the haemocytes are these dying cells, which would change the cascade of events in the model.

6. Where is Toll expressed? Is it more abundant on RasV12 cells priming them to respond to spz? Could that be inferred for the scRNA-Seq or from IFs?

7. The authors never discuss what determines the different responses between a RasV12//WT and a RasV12//M6 mosaic tissue? This is a major weakness, and authors should at least discuss this.

8. Ets21C is part of the oncomodule that was shown required for eye discs RasV12 clone growth. Its removal in restores the morphology of the discs, with increased M6 mutant tissues and separated GFP clones. This is similar to what is seen for tube or dl RNAi. The phenotypes thus looks thus dependent on some kind of cell competition. Could the author comment on this in terms of model (see point 7)?

The removal of Pvf1 does not make discs of similar shape as the removal of tube or dl. They are smaller, but the M6 tissue is almost completely eliminated. Is that specific to this particular disc (Fig 7H), or could this reflect another role of Pvf1?

Minor comments.

Authors should show the efficacy of the RNAi lines used, in particular those for which they did not observe modifications

Authors should not use the term of tumour heterogeneity and subclones. Indeed the genetic trick used here generates clones with an oncogene and then adjacent clones with M6 mutations. These are thus two independent alterations that will compete. This is not tumour heterogeneity, where some cells within the tumour mass acquire different mutations. At best it is two fighting or cooperating tumours, but not tumour heterogeneity. Furthermore, the sentence in the discussion lines 328-329 "Our data here reveals a complex intercellular crosstalk both within epithelial cells bearing distinct oncogenic mutations and between tumors and hemocytes" is misleading. Indeed M6 mutation is not oncogenic per se, since M6 mutant cells are normally eliminated.

Discussion is too short. Authors should discuss in greater details

- The specificity of RasV12//WT vs RasV12//M6
- How Toll could be activated specifically in RasV12 cells?
- How Toll/dl impinge on Hippo?
- How the RasV12//M6 compare to the undead cells model where Toll / Toll9 has been shown to initiate the caspase activation via hid/rpr (but later blocked) and to attract haemocytes which through JNK signalling promote the production of pro-proliferative signals (wg, upd...).

Referee #3:

The authors present a narrative built upon a series of genetic interaction experiments, proposing that RasV12 cells surrounded by M6 mutant cells secrete Pvf1, which in turn stimulates hemocytes to secrete Spz. Spz then promotes overgrowth of RasV12 cells by inhibiting the Hippo pathway. While the proposed mechanism is plausible, additional data would strengthen the authors' claims and help address some mechanistic gaps.

Main Comments

1. The rationale for focusing on M6 is not clearly articulated. The authors should clarify why M6 was chosen as a point of investigation. Specifically, how does M6 lead to Pvf1 upregulation in Ras^{V12} cells? Is the effect of M6 attributable solely to cell death or to structural defects in the cells? It would be informative to assess the impact of reaper-induced apoptosis or other manipulations of tricellular junctions to disentangle these possibilities.

2. The connection between Toll signaling and the Hippo pathway remains unclear. How exactly does Toll regulate Hippo? A mechanistic link between these pathways should be better defined. Generally, inhibition of Yorkie leads to suppression of cell proliferation, and this aspect should be addressed to clarify the proposed pathway.

3. The role of Pvf1 in hemocyte activation is insufficiently resolved. It is not clear whether Pvf1 functions to recruit hemocytes to Ras^{V12} cells or simply activates hemocytes irrespective of recruitment. Related to this, in Fig. 6B, how are hemocytes attracted to eye disc Ras tumors? According to Fig. 5E, Ras^{V12}-only tumors do not attract substantial numbers of hemocytes. Additionally, it appears that free Spz secreted by hemocytes alone is not sufficient to activate Toll signaling and drive Ras^{V12} malignant transformation.

Specific Comments

4. In Fig. 2, bulk RNA-seq of the disc likely includes hemocytes. This should be explicitly acknowledged and discussed.

5. In Fig. 5A, the side view is difficult to interpret. The Spz-expressing cells appear to be disc proper cells rather than hemocytes. Although later figures include NimC1 staining to mark hemocytes, the authors should clearly distinguish disc proper cells from hemocytes at this point in the figure.

6. The Discussion section should be expanded to better articulate the current gaps in knowledge and explain how the present findings contribute to addressing these open questions.

7. The authors note that *Drosophila* possesses multiple Toll receptors and ligands. Are other Toll ligands or receptors expressed in tumors or hemocytes? A brief re-analysis of the RNA-seq data to address this point would be valuable.

Response to reviewer's comments (MS# EMBOJ-2025-120470):

Dear Reviewers and Editor,

Thank you very much for your favorable review and insightful comments on our manuscript. We have conducted additional experiments and made necessary revisions to address the concerns raised. Below is a point-by-point response to the reviewers' comments. For your convenience, changes in the revised manuscript are highlighted in red font. Most of the new results have been included in the revised manuscript; however, due to space constraints, some results and explanations are provided exclusively in this response letter.

We sincerely hope that the revised manuscript meets your expectations and is deemed satisfactory.

Referee #1

This paper from the group of Dr. Ma and Dr. Kong is an extremely interesting and detailed dissection of the intercellular dynamics that can influence epithelial tumor growth in Drosophila. The authors discover that a mutation linked to the tricellular junction protein M6 in epithelial cells leads to the overgrowth of flanking neighbors carrying a tumorigenic Ras^{V12} mutation. The authors conduct RNA seq, both bulk and single cell, bioinformatic analysis, and mostly elegant and rigorous genetics to identify the signaling pathways involved. They find that the M6-/Ras^{V12} juxtaposition leads to activation of Ets21C and the expression of Pvf1-3 in the Ras^{V12} cells. This attracts macrophages, oddly only to the neighboring M6- cells, not the Ras^{V12} cells that express the attractant. Hemocytes secrete spaetzle, leading to activation of the Toll and Hippo pathway in the Ras^{V12} cells to induce their overgrowth.

The well conducted and rigorously quantified experiments that undergird the above conclusions are the following:

Through RNAi knockdown they show that the Toll pathway is necessary in the transformed cells for the influence of the M3 mutant neighbors and even show that activating the Toll pathway in transformed cells is sufficient to induce overgrowth when they are flanked by normal neighbors. They find that the Hippo pathway and Yorkie lie downstream of the Toll pathway in transformed cells for overgrowth. They demonstrate that hemocytes are attracted to the M6- mutant tissue flanking RasV12 epithelial cells, and that hemocyte production of Spaetzle is what induces overgrowth of RasV12 tissue by inducing Tl signaling. They show that activated spaetzle production by hemocytes is sufficient to induce their own recruitment to wt tissue flanking RasV12 clones and RasV12 clone overgrowth when flanked by wt neighbors. They elucidate that ablation of hemocytes is sufficient to prevent the overgrowth of the M6-/ RasV12 heterogeneous clones. They show that RasV12 clones flanked by M6- ones express higher levels of Ets21C and Pvf1-3, and hemocytes in these genetic backgrounds express higher levels of PVR. Knocking down Pvf1 expression in the RasV12 mutants or PVR in the hemocytes.

This paper is broadly interesting for the fly field, particularly those in epithelial biology, tumor growth, and immune action, as well as those studying Hippo signalling. I also believe vertebrate cancer biologists will find it interesting. I have a few questions which I believe can be addressed potentially by some experiments, but probably only by examination of their extant data and some textual additions.

Response: Thank you for recognizing the overall significance and the relevance of our research topic. We have made necessary revisions to further improve the manuscript.

Major questions:

1) What is happening with the AMPs?

a) Can the authors clarify what the role actually is for the AMPs? While the authors provide evidence that these are upregulated in the $M6^{-}/Ras^{V12}$ context, the importance of this is not examined. Have the authors knocked out one or a set of the AMP targets they identify and seen an effect on tumor overgrowth? If they have and there is no phenotype, they should state that, and say that there may be redundancy or other downstream targets of *Dl* and *Df* which mediate the proliferative effect.

Response: Thank you for your insightful feedback and for highlighting the critical need to clarify the role of AMPs in our tumor model. To address the role of AMPs in $Ras^{V12}/M6^{-}$ -induced tumorigenesis, we deleted one copy of *Defensin* (*Def*), a previous reported AMP responsible for fly tumor progression (PMID: 31358113). Additionally, we utilized a mutant strain deficient in multiple key AMPs, where one copy of each AMP gene (*Def*, *AttC*, *Dro*, *AttA*, *AttB*, and *Dpt*) was deleted in $Ras^{V12}/M6^{-}$ tumors. Our findings revealed that neither heterozygous mutations in *Def* nor the simultaneous knockout of multiple AMPs rescued the overgrowth of $Ras^{V12}/M6^{-}$ tumors (**Author response Figure 1**).

Author response Figure 1. Eye-antenna discs bearing *ey-Flp*-MARCM-induced mosaics of indicated genotypes.

Based on these observations, we hypothesize that the induction of AMPs in $Ras^{V12}/M6^{-}$ tumors likely reflects non-specific immune activation rather than a direct role in driving oncogenesis. We propose two non-exclusive scenarios: 1) **AMPs as correlative biomarkers**. The upregulation of AMPs in $Ras^{V12}/M6^{-}$ tumors may result from Toll pathway activation via the NF- κ B homolog *dl* (Fig. 2B-D). This suggests that AMPs are not essential for tumor growth but serve as secondary markers of immune activation. 2) **AMPs as potential modulators of TME**. Certain AMPs, such as *Defensin*, are known to degrade basement membrane components, facilitating their anti-cancer functions (PMID: 31358113).

Interestingly, some on-going projects in our lab also reveal a consistent induction of Toll pathway in multiple malignant tumors. As shown below, we have previously performed an EMS-induced genetic screen in fly eye epithelium, aiming to find novel tumors suppressor genes whose mutation could synergize with Ras^{V12} to induce malignancy, among which we uncovered multiple alleles targeting different genes (unpublished, **Author response Figure 2**).

Author response Figure 2. Eye-antennal discs bearing *ey-Flp*-MARCM-induced mosaics of indicated genotypes.

We further performed bulk-RNA-seq analysis using tumor samples dissected from different time points, we observed consistent upregulation of AMPs across multiple *Drosophila* tumor models, especially in late-stage tumor samples (unpublished, **Author response Figure 3**).

Author response Figure 3. Heatmaps of representative AMPs upregulated in multiple *Ras^{V12}*-related eye-malignant tumors.

Collectively, while AMPs are robustly induced in our tumor model and correlate with malignancy across contexts, our data imply that AMPs might NOT play a direct oncogenic role. Instead, AMPs likely reflect a parallel immune response shaped by Toll signaling. We agree that

additional investigations are necessary to dissect the roles of AMP induction in potential immune-tumor interactions during tumorigenesis. We have revised our conclusions accordingly to reflect these findings, and the new data have been incorporated into the revision.

b) The data on DIAP1 in particular seems inconsistent from assay to assay. The Diap1 antibody stain in 3H, I which shows an extremely high basal level of protein expression and only a small increase in the M6- condition does not match the gene expression analysis in 3J which shows a very high induction of Diap1. Is this an error in their figure assembly?

Response: We appreciate your attention to detail and thanks for pointing out the apparent discrepancy between DIAP1 protein and mRNA levels in our study. However, this is **NOT** caused by a error during figure assembling. *Diap1* mRNA is known to be directly induced by the Yki/Sd transcriptional complex (PMID: 18258486). However, the protein level of DIAP1 can be regulated post-transcriptionally, including ubiquitination. For instance, certain **pro-apoptotic factors** in *Drosophila*, such as *Reaper*, *Hid* (Head involution defective), and *Grim*, promote DIAP1 **degradation** by binding to it and facilitating its ubiquitination. This leads to proteasomal degradation of DIAP1, freeing caspases to initiate apoptosis (PMID: 10675328; 12021767; 10481910). In line with this, despite the overgrowth of GFP-positive tumor clones in *Ras^{V12}//M6^{-/-}* tumors, we observed strong induction of *rpr* transcription, as demonstrated by a *rpr-lacZ* reporter (**Author response Figure 4**). Therefore, the apparent paradox of high *Diap1* mRNA but modest DIAP1 protein arises from a dynamic equilibrium between Yki-mediated transcriptional upregulation and Rpr-mediated protein degradation.

Author response Figure 4. Immunofluorescent images of *rpr-lacZ* staining in eye-antennal discs bearing *WT* and *Ras^{V12}//M6^{-/-}* clones.

We have included the following description in the revised manuscript. “The modest increase in DIAP1 protein despite robust *Diap1* mRNA upregulation reflects a dynamic equilibrium between Yki-mediated transcriptional activation and rpr/rpr-like-mediated protein degradation.”

c) The authors data runs counter to previous studies showing a role for AMPs in inducing death of *dlg* tumors in flies (Parvy et al, 2019, PMID 31358113) and a role for AMPs in killing vertebrate tumors (PMID 28422728). In this paper the induction of AMPs is associated with tumor growth. It

can be that different tumor genetics might lead to different sensitivities. However, another explanation could be that the current authors' recruited hemocytes do not express TNFs. Analysis of their RNA seq data should clarify this. Some comments in the discussion as to the difference in AMP sensitivity between theirs and the previously published findings would be beneficial for the reader.

Response: Thank you for raising this point and highlighting the need to contextualize our findings within the broader literature. We agree with you that the apparent contradiction between our data (AMPs are generally elevated in fly tumors) and previous studies (AMPs suppress tumor growth) could be largely explained by the **difference in tumor genetic background**. The homozygous *dlg* mutants disrupt epithelial polarity, leading to invasive overgrowth. AMPs (e.g., Defensin) are secreted by Toll/IMD pathway activation in systemic immune organs (fat body/trachea) and directly kill polarity-deficient cells. While in our model, systemically heterozygous mutations in key AMPs do not significantly suppress *Ras^{V12}//M6^{-/-}* tumors (**Author response Figure 1**). Combining our unpublished observations that AMPs transcription were significantly elevated in 10 different types of *Ras*-mediated *Drosophila* eye epithelial tumors in a malignancy stage-dependent manner (**Author response Figure 2-3**), we speculate that AMPs upregulation may serve as a general biomarker for *Ras*-related tumor progression.

Regarding the critical point about TNF family ligands, which are classic mediators of AMP-induced cell death. In our bulk-seq data, the expression of *eiger* (*egr*) was not upregulated in *Ras^{V12}//M6^{-/-}* tumors ($\log_2FC = 0.48$, FDR = 0.42). Our scRNA-seq analysis reveals that tumor-associated hemocytes in *Ras^{V12}//M6^{-/-}* tumors significantly upregulated *spz* transcription, but not that of *egr*, the *Drosophila* homolog of TNF (**Author response Figure 5**), supporting a TNF-independent AMP secretion program. In contrast, *egr* mRNA was intensively elevated in *dlg* tumors (PMID: 24582964).

Author response Figure 5. UMAPs of *spz* (A) and *egr* (B) expression in tumor-associated hemocytes in *Ras^{V12}//M6^{-/-}* tumors.

Together, we acknowledge the apparent contradiction but respectively argue that it reflects contextual specificity and complexity of AMP in tumorigenesis. We totally agree that it would be interesting to further clarify the mechanisms underlying the differences in various contexts.

d) Alternatively, the authors could just remove all the AMP data. I think the paper is strong enough without it and the authors could clarify the precise role in later papers.

Response: Thank you for your thoughtful suggestion to streamline the manuscript by removing the AMP data. After careful consideration, we believe that retaining these data is critical to

preserving the scientific rigor and broader impact of the study. However, we propose the following revisions to address your concerns regarding focus and clarity, while preserving the integrity of the manuscript.

1. Rationale for retaining AMP data.

(a) Biological Relevance:

- The upregulation of AMPs in $Ras^{V12}/M6^{-/-}$ tumors is a hallmark of oncogenic stress observed across multiple tumor models in our laboratory (e.g., $Ras^{V12}/scrib^{-/-}$, $Ras^{V12}/Igf1^{-/-}$, or $Ras^{V12}/msn^{-/-}$ -driven tumors). This recurring pattern suggests a conserved link between oncogenic signaling and immune modulation, even if AMP functional roles vary contextually.
- Omitting this *data* risks obscuring a potential mechanism connecting oncogene-driven stress to immune dysregulation.

(b) Methodological Transparency:

- AMP expression was validated via orthogonal *methods* (qPCR and RNA-seq) and aligns with prior studies (PMID: 31358113). Removing these data would undermine the reproducibility and methodological robustness of our findings.

(c) Hypothesis Generation:

- While AMPs' role in our model remains unresolved, their correlation with tumor growth (supported by RNA-seq data from 10 independent tumor models) and Hippo pathway inactivation provides foundational insights for future studies. Eliminating this data would hinder exploration of non-canonical AMP functions in cancer biology.

2. Revisions to address concerns.

To enhance focus and readability, we relocate functional experiments (e.g., *Def* mutant studies) to EV Figures and clarify contextual limitations in the manuscript. We have included the following description in the revision: “*While AMPs are transcriptionally induced in $Ras^{V12}/M6^{-/-}$ tumors, their functional role in this context requires further investigation. Given the complexity of AMP-immune crosstalk, a further comprehensive mechanistic analysis is required in future studies.*”

2) Ras^{V12} independent effect of the Tl pathway on proliferation.

The authors bury the data in S2C which shows that activation of the Tl pathway in fact leads to cellular overgrowth even without the Ras^{V12} mutation. The size increase is around 2x less than in the presence of the Ras^{V12} mutation, but nonetheless is interesting and deserves a mention in the main text.

Response: We appreciate your insightful observation regarding the Toll pathway's role in promoting cellular overgrowth independent of Ras^{V12} , as illustrated in Figure S2C. We agree this nuance merits prominent discussion, particularly given the Toll pathway's well-characterized roles in immune regulation and embryonic development. Our findings reveal a previously unrecognized pro-proliferative capacity of Toll signaling in epithelial tissues; therefore, it needs to be emphasized in our manuscript.

We have revised the main text to emphasize these results as follows:

“Notably, clonal overexpression of *pll*, *dl*, or *Dif* significantly increased clone area compared to wild-type controls (Fig. EV2C,D), indicating that Toll pathway activation promotes

proliferation even in the absence of oncogenic *Ras*^{V12}. These findings expand our understanding of Toll signaling beyond its canonical immune functions and highlight its potential interplay with oncogenic processes in tumor-immune crosstalk.”

3) *Tl* and Hippo pathway intersection

The authors' findings run counter to a previous paper PMID: 26824654 in which the Toll pathway leads to Yki inactivation in the fat body to enable the expression of innate immunity including AMP expression. Again this could be due to distinctions in the two tissues' responses, but could the authors please comment on this in the discussion?

Response: Thank you for raising this critical point and for directing our attention to the important study (PMID: 26824654), which demonstrated that Toll pathway activation in the *fat body* leads to Yki inactivation and AMP induction. We are aware of this paper, as I was a postdoc at DJ's lab when this paper was submitted. Honestly, we were confused initially regarding this discrepancy, while our genetic data clearly demonstrated that *Ras*^{V12} synergize with Toll signaling activation to inactivate the Hippo pathway, enabling *Myc*, *CycE*, and *Diap1* upregulation (Figure 4). The possible expatiation could be tissue-specific and genetic context dependency. To further explore the underlying mechanisms, we examined the expression of Hippo pathway target genes upon clonal activation of Toll signaling in eye epithelium. As shown below, we found that clonal expression of *dl* or *Dif* under physiological conditions paradoxically inhibited the expression of Diap1 and CycE, despite the significant overgrowth phenotype (**Author response Figure 6**). Therefore, our study reveals a context-dependent Toll-Yki crosstalk and we speculate that *Ras*^{V12}-driven metabolic reprogramming may sensitize tumor cells to Toll signals, overriding Yki inhibition. In contrast, the fat bodies lack these oncogenic pressures, allowing Toll-Yki crosstalk to dominate.

Author response Figure 6. Eye-antennal discs bearing GFP-labeled WT, *dl*, or *Dif* overexpression clones were co-stained with anti-Diap1 and anti-CycE antibodies.

We have also included additional discussion in the revision:

“To further dissect the mechanisms by which the Toll pathway modulates Hippo and Yki activity, we performed clonal overexpression of *dl* or *Dif* under physiological conditions and analyzed

Hippo target gene expression. Surprisingly, while dl/Dif overexpression significantly induced tissue overgrowth, it paradoxically reduced the expression of Hippo targets Diap1 and CycE (Fig. EV4B,C). This apparent contradiction highlights context-dependent modulation of Hippo/Yki activity by Toll signaling, aligning with previous findings that Toll suppresses Yki activity in the fat body during microbial infection (Liu, Zheng et al., 2016), yet promotes Yki-dependent growth via JNK signaling in polarity-deficient cells (Katsukawa, Ohsawa et al., 2018). Specifically, we speculate that the metabolic stress induced by oncogenic Ras^{V12} may override the typical crosstalk between Toll and Yki. These data also underscore the importance of caution when extrapolating the role of Toll signaling from immune contexts to oncogenic settings. ”

Minor points:

3) Genetics

a) The authors use a well known M6^{W186*} mutation but do not show that this is the cause of the overgrowth phenotype of the Ras^{V12} neighbors by rescuing the phenotype through expression of M6. Doing the rescue is time consuming and is not key for their story since this paper does not focus on the mechanisms by which the "M6" mutant tissue is signaling to the Ras^{V12} neighbors to lead to the Pvf1 expression which recruits hemocytes and starts the Tl signaling. I don't ask for the experiment, but just for language stating that the evidence supports but does not prove that loss of a tricellular junction protein is the initiating cause.

Response: We combined Gal4-UAS system and Q-MARCM system to specifically overexpress M6 in M6 mutant clones surrounding QRas^{V12} tumors. As shown below, we observed a dramatic rescue on the overgrowth of neighboring GFP-labeled QRas^{V12} tumors (**Author response Figure 7**), confirming that the mutation of M6 is indispensable for the tumorous growth of surrounding clones.

Author response Figure 7. Eye-antennal discs bearing clones with indicated genotypes.

Next, to further explore whether M6 has a unique role in tumor heterogeneity or loss of tricellular junction (TCJ) protein could generally affect tumor heterogeneity, we utilized the dual system mentioned above to knockdown two additional TCJ proteins, including Gliotactin (Gli) and Anakonda (aka/bark). As shown below, RNAi-mediated knockdown of *Gli* or *aka* around QRas^{V12} clones (QRas^{V12}//*Gli*^{RNAi}, QRas^{V12}//*aka*^{RNAi}) phenocopied M6 loss, promoting overgrowth of surrounding QRas^{V12} clones (**Author response Figure 8**). These results demonstrate that TCJ dysfunction, rather than M6-specific depletion, is the critical determinant of oncogenic cooperation with Ras^{V12}. It would be very interesting to further explore the underlying

mechanisms by which TCJ disruption facilitates the growth of surrounding Ras tumors.

Author response Figure 8. Eye-antennal discs bearing clones with indicated genotypes.

We have included these findings in our revised manuscript:

“Strikingly, using the Gal4-UAS/Q-MARCM dual system to knock down two other TCJ proteins, *Gli* and *aka*, in cells neighboring *Ras^{V12}* clones (*QRas^{V12}//Gli^{RNAi}*, *QRas^{V12}//aka^{RNAi}*) phenocopied the *M6* loss-of-function phenotype, driving overgrowth of adjacent *QRas^{V12}* clones (Fig. 1G,H).”

b) While the authors use two RNAis to knockdown *dl*, *Dif*, and *Ets21c* in some of their experiments they don't do it in all of them. They do not use two *Tl* RNAis, nor two *Tub* RNAis, nor two *Pl* RNAis. The fact that they are knocking down multiple members of the same pathway and show the same phenotype, and show the opposite phenotype upon overexpression of members of that pathway convinces me not to request the usual level of rigor for RNAi use for each gene that is standard in the field (ie two independent RNAis per gene). However, it would have been cleaner data if they had. In particular the absence of an effect from the *Pvf2* RNAi is not convincing if only one.

Response: We appreciate your thoughtful critique and for highlighting the importance of methodological rigor in RNAi experiments, we also appreciate your trust in our data and conclusion. We do have more than one RNAi strain per gene in hand, however, due to the genetic complexity of our tumor model, it is very difficult to recombine those RNAi strains on the third chromosome, especially those with *y* genetic background.

To further verify whether the knockdown of *Pvfs* has an effect on *Ras^{V12}//M6^{-/-}* tumor overgrowth, we utilized two additional RNAi targeting *Pvf1* and *Pvf2*, respectively. We observed significant knockdown efficiency for all four RNAi strains under the control of *GMR-Gal4*, as demonstrated by qRT-PCR analysis using dissected adult heads (Author response Figure 9). Moreover, we noticed that newly used *Pvf2* RNAi strain failed to suppress *Ras^{V12}//M6^{-/-}*-induced tumor overgrowth, whereas the independent *Pvf1* RNAi consistently reduced tumor size (Author response Figure 9).

Author response Figure 9. The qRT-PCR validation of the corresponding RNAi strains (A,B). Eye-antennal discs bearing GFP⁺ clones with indicated genotypes. (C,D).

Textual issue:

3I y axis, I think they mean ratio not ration

Response: We have corrected this typo in the revised manuscript.

5G is this showing the size of the sum of all the clusters for an eye disc or the size of each individual cluster?

Response: Figure 5G shows the total area of all hemocyte clusters within the $Ras^{V12}/M6^{-/-}$ eye disc, not the size of individual clusters. Each data point represents the cumulative clustered area per disc. We have revised the legend and methods to explicitly state this.

Referee #2:

In this study, Zhao, Guo and collaborators use a modified version of the MARCM technique to generate mosaic eye disc with neighbouring clones either expressing a constitutively active form of the oncogene Ras (Ras^{V12}), or mutant for the tricellular component M6. They show that Ras^{V12} clones grow massively only when confronted with M6 mutant clones ($Ras^{V12}/M6$) and not when confronted to wild-type tissues (Ras^{V12}). Using scRNA-Seq of these $Ras^{V12}/M6$ tumours they identify the transcriptomes of their constitutive cell types identifying different clusters for the eye disc epithelial cells, glial cells, and attached haemocytes. Mechanistically, they show that the Toll/tube/pelle/dorsal/Dif pathway is activated and required in the Ras^{V12} overgrowing tissues. They propose that the Toll pathway promotes Yki activity thus promoting growth. Using dual expression systems and transplantation assays, they show that spz producing haemocytes are required for $Ras^{V12}/M6$ tissue growth and dl activation. Finally, they identify Pvf1 secreted by the Ras^{V12} cells as the activator of haemocytes.

This is a huge amount of work, and the majority of experiments supports the claims which are original. Indeed, the main originality lies in the difference of behaviour between Ras^{V12} and $Ras^{V12}/M6$ mosaic tissues in terms of growth potential. The second original finding is that the Toll pathway is required for tumour growth, while it had been suggested to promote the death of dl or scrib mutant tissues. There are however some points that need to be clarified.

Response: Thank you for recognizing the originality and robustness of our work. We believe that the revisions we have made in response to your comments have strengthened the manuscript, and we hope that it now meets your expectations.

Main comments.

1. The authors need to state exactly at what age the different images and measurements of the discs have been performed and ensure as much as possible that they compare discs of similar age. Indeed, one of the obvious observation from Figure 1C is that $Ras^{V12}/M6$ animals have an extended larval life (even though most will pupate Fig 1F), which would translate into increased time for proliferation. Strict timing of the different conditions is critical, and should be indicated.

Response: Thank you for raising this critical point regarding developmental staging and age consistency. To address this, we include additional description in the Methods section of the revised manuscript and have clarified the timing of the measurements as follows:

For *Drosophila* larvae carrying Ras^{V12}/WT benign tumors, they typically reach the late third-instar larval stage by day 6 post-oviposition at 25°C and begin extensive pupation thereafter. As a result, dissections and clone area quantifications are uniformly performed on day 6. In

contrast, $Ras^{V12}/M6^{-/-}$ intercellular heterogeneous tumors exhibit a slower proliferation rate compared to the highly malignant intracellular $Ras^{V12}/scrib^{-/-}$ tumors. Since $Ras^{V12}/M6^{-/-}$ tumors do not undergo massive overgrowth, interventions, such as bsk^{DN} expression, applied to $Ras^{V12}/M6^{-/-}$ tumors on day 6 yield inconclusive results regarding their efficacy in inhibiting tumor proliferation. This is because inhibitory effects are less pronounced at this stage. To address this, we allow the larvae to develop until the very late third-instar stage, performing dissections only when the pupariation rate exceeds 50%. At this stage, we can clearly determine whether RNAi or mutational interventions impact $Ras^{V12}/M6^{-/-}$ tumor proliferation. The delayed pupation rate, commonly regarded as a hallmark of tumor malignancy in *Drosophila*, further supports this approach. Although some experimental groups (e.g., $Ras^{V12}/M6^{-/-}$ tumors with bsk^{DN} overexpression) may be dissected on different days compared to controls ($Ras^{V12}/M6^{-/-}$ tumors), all dissections are conducted at the developmental limit specific to each genotype. This ensures more reliable and reproducible conclusions regarding the inhibition of $Ras^{V12}/M6^{-/-}$ tumor proliferation.

In addition, previous studies have demonstrated that growth control of *Drosophila* imaginal discs is primarily intrinsic (PMID: 6393189), as artificially delaying pupation generally does not lead to disc overgrowth (PMID: 17947427). This indicates that the size of imaginal discs prior to pupation can reflect the proliferative capacity of tumors.

2. Many IF panels have a high magnification which makes interpretation complicated. This is especially true for Fig 3H, 4A, 5D, S3E. In these figures, the point is to compare Ras^{V12} GFP positive tissues to GFP negative tissues (M6 mutant or WT), and compare the levels of staining. At the magnification shown, it is impossible to find enough GFP negative tissues to compare.

Response: To improve clarity, we have replaced some of the original panels with new images. In the revised figures, we have included sufficient GFP⁻ tissues to make proper conclusion (**Author response Figure 10**). We replaced Diap1 staining in original Fig. 3H with a lower-magnification image to clearly show GFP⁺ and GFP⁻ clones in $Ras^{V12}/M6^{-/-}$ tumors. We also adjusted magnification to allow direct comparison of staining intensity between GFP⁺ and GFP⁻ regions in original Figs. 4A, 5D, and S3E. We believe that these revisions ensure that readers can reliably compare staining patterns across genotypes.

Author response Figure 10. Eye-antennal discs bearing GFP-labeled clones with indicated genotypes were stained with corresponding antibodies.

3. The Hippo pathway down-regulation is hard to interpret. Indeed, removing *yki* or overexpressing *wt5* would severely reduce the growth of any imaginal disc tissue. The fact that they are epistatic to *Ras^{V12}//M6* and to Toll pathway is thus not that strong of an argument. We are thus left with the *yki* targets, many of which are not specific to the Hippo pathway (*myc*, *Diap1*, *CycE*, *ban*, *wg*, *ff* are also targets of Notch for instance). The pictures shown for *DIAP1* are not convincing given point 2. *Ex-LacZ* is more specific. A more detailed analysis should be performed at earlier time points (smaller clones) to better document *Ex-LacZ* activation.

Response: Thank you for highlighting the importance of rigorous validation regarding Yki activation. While we appreciate your perspective, we respectfully disagree with your assertion that examining multiple canonical Hippo pathway target genes, such as *myc*, *Diap1*, *CycE*, *ban*, *wg*, and *ff*, is insufficient to conclude that Hippo pathway is inhibited. Numerous studies conducted by researchers worldwide have consistently confirmed that the transcriptional upregulation of these genes serves as a hallmark of Hippo signaling inhibition and acts as targets of Yki (PMID: 18258486, 23725764, 34862372, 28143945, 33042412, 30679505, 20951343, 21215937, 29695716, 19762509, 31553691, 33298454, 32888651, 25027438, 26474042, 39106181, 26324895). That said, we wholeheartedly agree that it is crucial to remain open to alternative explanations and to approach scientific conclusions with a healthy degree of skepticism. Nevertheless, we believe the data presented in our manuscript provide robust evidence supporting the conclusion that *Ras^{V12}//M6^{-/-}* tumors inactivated Hippo signaling and upregulate multiple target genes of this pathway.

Following your suggestion, to further strengthen our findings, we conducted time-course

analyses of *ex-lacZ*, a direct Yki target, during early stages of tumor development. At 6 days, *Ras^{V12}* clones exhibited mild upregulation of *ex-lacZ* compared to the surrounding wild-type clones, whereas *Ras^{V12}//M6^{-/-}* GFP⁺ clones demonstrated significantly higher *ex-lacZ* levels, indicative of increased Yki activity. By 9 days, at relatively late tumor stages, GFP⁺ *Ras^{V12}//M6^{-/-}* clones had nearly overtaken the entire disc, with *ex-lacZ* expression further elevated in these regions, consistent with enhanced Yki activity (**Author response Figure 11**).

Author response Figure 11. Eye-antennal discs bearing GFP-labeled clones were stained with anti- β -galactosidase antibody to visualize the *ex-lacZ* reporter.

The authors should also discuss models as to how Toll/pll/dl/Dif affects Hippo and Yki activity.

Response: We appreciate the opportunity to clarify the relationship between Toll signaling and Hippo pathway regulation. To explore this interaction, we performed clonal overexpression of *dl* or *Dif* under physiological conditions and analyzed Hippo target gene expression. Intriguingly, while *dl/Dif* overexpression induced significant tissue overgrowth, we paradoxically observed reduced expression of Hippo targets *Diap1* and *CycE* (**Author response Figure 12A-B**). On the contrary, with the presence of oncogenic *Ras*, Toll activation strongly upregulated Hippo target gene expression (**Author response Figure 12C**). This apparent contradiction suggests context-dependent modulation of Hippo/Yki activity by Toll signaling, consistent with previous findings that Toll activation in polarity-deficient cells promotes Yki activity via JNK signaling (PMID: 29804808).

Author response Figure 12. Eye-antennal discs bearing GFP-labeled clones with indicated genotypes were stained with anti-Diap1, anti-CycE, and anti-β-galactosidase antibodies.

Further supporting the role of JNK signaling, KEGG pathway analysis of GFP⁺ *Ras*^{V12}//*M6*^{-/-} tumor cells revealed enrichment of MAPK signaling components (**Author response Figure 13**). Functional validation confirmed JNK activation through significant *Mmp1* upregulation (**Author response Figure 14A-B**). Notably, genetic inhibition of JNK signaling through the expression of a dominant-negative form of *bsk* (*bsk*^{DN}) suppressed *Ras*^{V12}//*M6*^{-/-}-driven tumorigenesis and abolished upregulation of *Mmp1* expression and *ex* transcription (**Author Response Figure 14A-D**). These results indicate that *Ras*^{V12}//*M6*^{-/-} tumors activate JNK signaling to regulate Hippo pathway activity.

Author response Figure 13. KEGG enrichment of DEGs from scRNA-seq in *Ras*^{V12}//*M6*^{-/-} specific GFP⁺ cell (A) and KEGG enrichment of all DEGs from bulk RNA-seq comparing *Ras*^{V12}//*M6*^{-/-} and *Ras*^{V12}//*WT* (B).

Author response Figure 14. Eye-antennal discs bearing GFP-labeled clones were stained with anti-MMP1, and anti-β-galactosidase antibodies.

Next, we explored whether Toll pathway activation is sufficient to activate JNK signaling. Under physiological conditions, ectopic expression of *dl* or *Dif* activates JNK signaling (*puc-lacZ* and *Mmp1* upregulation), along with slight F-actin accumulation (**Author response Figure 15**). Under oncogenic stress, Toll pathway activation synergized with *Ras*^{V12}, resulting in robust JNK signaling activation and strong F-actin polymerization (**Author response Figure 15**). Given that F-abnormal actin accumulation is known to activate Yki/YAP (PMID: 21556047, 21525075, 24648494), these findings collectively support a model where *Ras*^{V12}//*M6*^{-/-} tumors act through the Toll-JNK-Hippo signaling cascade to promote tumorigenesis, corroborating that JNK

suppresses Yki activity in *Ras*-deficient contexts but enhances it via F-actin accumulation when *Ras* is present (PMID: 25967126).

Author response Figure 15. Eye-antennal discs bearing GFP-labeled clones were stained with anti- β -galactosidase, anti-MMP1 antibodies, and Alexa Fluor® 555 Phalloidin.

4. The description of the scRNA-Seq analysis is hard to follow, in particular why the different re-clusterings and fusions are made. I am not sure I understood the >65% occupancy and the way it is documented in Fig. 3D. Which clusters are GFP? Do they correspond mainly to *Ras*^{V12} tumoral specific tissues?

Response: We apologize for any confusion caused by our original manuscript. Below, we provide a detailed explanation of the process used to isolate GFP cells from *Ras*^{V12}//*M6*^{-/-} tumors. Firstly, glia cells and hemocytes were removed from the dataset, and epithelial cells were retained for re-clustering. Cell-type annotations were performed based on gene markers reported in previous studies (PMID: 32431014, 30479347). Using the *subset* function, all epithelial cells were extracted, and the epithelial cells of *Ras*^{V12}//*M6*^{-/-} tumors were grouped into 19 subclusters. The proportions of GFP-labeled and non-GFP-labeled cells within these 19 subclusters were determined according to the expression levels of the exogenously inserted GFP gene sequence (**Author response Figure 16A**). However, due to the limitations in sc-seq, such as the median number of detectable genes and gene alignment principles, not all GFP-expressing cells can be reliably detected. Consequently, relying solely on GFP expression for determining cellular labeling status may lead to misclassification: GFP+ cells might be incorrectly categorized as non-GFP, and vice versa.

To address this, we utilized the transcriptomic similarity within clusters to determine the predominant cell type in each subcluster, distinguishing whether they were mainly GFP-labeled tumor cells (*Ras*^{V12}) or non-GFP-labeled cells (*M6*^{-/-}). We applied a 65% occupancy threshold to define predominant cell types in the *Ras*^{V12}//*M6*^{-/-} scRNA-seq dataset (**Author response Figure 16B**), a criterion adapted from a prior study (PMID: 32451460, 36321803)).

Subsequently, GFP-expressing cells were isolated from the GFP-predominant subclusters (cluster 2, 4, 7, 8, 11, 12, 16, and 17) of *Ras*^{V12}//*M6*^{-/-} tumors and integrated them with *WT* cells for secondary filtration, aiming to identify transcriptomic differences between those *Ras*^{V12}//*M6*^{-/-} unique GFP+ cells (*Ras*^{V12} cells) and *WT* cells (**Author response Figure 16C**). Following a strategy previously described in fly tumor models (PMID: 36321803), we further isolated GFP+ cells with significantly altered transcripts in *Ras*^{V12}//*M6*^{-/-} tumors (compared to *WT* cells) based on sample origin (*WT* or *Ras*^{V12}//*M6*^{-/-} tumors), including clusters 5, 11, 12, 13, and 14 (**Author**

response **Figure 16D-G**). Differentially expressed genes were then identified by comparing $Ras^{V12}/M6^{-/-}$ unique GFP+ cells to WT cells. Functional annotations for each cluster, along with the top-ranking KEGG terms, are presented in the heatmap (**Author response Figure 16H**).

This strategy offers two key advantages: First, when the proportion of cells within a cluster exceeds our predefined threshold, it enables the identification of the predominant cell type. Extracting these predominant cells reduces the likelihood of misclassification (the predominant subclusters of GFP+ and GFP- are distinguished by a 65% occupancy rate GFP). Second, subpopulations with proportions below the threshold suggest minimal transcriptomic differences between the two cell types and were therefore excluded from further analysis to focus on cells that are more unique to the $Ras^{V12}/M6^{-/-}$ dataset.

Author response Figure 16. Step-by-step illustration of the identifications of $M6^{-/-}/Ras^{V12}$ unique GFP cells in epithelium.

5. The *spz* expressing haemocytes appear to be concentrated on *M6* mutant tissues which will be eliminated (Fig. 5A&H). Could authors show *Dcp1* staining? Is there increased cell death at *M6* tissues (like in WT discs)? Is there also increased haemocytes expressing *spz* attracted to *M6* mutant cells in WT discs (Fig S1)? This is important, since similar haemocytes recruitment to *M6* mutant clones would thus indicate that one signal attracting the haemocytes are these dying cells, which would change the cascade of events in the model.

Response: Thank you for raising these critical questions. While clonal mutation of *M6* alone

induces apoptosis (as shown in **Author response Figure 17**), this does not result in increased hemocyte recruitment compared to wild-type (WT) tissues. This suggests that apoptosis alone is insufficient to drive enhanced hemocyte recruitment.

Author response Figure 17. Immunofluorescent images of Dcp-1 and NimC1 staining in eye-antennal discs bearing *WT* and *Ras^{V12}//M6^{-/-}* clones.

To directly address the role of cell death in *M6*-mutant tissues, we performed Dcp-1 staining in *Ras^{V12}//M6^{-/-}* tumors. Apoptosis was predominantly observed in GFP⁺ *Ras^{V12}* tumor clones rather than in adjacent *M6^{-/-}* clones (**Author response Figure 18**).

Author response Figure 18. Immunofluorescent images of Dcp-1 staining in eye-antennal discs bearing *WT* and *Ras^{V12}//M6^{-/-}* clones. White arrows indicate hemocytes.

To further corroborate these results, we analyzed apoptosis using the *rpr-lacZ* reporter (*reaper*, a pro-apoptotic gene; PMID: 10675328, 12021767). Despite the overgrowth of *Ras^{V12}* tumor clones in *Ras^{V12}//M6^{-/-}* tissues, *rpr* transcription was robustly induced in GFP⁺ tumor clones (**Author Response Figure 19**), which is consistent with the large amount of apoptosis induction we found in GFP-positive *Ras^{V12}* clones within *Ras^{V12}//M6^{-/-}* tumor.

Author response Figure 19. Immunofluorescent images of *rpr-lacZ* staining in eye-antennal discs bearing *WT* and *Ras^{V12}//M6⁻* clones.

These data collectively demonstrate that while *M6* mutation alone triggers apoptosis, this apoptotic signal does not drive significant hemocyte recruitment. Instead, *M6⁻* likely functions as a signaling amplifier, enhancing both the proliferative capacity of adjacent *Ras^{V12}*-driven tumor clones and their ability to recruit hemocytes. This refined model better explains the tumor-hemocyte interplay observed in our system. However, a limitation of our study is the lack of mechanistic exploration into the relationship between apoptosis and ROS or Pvf1 secretion within *Ras^{V12}//M6⁻* tumors. While extensive apoptosis was observed in these tumors, the interplay between ROS-mediated oxidative stress and Pvf1-driven hemocyte recruitment remains uncharacterized.

6. Where is *Toll* expressed? Is it more abundant on *Ras^{V12}* cells priming them to respond to spz? Could that be inferred for the scRNA-Seq or from IFs?

Response: We analyzed *Toll* expression at both transcriptional and protein levels. In our scRNA-seq data, we observed slightly increased *Toll* mRNA expression in *Ras^{V12}* clones of *Ras^{V12}//M6⁻* tumors (**Author Response Figure 20A**). However, paradoxically, *Toll* protein levels, as measured by an endogenous *Toll*-GFP reporter *in vivo*, were significantly upregulated in the *Ras^{V12}* clones within *Ras^{V12}//M6⁻* tumors (**Author Response Figure 20B,C**). This discrepancy between transcriptional and post-transcriptional regulation suggests that *Toll* protein abundance in *Ras^{V12}* cells may not depend on transcriptional activation but could instead involve mechanisms like translational control or protein stabilization.

Author response Figure 20. UMAP of Toll expression in $Ras^{V12}/M6^{-/-}$ GFP⁺ tumor cells (A) Toll-GFP expression in eye-antennal discs with Ras^{V12}/WT and $Ras^{V12}/M6^{-/-}$ clones (B), and quantification of Toll-GFP intensity (C).

7. The authors never discuss what determines the different responses between a Ras^{V12}/WT and a $Ras^{V12}/M6$ mosaic tissue? This is a major weakness, and authors should at least discuss this.

Response: Thank you for highlighting this critical gap in our manuscript. The distinct outcomes observed in Ras^{V12}/WT versus $Ras^{V12}/M6^{-/-}$ mosaic tissues led us to investigate two non-mutually exclusive mechanisms: (1) apoptosis-induced compensatory proliferation and (2) tricellular junction (TCJ) dysfunction. Since $M6$ depletion induces apoptosis, we tested whether localized apoptotic stress could drive tumorigenic cooperation. Using a Gal4-UAS/Q-MARCM dual system, we induced targeted overexpression of the pro-apoptotic gene *Hid* (*Head involution defective*) specifically in cells surrounding $QRas^{V12}$ clones. While mild compensatory proliferation occurred in adjacent $QRas^{V12}$ clones (**Author response Figure 21**), the overall eye-disc size remained unchanged, indicating that heterogeneous apoptosis induction does not robustly promote oncogenesis.

Given $M6$'s role as a TCJ component, we extended our analysis to other TCJ proteins: Gliotactin (*Gli*) and Anakonda (*aka/bark*). Strikingly, RNAi knockdown of *Gli* or *aka* in cells surrounding $QRas^{V12}$ clones ($QRas^{V12}/Gli^{RNAi}$, $QRas^{V12}/aka^{RNAi}$) phenocopied $M6$ loss of function phenotype, driving overgrowth of surrounding $QRas^{V12}$ clones (**Author response Figure 21**). This demonstrates that TCJ architectural disruption, not $M6$ -specific depletion, is the unifying factor enabling oncogenic cooperation with Ras^{V12} .

Author response Figure 21. GFP-labeled clones in eye-antennal discs bearing indicated genotypes.

These findings reposition TCJ integrity as a gatekeeper of tumor-microenvironment crosstalk. While our work identifies TCJ dysfunction as a key driver of Ras^{V12} -driven tumorigenesis, we fully agree that elucidating the mechanistic link between TCJ breakdown and Ras^{V12} hyperactivation (e.g., via altered signaling, mechanical stress, or metabolic reprogramming)

represents an essential next step. Again, we appreciate your insightful comments, which have directed us toward these novel findings, and we are actively pursuing these questions in follow-up studies.

8. *Ets21C* is part of the on comodule that was shown required for eye discs *Ras^{V12}* clone growth. Its removal in restores the morphology of the discs, with increased *M6* mutant tissues and separated GFP clones. This is similar to what is seen for *tube* or *dl* RNAi. The phenotypes thus looks thus dependent on some kind of cell competition. Could the author comment on this in terms of model (see point 7)? The removal of *Pvf1* does not make discs of similar shape as the removal of *tube* or *dl*. They are smaller, but the *M6* tissue is almost completely eliminated. Is that specific to this particular disc (Fig 7H), or could this reflect another role of *Pvf1*?

Response: Thank you for these insightful comments. *Ets21C* is established as a downstream effector of JNK/AP-1 signaling across multiple contexts (PMID: 35820420, 31167145, 27713480). Both JNK inhibition and *Ets21C* knockdown similarly suppress *Ras^{V12}//M6^{-/-}* tumor overgrowth (Author response Figure 22A), suggesting a shared mechanistic hierarchy. We think the phenotypic differences observed upon *Pvf1* knockdown (vs. *tube/dl* RNAi or JNK/*Ets21C* inhibition) likely arise from two key factors: 1) *Pvf1* is a transcriptional target of *Ets21C* (PMID: 35820420). Consistent with this, *Ets21C* inhibition abolishes *Ras^{V12}//M6^{-/-}* tumor-induced *Pvf1* upregulation (Author Response Figure 22B). Since *Pvf1* represents only one downstream effector of *Ets21C*/JNK signaling, its inhibition would naturally produce a weaker phenotype compared to blocking the upstream regulator (*Ets21C*) or the broader JNK pathway. 2) qRT-PCR revealed ~50% residual *Pvf1* expression after RNAi in *Drosophila* adult heads (Author Response Figure 22C). This incomplete knockdown may explain the milder suppression of *Ras^{V12}*-driven overgrowth.

Author response Figure 22 (A) GFP-labeled clones in eye-antennal discs bearing indicated genotypes. (B-C) qRT-PCR analysis of *Pvf1* transcription in indicated tissues/tumor samples.

We agree that cell competition could contribute to the observed phenotypes. JNK-inhibited *Ras^{V12}* clones exhibit reduced growth compared to *Ras^{V12}//M6^{-/-}* tumors, potentially diminishing their ability to compete with surrounding *M6* mutant tissue. While dissecting cell competition's role in *M6*-mediated tumor progression is intriguing, this investigation extends beyond the current study's scope.

Minor comments.

Authors should show the efficacy of the RNAi lines used, in particular those for which they did not observe modifications

Response: Thank you for your suggestion regarding the validation of knockdown efficiency, especially for RNAi lines yielding negative results. Accurate conclusions depend on robust validation, and we have addressed this concern by assessing the knockdown efficiency of the RNAi strains. Specifically, we crossed GMR-Gal4 with *Pvf1* and *Pvf2* RNAi lines and performed qRT-PCR analysis on dissected adult heads. The results showed that all *Pvf1* and *Pvf2* RNAi lines reduced their respective mRNA levels by approximately 50%. Further, in the context of *Ras^{V12}//M6^{-/-}* tumors, two independent *Pvf2* RNAi strains did not suppress tumor growth. However, two independent *Pvf1* RNAi strains significantly reduced tumor size, as illustrated in **Author Response Figure 23**. These findings highlight the functional distinction between *Pvf1* and *Pvf2* in this experimental setting.

Author response Figure 23. The qRT-PCR validation of the corresponding RNAi strains (A,B). Eye-antennal discs bearing GFP⁺ clones with indicated genotypes and quantification (C,D).

Authors should not use the term of tumour heterogeneity and subclones. Indeed the genetic trick used here generates clones with an oncogene and then adjacent clones with M6 mutations. These are thus two independent alterations that will compete. This is not tumour heterogeneity, where some cells within the tumour mass acquire different mutations. At best it is two fighting or cooperating tumours, but not tumour heterogeneity. Furthermore, the sentence in the discussion lines 328-329 "Our data here reveals a complex intercellular crosstalk both within epithelial cells bearing distinct oncogenic mutations and between tumors and hemocytes" is misleading. Indeed, M6 mutation is not oncogenic per se, since M6 mutant cells are normally eliminated.

Response: We sincerely appreciate the thoughtful critique and opportunity to clarify our perspective. Below, we address the concerns raised, with the aim of fostering a constructive dialogue on terminology and mechanistic interpretation:

1. On Tumor Heterogeneity and Subclones:

The term "tumor heterogeneity" encompasses genetic, phenotypic, and microenvironmental diversity within a tumor, including clonal subpopulations arising through distinct mechanisms. While the reviewer emphasizes that the *M6* and *Ras* clones in our model originate via independent genetic events (rather than sequential mutations within the same lineage), we propose that their coexistence within a shared tumor microenvironment still constitutes intra-tumor heterogeneity. This aligns with established frameworks where spatial and functional interactions between distinct clones, even those arising independently, contribute to tumor evolution (e.g., "clonal cooperation" in Cleary et al., 2014, *Nature*; Marusyk et al., 2014, *Nature*). We acknowledge that the clonal dynamics here differ from classical subclonal evolution driven by branching mutations; however, the competitive and cooperative interactions between these genetically distinct populations remain a critical facet of heterogeneity.

2. Oncogenic Potential of *M6* Mutations:

The reviewer rightly notes that *M6* mutant clones are eliminated in the presence of WT cells, consistent with cell competition. However, their intrinsic oncogenic potential is context-dependent. Analogous to *scrib* mutants, which are apoptotically eliminated in normal tissue but form aggressive tumors when competition is absent (*Bilder et al., 2002, Science; Igaki et al., 2009, Dev Cell*), *M6* mutant clones exhibit latent tumorigenic capacity. Specifically, *M6* homozygous mutated larvae drive similar neoplastic growth (**Author response Figure 24**, *Dunn et al., 2018, PNAS*). Thus, labeling *M6* as “non-oncogenic” oversimplifies their biology; instead, their elimination in competitive contexts reflects tumor-suppressive mechanisms rather than a lack of oncogenic driver potential.

Author response Figure 24. Both *scrib* (left) and *M6* (right) mutant larvae display an extended larval life span and overgrow but fail to pupate. Images from Papagiannouli & Mechler, book chapter, <http://dx.doi.org/10.5772/55686> and *Dunn et al., 2018, PNAS*.

3. Intercellular Crosstalk and Tumor-Hemocyte Interactions:

The phrase "complex intercellular crosstalk" refers to our observation that *M6* clones (prior to elimination) promote Ras clone proliferation, while hemocytes simultaneously exert tumor-promoting effects. We agree that the wording in lines 328-329 could be refined to avoid ambiguity and have revised it to:

*"Our data reveal spatially resolved crosstalk between epithelial clones with distinct driver mutations (Ras^{V12} and *M6*) and between tumors and hemocytes, highlighting complex signaling networks that shape tumor progression."*

Collectively, while semantic distinctions are important, our central thesis, that clonal interactions between genetically distinct populations drive tumor progression, aligns with broader definitions of heterogeneity. Thank you again for prompting deeper scrutiny of these concepts and having incorporated changes to improve clarity.

Discussion is too short. Authors should discuss in greater detail

- The specificity of Ras^{V12} //WT vs Ras^{V12} //*M6*
- How Toll could be activated specifically in Ras^{V12} cells?
- How Toll/dl impinge on Hippo?
- How the Ras^{V12} //*M6* compared to the undead cells model where Toll / Toll9 has been shown to initiate caspase activation via *hid/rpr* (but later blocked) and to attract haemocytes which through JNK signalling promote the production of pro-proliferative signals (*wg, upd...*).

Response: Thank you for bringing up these insightful questions. In the revised manuscript, we have updated the discussion section as follows (revisions are highlighted in red font in the revision):

“Tumor heterogeneity is well-established feature across diverse cancer types, with profound implications for tumorigenesis and therapeutic resistance (*Gavish et al., 2023, Gonzalez-Silva et*

al., 2020, Vitale et al., 2021). While intercellular communication between tumors and their microenvironment is recognized as a critical driver of malignancy, the *in vivo* mechanisms governing these interactions during tumor progression remain poorly understood, largely due to a lack of tractable experimental models. To address this gap, we developed a *Drosophila* tumor heterogeneity model by selectively disrupting the tricellular junction protein *M6* in cells adjacent to benign *Ras^{V12}* tumors. Our findings revealed that loss of *M6* promotes tumor malignancy in neighboring *Ras^{V12}* tumors by inducing secretion of Pvf1. Pvf1 acts as a chemotactic factor, recruiting hemocytes via the Pvr receptor. These hemocytes, in turn, activate a paracrine Spz-Toll signaling axis within *Ras^{V12}* clones, establishing a complex interplay between the tumor and its microenvironment. Eventually, the activation of the Toll pathway synergizes with oncogenic *Ras* to induce tumor progression through a JNK-Hippo signaling cascade (Fig. EV8G). This intricate signaling network highlights the cooperative interactions between clones bearing distinct mutations and the immune microenvironment, providing mechanistic insights into how tumor heterogeneity and intercellular communication contribute to malignancy. Our study underscores the importance of understanding tumor-microenvironment dynamics and presents the *Drosophila* model as a powerful platform for dissecting these processes.

Dynamic Regulation of Hemocyte-Tissue Adhesion

Previous studies have demonstrated that ROS induced by apoptotic caspase promotes hemocytes recruitment in *Drosophila* eye epithelium, facilitating malignant tumor progression (Perez et al., 2017). Similarly, Toll activation in “undead cells” recruits hemocytes through caspase-dependent ROS signaling, driving JNK-mediated proliferation (Shields, Amcheslavsky et al., 2022). In this study, we reveal that *M6* mutant clones alone undergoing apoptosis fail to recruit hemocytes. These findings align with recent evidence showing that apoptosis induction is dispensable for hemocyte recruitment during classical cell competition (Zhu et al., 2025). This suggests that hemocyte recruitment to apoptotic sites in *Drosophila* is context-dependent and may require simultaneous ROS generation, or cytokine secretion such as Pvf1. Crucially, the functional outcome of hemocyte recruitment—whether promoting cell death or proliferation—appears to depend on the specific cellular context. For instance, when only the polarity gene *scrib* is mutated, hemocyte-secreted Eiger facilitates the clearance of *scrib*-mutant cells (Parisi et al., 2014). In contrast, within *Ras/scrib^{-/-}* tumors, activation of the *Ras* oncogene can subvert this hemocyte-mediated clearance process, instead hijacking it instead to promote tumor proliferation (Cordero et al., 2010, Perez et al., 2017). Furthermore, hemocytes can detect cell competition within tissues and actively participate by promoting the competitive interactions between “winner” and “loser” cells (Zhu et al., 2025). In our *Ras^{V12}//M6^{-/-}* tumor model, hemocytes again demonstrate specificity by homing to tumor cells, guided by Pvf1 secretion, and contributing to tumor progression.

These collective findings indicate that hemocytes possess a sophisticated ability to precisely sense genetic heterogeneity within tissues. Based on this discrimination, they can activate specific downstream signaling pathways that ultimately modulate the outcome of cellular interactions, promoting the clearance or proliferation of specific genotypic clones. Interestingly, both the “undead cell” model and the tumor heterogeneity model described here rely on Toll signaling to recruit hemocytes and activate JNK pathways for tumor progression. In the undead cell model, Toll/Toll-9 activation by *hid/rpr*-initiated caspase activity (which is subsequently inhibited),

leading to hemocyte infiltration. In contrast, our model uniquely integrates Pvf1/Pvr-mediated adhesion to reprogram the tumor microenvironment.

Collectively, these findings underscore the complexity of hemocyte-mediated immune surveillance in *Drosophila*. However, a limitation of our study is the lack of mechanistic exploration into the relationship between apoptosis and ROS or Pvf1 secretion within *Ras^{V12}//M6^{-/-}* tumors. While extensive apoptosis was observed in these tumors, the interplay between ROS-mediated oxidative stress and Pvf1-driven hemocyte recruitment remains uncharacterized. It is also worth noting that bulk RNA-seq data inherently include contributions from infiltrating hemocytes, which play a role in tumor microenvironments by influencing the local transcriptome. These cells are recognized as a potential source of immune-related transcriptional signals. To address this, we performed scRNA-seq analyses, which not only validated our bulk findings (e.g., Pvf1 differential expression) but also revealed previously unappreciated cellular heterogeneity within tumor-associated hemocytes. Hemocytes demonstrated remarkable functional plasticity, integrating spatial gradients, metabolic cues, and genetic alterations to execute context-dependent responses, participating in the competition and cooperation among different clones. Future investigations should focus on elucidating how tumor heterogeneity and dynamic microenvironmental signals regulate hemocyte functional states, particularly their dual roles in both anti-tumor defense and immunosuppressive microenvironment formation.

Pvf1/Pvr Signaling as a Multifunctional Axis in Systemic Dysregulation

Pvf1/Pvr signaling has emerged as a critical mediator of tumor-host organ interactions, playing a pivotal role in systemic dysregulation. Recent studies in *Drosophila* have demonstrated that Pvf1, secreted by intestinal tumors, can activate the Pvr receptor in distant organs, including the Malpighian tubules and AKH-producing cells (APCs). This activation leads to pathological outcomes such as renal dysfunction (Xu, Liu et al., 2024) and systemic host wasting (Ding, Li et al., 2025). Despite these findings, the mechanisms by which Pvf1 selectively triggers distinct downstream Pvr signaling pathways, such as JNK and ERK, in different target organs remain incompletely understood. For instance, Pvr activation in the Malpighian tubules primarily regulates uric acid metabolism through the JNK/Jra signaling axis (Xu et al., 2024). In contrast, Pvr activation in APCs appears to drive AKH release via mechanisms involving extracellular matrix (ECM) remodeling mediated by Mmp2 and enhanced innervation by excitatory cholinergic neurons (Ding et al., 2025). These distinct signaling outcomes emphasize the complexity and organ-specificity of Pvf1/Pvr interactions. Adding to this complexity, we discovered that Pvf1 secreted by *Ras^{V12}//M6^{-/-}* tumors facilitate the recruitment and the activation of hemocytes. Activated hemocytes, in turn, upregulate the Toll ligand Spz via a mechanism that remains unidentified. Further investigation is needed to clarify the spatiotemporal dynamics of Pvf1-Pvr complexes and their interplay with the tumor microenvironment, which might provide deeper insights into the systemic effects of tumor-derived signals and their contributions to host dysfunction.

Toll Receptor Diversity: A Double-Edged Sword in Tumor Immunity

Our data reveal spatially resolved crosstalk between epithelial clones with distinct driver mutations (*Ras^{V12}* and *M6*) and between tumors and hemocytes, highlighting complex signaling networks that shape tumor progression. Our study highlights the tumor-promoting role of Toll pathway activation in the presence of oncogenic *Ras* mutation. Beyond the canonical Toll receptor

(Cleary et al.), the *Drosophila* genome encodes eight additional Toll family receptors (TLRs), whose roles in tumorigenesis are context-dependent, exerting both pro- and anti-tumorigenic effects across flies and human (Pradere, Dapito et al., 2014).

For instance, Toll signaling activation has been shown to impede the elimination of polarity-deficient cells, thereby fostering their tumorigenic growth (Katsukawa et al., 2018). Similarly, Toll-7 overexpression drives tumor overgrowth, while Toll-9 induction promotes undead apoptosis-induced proliferation (Ding, Li et al., 2022, Shields et al., 2022). Additionally, in tumor-associated hemocytes, we observed elevated expression of the ligand *spz* and the receptor *Toll-4*. In *Ras^{VI2}//M6* tumors, we detected upregulation of receptors *18w (Toll-2)*, *Toll-7*, and *Tollo*, along with the ligand *spz3* in GFP⁺ *Ras^{VI2}* tumor cells. Intriguingly, prior studies demonstrate that Spz ligands can bind multiple Toll receptors, including Toll-7, a receptor previously implicated in tumor promotion (Chowdhury, Li et al., 2019, Ding, Li et al., 2022). Our findings align with these reports and highlight the need to further dissect the functional roles of these upregulated ligands/receptors in tumorigenesis.

Conversely, Toll activation in the fat body upregulates AMP expression, leading to the remote triggering of tumor cell death (Parisi et al., 2014, Parvy et al., 2019). Moreover, we have previously demonstrated that Toll-6 activation facilitates the elimination of precancerous *scrib* clones by inducing mechanical tension-mediated Hippo activation (Kong, Zhao et al., 2022). Collectively, these results illustrate the multifaceted roles of Toll-like receptors in tumor immunity: they act as both tumor promoters and suppressors, with their functional outcomes dictated by specific receptor identity, genetic context, and tissue microenvironment.

***Drosophila* as a Paradigm for Conserved Tumor-Immune Dynamics**

The fruit fly *Drosophila* has emerged as a powerful model organism for investigating fundamental questions in human cancer biology. This is largely attributed to its suite of sophisticated genetic tools, the high level of genetic conservation, the short life cycle, and its potential for *in vivo* testing of anti-cancer drugs (Bang, Ang et al., 2019, Bilder et al., 2021, Dar, Das et al., 2012, Hsi, Ong et al., 2023, Liu et al., 2022c). In this study, we underscore the critical value of *Drosophila* in dissecting tumor heterogeneity and unraveling the intricate intercellular communication networks within the tumor microenvironment. The success of *Drosophila* models in elucidating conserved molecular mechanisms underlying tumor progression highlights their potential to offer insights into human tumor heterogeneity. Moving forward, translating these insights gained from *Drosophila* studies may enable the development of novel therapeutic strategies, either by disrupting tumor-hemocyte (immune cell) interactions or by leveraging innate immune pathways. Such advancements will further strengthen the connection between *Drosophila* models and human cancer biology, bridging the gap between basic research and clinical applications”.

Referee #3:

The authors present a narrative built upon a series of genetic interaction experiments, proposing that Ras^{VI2} cells surrounded by M6 mutant cells secrete Pvf1, which in turn stimulates hemocytes to secrete Spz. Spz then promotes overgrowth of Ras^{VI2} cells by inhibiting the Hippo pathway. While the proposed mechanism is plausible, additional data would strengthen the authors' claims and help address some mechanistic gaps.

Main Comments

1. The rationale for focusing on *M6* is not clearly articulated. The authors should clarify why *M6* was chosen as a point of investigation. Specifically, how does *M6* lead to *Pvf1* upregulation in *Ras^{V12}* cells? Is the effect of *M6* attributable solely to cell death or to structural defects in the cells? It would be informative to assess the impact of reaper-induced apoptosis or other manipulations of tricellular junctions to disentangle these possibilities.

Response: Thank you for raising these critical questions.

1. Rationale for Focusing on *M6*: *M6* was selected due to its established role as a core tricellular junction (TCJ) component essential for epithelial integrity. While prior studies demonstrated that clonal *scrib* mutations (a polarity/bicellular junction gene) near *Ras^{V12}*-expressing clones promote tumorigenesis (PMID: 20072127), the contribution of TCJ-specific perturbations to heterogeneity-driven tumor progression remained unexplored. *M6* provides a unique entry point to investigate TCJ-specific effects distinct from bicellular junction disruptions (e.g., *scrib*), enabling us to dissect how TCJ dysfunction influences tumor-microenvironment crosstalk.

2. Mechanism of *M6*-Driven *Pvf1* Upregulation: Our additional experiments reveal that *M6* loss in cells surrounding *Ras^{V12}* tumor cells activates the JNK signaling pathway, which genetically acts upstream of *Ets21C*, a direct transcriptional activator of *Pvf1*. As shown in the Author response Figure 25, *Ras^{V12}//M6^{-/-}* tumors induce JNK and *Ets21C*-dependent overgrowth and exhibit a 4.5-fold upregulation of *Ets21C* compared to controls, which can be suppressed by JNK inhibition. Furthermore, knocking down *Ets21C* in *Ras^{V12}//M6^{-/-}* tumors significantly inhibit *Pvf1* transcription upregulation, establishing a direct regulatory cascade in *Ras^{V12}* tumor cells (JNK activation → *Ets21C* upregulation → *Pvf1* induction).

Author response Figure 25. (A) Eye-antennal discs bearing GFP⁺ clones with indicated genotypes. (B-C) The qRT-PCR analyses of gene expression with indicated genotypes.

3. Distinguishing Structural Defects vs. Cell Death Effects: As suggested, we investigated these two non-mutually exclusive mechanisms: (1) apoptosis-induced compensatory proliferation and (2) tricellular junction (TCJ) dysfunction. Firstly, we tested whether localized apoptotic stress could drive tumorigenic cooperation. Using a Gal4-UAS/Q-MARCM dual system, we induced targeted overexpression of the pro-apoptotic gene *Hid* (*Head involution defective*) specifically in cells surrounding *QRas^{V12}* clones. While mild compensatory proliferation occurred in adjacent *QRas^{V12}* clones (Author response Figure 26), the overall eye-disc size remained unchanged, indicating that heterogeneous apoptosis induction does not robustly promote oncogenesis. Given *M6*'s role as a TCJ component, we extended our analysis to other TCJ proteins: Gliotactin (*Gli*) and Anakonda (*aka/bark*). Strikingly, RNAi knockdown of *Gli* or *aka* in cells surrounding

QRas^{V12} clones (*QRas^{V12}//Gli^{RNAi}*, *QRas^{V12}//aka^{RNAi}*) phenocopied *M6* loss of function phenotype, driving overgrowth of surrounding *QRas^{V12}* clones (Author response Figure 26). This demonstrates that TCJ architectural disruption, not *M6*-specific depletion, is the unifying factor enabling oncogenic cooperation with *Ras^{V12}*.

Author response Figure 26. GFP-labeled clones in eye-antennal discs bearing indicated genotypes.

These findings redefine TCJ integrity as a critical gatekeeper in regulating intercellular crosstalk-mediated tumor progression. While our study establishes TCJ dysfunction as a central driver of *Ras^{V12}*-driven tumorigenesis, we fully acknowledge the importance of unraveling the mechanistic link between TCJ breakdown and *Ras^{V12}* hyperactivation. We sincerely appreciate your insightful comments, which have guided us toward these interesting discoveries. Rest assured, we are actively investigating these questions in our ongoing follow-up studies.

2. The connection between Toll signaling and the Hippo pathway remains unclear. How exactly does Toll regulate Hippo? A mechanistic link between these pathways should be better defined. Generally, inhibition of Yorkie leads to suppression of cell proliferation, and this aspect should be addressed to clarify the proposed pathway.

Response: We appreciate the opportunity to clarify the relationship between Toll signaling and Hippo pathway regulation. To explore this interaction, we performed clonal overexpression of *dl* or *Dif* under physiological conditions and analyzed Hippo target gene expression. Intriguingly, while *dl/Dif* overexpression induced significant tissue overgrowth, we paradoxically observed reduced expression of Hippo targets *Diap1* and *CycE* (Author response Figure 27A-B). On the contrary, with the presence of oncogenic *Ras*, Toll activation strongly upregulated Hippo target gene expression (Author response Figure 27C). This apparent contradiction suggests context-dependent modulation of Hippo/Yki activity by Toll signaling, consistent with previous findings that Toll activation in polarity-deficient cells promotes Yki activity via JNK signaling (PMID: 29804808).

Author response Figure 27. Eye-antennal discs bearing GFP-labeled clones with indicated genotypes were stained with anti-Diap1, anti-CycE, and anti- β -galactosidase antibodies.

Further supporting the role of JNK signaling, KEGG pathway analysis of GFP⁺ *Ras*^{V12}//*M6*^{-/-} tumor cells revealed enrichment of MAPK signaling components (**Author response Figure 28**). Functional validation confirmed JNK activation through significant *Mmp1* upregulation (**Author response Figure 29A-B**). Notably, genetic inhibition of JNK signaling through the expression of a dominant-negative form of *bsk* (*bsk*^{DN}) suppressed *Ras*^{V12}//*M6*^{-/-}-driven tumorigenesis and abolished upregulation of *Mmp1* expression and *ex* transcription (**Author Response Figure 29A-D**). These results indicate that *Ras*^{V12}//*M6*^{-/-} tumors activate JNK signaling to regulate Hippo pathway activity.

Author response Figure 28. KEGG enrichment of DEGs from scRNA-seq in *Ras*^{V12}//*M6*^{-/-} specific GFP⁺ cell (A) and KEGG enrichment of all DEGs from bulk RNA-seq comparing *Ras*^{V12}//*M6*^{-/-} and *Ras*^{V12}//WT (B).

Author response Figure 29. Eye-antennal discs bearing GFP-labeled clones were stained with anti-Mmp1 antibody and anti- β -galactosidase antibody to visualize the *ex-lacZ* reporter.

Next, we explored whether Toll pathway activation is sufficient to activate JNK signaling. Under physiological conditions, ectopic expression of *dl* or *Dif* activates JNK signaling (*puc-lacZ* and *Mmp1* upregulation), along with slight F-actin accumulation (**Author response Figure 30**). Under oncogenic stress, Toll pathway activation synergized with *Ras^{V12}*, resulting in robust JNK signaling activation and strong F-actin polymerization (**Author response Figure 30**). Given that abnormal F-actin accumulation is known to activate Yki/YAP (PMID: 21525075, 24648494), these findings collectively support a model where *Ras^{V12}//M6^{-/-}* tumors act through the Toll-JNK-Hippo signaling cascade to promote tumorigenesis.

Author response Figure 30. Eye-antennal discs bearing GFP-labeled clones were stained with anti- β -galactosidase, anti-MMP1 antibodies, and Alexa Fluor® 555 Phalloidin.

3. The role of *Pvf1* in hemocyte activation is insufficiently resolved. It is not clear whether *Pvf1* functions to recruit hemocytes to *Ras^{V12}* cells or simply activates hemocytes irrespective of recruitment. Related to this, in Fig. 6B, how are hemocytes attracted to eye disc *Ras* tumors? According to Fig. 5E, *Ras^{V12}*-only tumors do not attract substantial numbers of hemocytes. Additionally, it appears that free Spz secreted by hemocytes alone is not sufficient to activate Toll signaling and drive *Ras^{V12}* malignant transformation.

Response: Thank you for these insightful comments regarding the role of *Pvf1* in hemocyte activation and recruitment. Below we address these points systematically:

1. *Pvf1*'s role in hemocyte recruitment and activation

To elucidate the *in vivo* role of *Pvf1*-*Pvr* signaling in regulating hemocyte phenotype during *Ras^{V12}//M6^{-/-}* induced tumorigenesis, we developed a *QRas^{V12}//M6^{-/-}* tumor model in the eye-antennal disc of *Drosophila* larvae using a dual system and inhibited *Pvr* in hemocytes by overexpressing a dominant negative form (*Pvr^{DN}*) via *He-Gal4*. Notably, *Pvr* inhibition in hemocytes significantly attenuated tumor growth and reduced hemocyte recruitment to *QRas^{V12}//M6^{-/-}* tumors (**Author response Figure 31**).

Author response Figure 31. Eye-antennal discs bearing GFP-labeled QMARCM clones were stained with anti-NimC1 antibody.

Furthermore, genetic inhibition of *Ets21C* or *Pvf1* significantly suppressed hemocyte recruitment to *Ras*^{V12}//*M6*^{-/-} tumors. Our data indicate that *Ras*^{V12}//*M6*^{-/-} tumors upregulate *Pvf1* transcription in an *Ets21C*-dependent manner (**Author response Figure 32**). Collectively, these findings suggest a model in which *Ras*^{V12}//*M6*^{-/-} tumor-secreted *Pvf1* functions as a chemoattractant to guide and activate the *Pvr* signaling in hemocytes, facilitating their recruitment to the tumor site. Consistent with our findings, previous studies have demonstrated that *Pvf* signaling via the *Pvr* receptor is indispensable for hemocyte migration (PMID: 11595182, 11595438).

Author response Figure 32. (A) Eye-antennal discs bearing GFP-labeled tumor clones were stained with anti-NimC1 antibody. (B) The qRT-PCR analyses of *Pvf1* expression with indicated genotypes.

2. Hemocyte recruitment to *Ras*^{V12} tumors and *spz*^{ACT} expression

We apologize for any confusion caused in the original manuscript regarding hemocyte recruitment to *Ras* tumors expressing *spz*^{ACT} in hemocytes. As shown in **Author response Figure 33**, clonal expression of *Ras*^{V12} alone does not significantly increase hemocyte recruitment. However, ectopic expression of *spz*^{ACT} in hemocytes under the control of *He-Gal4* or *cg-Gal4* promotes overgrowth of distal *Ras*^{V12} clones in the eye epithelium and hemocyte recruitment.

Author response Figure 33. (A) Cartoon illustrating the dual expression system to overexpress activated spz (spz^{ACT}) in the hemocytes and simultaneously induce GFP labeled QRas^{V12} clones in the eye-antennal disc controlled by the Q system. (B,C) Eye-antennal discs bearing GFP-labeled tumor clones were stained with anti-NimC1 antibody.

Furthermore, we observed strong JNK activation and Toll pathway activation in distal Ras^{V12} clones when spz^{ACT} is ectopically expressed in the hemocytes (**Author response Figure 34**), which phenocopies the Ras^{V12}//M6^{-/-} tumor condition. Given that our findings support a model in which Ras^{V12}//M6^{-/-} tumors promote tumorigenesis via the Toll-JNK-Hippo signaling cascade, it is reasonable to speculate that in this scenario, increased Toll-JNK signaling also facilitates hemocytes recruitment via the Pvf1-Pvr axis.

Author response Figure 34. Eye-antennal discs bearing GFP-labeled QMARCM tumor clones were stained with corresponding antibodies.

Specific Comments

4. In Fig. 2, bulk RNA-seq of the disc likely includes hemocytes. This should be explicitly acknowledged and discussed.

Response: Thank you for highlighting this critical point. In the revised manuscript, we have clarified and addressed this issue as follows:

“Bulk RNA-seq data inherently include contributions from infiltrating hemocytes, which play a role in tumor microenvironments by influencing the local transcriptome. These cells are recognized as a potential source of immune-related transcriptional signals. To validate key findings, such as the differential expression of Pvf1, we conducted scRNA-seq analyses, which confirmed these results and provided additional resolution.”

5. In Fig. 5A, the side view is difficult to interpret. The Spz-expressing cells appear to be disc proper cells rather than hemocytes. Although later figures include NimC1 staining to mark

hemocytes, the authors should clearly distinguish disc proper cells from hemocytes at this point in the figure.

Response: We appreciate your insightful comment regarding the cellular identification in Figure 5A. To clarify the spatial relationship between NimC1-positive hemocytes and tumor cells, we replaced the original images in 5H with updated ones and included a Movie EV1. Our analysis reveals that NimC1/Spz double-positive hemocytes are predominantly localized to the peripheral regions of tumor clones (**Author response Figure 35**).

Author response Figure 35. Eye-antennal disc bearing clones of $Ras^{V12}/M6^{-/-}$ were stained with both anti-spz and anti-NimC1 antibodies.

Furthermore, our scRNA-seq data analysis confirm this cellular distinction at the transcriptional level. *spz* expression was exclusively observed in hemocytes, with no detectable expression in GFP-labeled Ras^{V12} tumor clones or adjacent GFP-negative $M6$ -mutant cells (**Author response Figure 36**). This molecular evidence supports our conclusion that Spz production originates specifically from hemocytes rather than disc proper cells.

Author response Figure 36. UMAPs of *spz* expression in $Ras^{V12}/M6^{-/-}$ tumor-associated hemocytes (B) and in $Ras^{V12}/M6^{-/-}$ epithelial cells (C).

6. The Discussion section should be expanded to better articulate the current gaps in knowledge and explain how the present findings contribute to addressing these open questions.

Response: Thank you for bringing this up. In the revised manuscript, we have updated the discussion section as follows:

“Tumor heterogeneity is well-established feature across diverse cancer types, with profound implications for tumorigenesis and therapeutic resistance (Gavish et al., 2023, Gonzalez-Silva et al., 2020, Vitale et al., 2021). While intercellular communication between tumors and their microenvironment is recognized as a critical driver of malignancy, the *in vivo* mechanisms governing these interactions during tumor progression remain poorly understood, largely due to a lack of tractable experimental models. To address this gap, we developed a *Drosophila* tumor

heterogeneity model by selectively disrupting the tricellular junction protein *M6* in cells adjacent to benign *Ras*^{V12} tumors. Our findings revealed that loss of *M6* promotes tumor malignancy in neighboring *Ras*^{V12} tumors by inducing secretion of Pvf1. Pvf1 acts as a chemotactic factor, recruiting hemocytes via the Pvr receptor. These hemocytes, in turn, activate a paracrine Spz-Toll signaling axis within *Ras*^{V12} clones, establishing a complex interplay between the tumor and its microenvironment. Eventually, the activation of the Toll pathway synergizes with oncogenic *Ras* to induce tumor progression through a JNK-Hippo signaling cascade (Fig. EV8G). This intricate signaling network highlights the cooperative interactions between clones bearing distinct mutations and the immune microenvironment, providing mechanistic insights into how tumor heterogeneity and intercellular communication contribute to malignancy. Our study underscores the importance of understanding tumor-microenvironment dynamics and presents the *Drosophila* model as a powerful platform for dissecting these processes.

Dynamic Regulation of Hemocyte-Tissue Adhesion

Previous studies have demonstrated that ROS induced by apoptotic caspase promotes hemocytes recruitment in *Drosophila* eye epithelium, facilitating malignant tumor progression (Perez et al., 2017). Similarly, Toll activation in “undead cells” recruits hemocytes through caspase-dependent ROS signaling, driving JNK-mediated proliferation (Shields, Amcheslavsky et al., 2022). In this study, we reveal that *M6* mutant clones alone undergoing apoptosis fail to recruit hemocytes. These findings align with recent evidence showing that apoptosis induction is dispensable for hemocyte recruitment during classical cell competition (Zhu et al., 2025). This suggests that hemocyte recruitment to apoptotic sites in *Drosophila* is context-dependent and may require simultaneous ROS generation, or cytokine secretion such as Pvf1. Crucially, the functional outcome of hemocyte recruitment—whether promoting cell death or proliferation—appears to depend on the specific cellular context. For instance, when only the polarity gene *scrib* is mutated, hemocyte-secreted Eiger facilitates the clearance of *scrib*-mutant cells (Parisi et al., 2014). In contrast, within *Ras/scrib*^{-/-} tumors, activation of the *Ras* oncogene can subvert this hemocyte-mediated clearance process, instead hijacking it instead to promote tumor proliferation (Cordero et al., 2010, Perez et al., 2017). Furthermore, hemocytes can detect cell competition within tissues and actively participate by promoting the competitive interactions between “winner” and “loser” cells (Zhu et al., 2025). In our *Ras*^{V12}//*M6*^{-/-} tumor model, hemocytes again demonstrate specificity by homing to tumor cells, guided by Pvf1 secretion, and contributing to tumor progression.

These collective findings indicate that hemocytes possess a sophisticated ability to precisely sense genetic heterogeneity within tissues. Based on this discrimination, they can activate specific downstream signaling pathways that ultimately modulate the outcome of cellular interactions, promoting the clearance or proliferation of specific genotypic clones. Interestingly, both the “undead cell” model and the tumor heterogeneity model described here rely on Toll signaling to recruit hemocytes and activate JNK pathways for tumor progression. In the undead cell model, Toll/Toll-9 activation by *hid/rpr*-initiated caspase activity (which is subsequently inhibited), leading to hemocyte infiltration. In contrast, our model uniquely integrates Pvf1/Pvr-mediated adhesion to reprogram the tumor microenvironment.

Collectively, these findings underscore the complexity of hemocyte-mediated immune surveillance in *Drosophila*. However, a limitation of our study is the lack of mechanistic exploration into the relationship between apoptosis and ROS or Pvf1 secretion within *Ras*^{V12}//*M6*^{-/-}

tumors. While extensive apoptosis was observed in these tumors, the interplay between ROS-mediated oxidative stress and Pvf1-driven hemocyte recruitment remains uncharacterized. It is also worth noting that bulk RNA-seq data inherently include contributions from infiltrating hemocytes, which play a role in tumor microenvironments by influencing the local transcriptome. These cells are recognized as a potential source of immune-related transcriptional signals. To address this, we performed scRNA-seq analyses, which not only validated our bulk findings (e.g., Pvf1 differential expression) but also revealed previously unappreciated cellular heterogeneity within tumor-associated hemocytes. Hemocytes demonstrated remarkable functional plasticity, integrating spatial gradients, metabolic cues, and genetic alterations to execute context-dependent responses, participating in the competition and cooperation among different clones. Future investigations should focus on elucidating how tumor heterogeneity and dynamic microenvironmental signals regulate hemocyte functional states, particularly their dual roles in both anti-tumor defense and immunosuppressive microenvironment formation.

Pvf1/Pvr Signaling as a Multifunctional Axis in Systemic Dysregulation

Pvf1/Pvr signaling has emerged as a critical mediator of tumor-host organ interactions, playing a pivotal role in systemic dysregulation. Recent studies in *Drosophila* have demonstrated that Pvf1, secreted by intestinal tumors, can activate the Pvr receptor in distant organs, including the Malpighian tubules and AKH-producing cells (APCs). This activation leads to pathological outcomes such as renal dysfunction (Xu, Liu et al., 2024) and systemic host wasting (Ding, Li et al., 2025). Despite these findings, the mechanisms by which Pvf1 selectively triggers distinct downstream Pvr signaling pathways, such as JNK and ERK, in different target organs remain incompletely understood. For instance, Pvr activation in the Malpighian tubules primarily regulates uric acid metabolism through the JNK/Jra signaling axis (Xu et al., 2024). In contrast, Pvr activation in APCs appears to drive AKH release via mechanisms involving extracellular matrix (ECM) remodeling mediated by Mmp2 and enhanced innervation by excitatory cholinergic neurons (Ding et al., 2025). These distinct signaling outcomes emphasize the complexity and organ-specificity of Pvf1/Pvr interactions. Adding to this complexity, we discovered that Pvf1 secreted by *Ras*^{VI2}//*M6*^{-/-} tumors facilitate the recruitment and the activation of hemocytes. Activated hemocytes, in turn, upregulate the Toll ligand Spz via a mechanism that remains unidentified. Further investigation is needed to clarify the spatiotemporal dynamics of Pvf1-Pvr complexes and their interplay with the tumor microenvironment, which might provide deeper insights into the systemic effects of tumor-derived signals and their contributions to host dysfunction.

Toll Receptor Diversity: A Double-Edged Sword in Tumor Immunity

Our data reveal spatially resolved crosstalk between epithelial clones with distinct driver mutations (*Ras*^{VI2} and *M6*) and between tumors and hemocytes, highlighting complex signaling networks that shape tumor progression. Our study highlights the tumor-promoting role of Toll pathway activation in the presence of oncogenic *Ras* mutation. Beyond the canonical Toll receptor (Cleary et al.), the *Drosophila* genome encodes eight additional Toll family receptors (TLRs), whose roles in tumorigenesis are context-dependent, exerting both pro- and anti-tumorigenic effects across flies and human (Pradere, Dapito et al., 2014).

For instance, Toll signaling activation has been shown to impede the elimination of polarity-deficient cells, thereby fostering their tumorigenic growth (Katsukawa et al., 2018).

Similarly, Toll-7 overexpression drives tumor overgrowth, while Toll-9 induction promotes undead apoptosis-induced proliferation (Ding, Li et al., 2022, Shields et al., 2022). Additionally, in tumor-associated hemocytes, we observed elevated expression of the ligand *spz* and the receptor *Toll-4*. In *Ras^{VI2}//M6* tumors, we detected upregulation of receptors *18w* (*Toll-2*), *Toll-7*, and *Tollo*, along with the ligand *spz3* in GFP⁺ *Ras^{VI2}* tumor cells. Intriguingly, prior studies demonstrate that Spz ligands can bind multiple Toll receptors, including Toll-7, a receptor previously implicated in tumor promotion (Chowdhury, Li et al., 2019, Ding, Li et al., 2022). Our findings align with these reports and highlight the need to further dissect the functional roles of these upregulated ligands/receptors in tumorigenesis.

Conversely, Toll activation in the fat body upregulates AMP expression, leading to the remote triggering of tumor cell death (Parisi et al., 2014, Parvy et al., 2019). Moreover, we have previously demonstrated that Toll-6 activation facilitates the elimination of precancerous *scrib* clones by inducing mechanical tension-mediated Hippo activation (Kong, Zhao et al., 2022). Collectively, these results illustrate the multifaceted roles of Toll-like receptors in tumor immunity: they act as both tumor promoters and suppressors, with their functional outcomes dictated by specific receptor identity, genetic context, and tissue microenvironment.

***Drosophila* as a Paradigm for Conserved Tumor-Immune Dynamics**

The fruit fly *Drosophila* has emerged as a powerful model organism for investigating fundamental questions in human cancer biology. This is largely attributed to its suite of sophisticated genetic tools, the high level of genetic conservation, the short life cycle, and its potential for *in vivo* testing of anti-cancer drugs (Bang, Ang et al., 2019, Bilder et al., 2021, Das, Das et al., 2012, Hsi, Ong et al., 2023, Liu et al., 2022c). In this study, we underscore the critical value of *Drosophila* in dissecting tumor heterogeneity and unraveling the intricate intercellular communication networks within the tumor microenvironment. The success of *Drosophila* models in elucidating conserved molecular mechanisms underlying tumor progression highlights their potential to offer insights into human tumor heterogeneity. Moving forward, translating these insights gained from *Drosophila* studies may enable the development of novel therapeutic strategies, either by disrupting tumor-hemocyte (immune cell) interactions or by leveraging innate immune pathways. Such advancements will further strengthen the connection between *Drosophila* models and human cancer biology, bridging the gap between basic research and clinical applications”.

7. The authors note that *Drosophila* possesses multiple Toll receptors and ligands. Are other Toll ligands or receptors expressed in tumors or hemocytes? A brief re-analysis of the RNA-seq data to address this point would be valuable.

Response: Thank you for raising this important point. As suggested, we re-analyzed the scRNA-seq data to systematically assess the expression of all annotated *Toll* receptors and ligands in both hemocytes and tumor cells. After excluding genes with undetectable expression, we identified distinct patterns of dysregulation: In **tumor-associated hemocytes**, we observed elevated expression of the ligand *spz* and the receptor *Toll-4*. In **GFP⁺ *Ras^{VI2}* tumor cells**, besides a slight increase in *Toll*, we detected upregulation of receptors *18w* (*Toll-2*), *Toll-7*, and *Tollo*, as well as the ligand *spz3* (**Author response Figure 37**). Intriguingly, prior studies demonstrate that Spz ligands can bind multiple Toll receptors, including Toll-7, which itself has been implicated in tumor promotion (PMID: 31088910, 35050535). Our findings align with these reports and

highlight the need to further dissect the functional roles of these upregulated ligands/receptors in tumorigenesis.

Author response Figure 37. UMAPs of Toll receptors and ligands expression in *Ras^{V12}//M6^{-/-}* tumor-associated hemocytes (A) and GFP⁺ tumor cells (B).

Notably, while *Toll* mRNA levels were only slightly upregulated in the *Ras^{V12}* tumor cells, protein levels of Toll-GFP—quantified via an endogenous reporter in vivo—were significantly elevated in *Ras^{V12}* clones within *Ras^{V12}//M6^{-/-}* tumors (**Author Response Figure 38**). This discrepancy suggests that Toll protein abundance in *Ras^{V12}* cells may be regulated post-transcriptionally, potentially through mechanisms such as enhanced translation or protein stabilization, rather than transcriptional activation.

Author response Figure 38. UMAP of Toll expression in *Ras^{V12}//M6^{-/-}* GFP⁺ tumor cells (A) *Toll-GFP* expression in eye-antennal discs with *Ras^{V12}//WT* and *Ras^{V12}//M6^{-/-}* clones (B), and quantification of Toll-GFP intensity (C).

These findings highlight the intricate regulation of the Toll pathway within tumor microenvironments and pave the way for investigating its context-specific roles in cancer. Considering the evolutionary conservation of Toll-like receptors (TLRs) in mammalian systems,

future research aimed at unraveling these mechanisms could provide valuable insights for designing potential therapeutic strategies that target Toll signaling in human cancers.

Dear Xianjue,

Thank you for submitting a revised version of your manuscript. It has now been seen by all original reviewers, and I have copied their comments below.

As you can see, while reviewers #1 and #3 are fully satisfied with the added revisions, reviewer #2 finds that some of their initial requests were not clarified satisfactorily. Please clarify these remaining points by adding the requested information or toning down the statements.

Additionally, there are a few editorial points that need addressing before I can extend official acceptance of the manuscript:

1. Please update references according to The EMBO Journal style - it should be alphabetical. Where there are more than 10 authors on a paper, the first 10 should be listed, followed by 'et al.' Please see further information here: <https://www.embopress.org/page/journal/14602075/authorguide#referencesformat>
2. Please rename "Data and code availability" section into "Data availability".
3. Please zip the movie legend as a README text file together with the movie file. Further information is available here: <https://www.embopress.org/page/journal/14602075/authorguide#expandedview>
4. Source data for Fig 1E (WT-WT) does not seem to match the figure. Please carefully check also the other image panels. Our data editors have flagged the following issues in figure legends that need correcting:
 - Please provide the exact p values in the legends of figures 1D, H; 2C, F, H, J; 3K, 4B, D, E, I, K; 5C, I; 6C, H; EV1 B, D, E, I, J; EV2 D, F, H; EV3 D, E; EV4 C, EV6 C, D, G, H; EV7 A, F, G, I, L; EV8 A, B, D, E.
 - Please indicate the statistical test used for data analysis in the legends of figures 2A, 3G, EV7A.
 - Please provide information on the number and nature of replicates in the legend of figure 7G.
6. Papers published in The EMBO Journal are accompanied online by a 'Synopsis' to enhance discoverability of the manuscript. It consists of A) a short (1-2 sentences) summary of the findings and their significance, B) 3-4 bullet points highlighting key results and C) a synopsis image that is 550x300-600 pixels large (width x height, jpeg or png format). You can either show a model or key data in the synopsis image. Please note that the image size is rather small and that text needs to be readable at the final size.

With best wishes,

Ieva

Ieva Gailite, PhD
Senior Scientific Editor
The EMBO Journal
Meyerohofstrasse 1
D-69117 Heidelberg
Tel: +4962218891309
i.gailite@embojournal.org

We realize that it is difficult to revise to a specific deadline. In the interest of protecting the conceptual advance provided by the work, we recommend a revision within 3 months (14th Oct 2025). Please discuss the revision progress ahead of this time with the editor if you require more time to complete the revisions.

Referee #1:

The authors have completely addressed my review comments, even conducting more experiments than I asked for.

I also particularly appreciate the longer discussion.

The paper is highly rigorous and comprehensive, and provide exciting new findings on the importance of hemocytes in promoting tumor overgrowth in conditions of tumor heterogeneity. I believe it should be accepted.

Referee #2:

The authors have performed a lot of text changes and added a lot of new data. Regarding the comments I raised during the review, there are two main points that I feel have not been completely addressed:

- The developmental timing
- The Hippo pathway regulation

For these two aspects, changes in the text and figure legends/presentations would answer most of my concerns. Below is a detailed review of the different points initially raised and how the authors have answered.

Major comments

1. Developmental timing.

While I do understand that RasV12//M6 animals are delayed and that one cannot really compare RasV12//WT with RasV12//M6 at same age for all phenotypes or assays, I still would like to see the exact time of the discs shown in the different figure panels. The authors acknowledge that the growth control of disc is primarily intrinsic, and so comparing discs of different growth potentials which have been allowed to grow for different periods could introduce clear bias. The authors have not really answered to my comment, and I think the text added in the M&M actually adds confusion as it is not clear how the 50% pupariation is estimated, and implies that each genotype is dissected at different time/interval. To be clear, I am not asking that all discs are shown at the same time point across the whole study. But I would like to see discs of similar age when rescue experiments are being made and conclusions are being drawn on difference of GFP extent.

So what I would like to see is the age of discs indicated on the figures

Fig 1E and 1G
Fig 2D, 2E, 2G, 2I
Fig 3I, 3J
Fig 4A, 4C, 4F, 4H, 4J
Fig 5B, 5D, 5E, 5H
Fig 6B, 6E
Fig 7H, 7I

Furthermore, since rescues are being claimed based on GFP extent, I would like to be sure that discs shown and measured are of similar age in

Fig 2G, 2I
Fig 3J
Fig 4C, 4F, 4H (on the M6//RasV12 panels), 4J
Fig 5B, 5D
Fig 6B (for the two panels of the M6//RasV12)
Fig 7H, 7I

Similar details also apply to the relevant EV figures.

2. Magnification of discs

The authors have answered satisfactorily to my comment. Thank you.

3. The Hippo pathway regulation

I think there is a bit of a misunderstanding. I know that many other studies have looked at a variety of putative Hippo (Yki) target genes for monitoring Yki activity; but my point is that many of them (*wg*, *Diap1*, *myc*, *CycE*...) are not specific to Hippo/Yki (depending on target considered, they can be targets of Notch, EGFR, or other pathways). The authors acknowledge that *Diap1* protein level (and thus staining in discs) is under the control of *rpr*, and might not thus reflect directly Hippo pathway or Yki activities. Expanded is a more specific target for Yki and I would put more emphasis on that target rather than on others. If authors want to use *Diap1*, they should use the *Diap1*-LacZ element specific of the Sd/Yki sites made by Wu et al., 2008 PMID: 18258486.

Thus, claims that rely only on wg, Diap1 or CycE staining should be either nuanced in the text or back-up with ex-LacZ or Diap-LacZ (similarly to what is shown in 3H&I): Fig 4A, EV4A, EV4B, EV5B, EV6E.

To be completely clear, I think authors should either show the ex-LacZ for the critical experiments stated, or alternatively be more cautious in the text. While I am convinced that increased yki-mediated transcription is occurring in the M6//RasV12 eye disc tumour paradigm (ex-LacZ), and that it participates to tumour overgrowth, I think authors should be more cautious in the text. In the text and in the schematics shown, it looks like the authors imply that the Tl/dl-Dif pathway regulates the JNK and Hippo pathways. Since the mechanism for this "regulation" is not identified, I would ask the authors to be more nuanced in the text. Indeed, one possible model is that the Tl/dl-Dif pathway, rather than inhibiting the Hippo pathway, actually sensitises the wing discs cells to unchanged Yki levels (e.g. increased chromatin accessibility, cooperation between transcription factors at the level of gene promoters/enhancers...). Actually such a model would fit with the observation that Tl/dl in normal discs does not promote Diap1 and CycE expression. Here Tl/dl action is context-dependent and sensitises the discs to the oncomodule put in place by the RasV12 and JNK overactivity in eye discs Atkins et al., 2016 PMID: 27476594.

Finally, on the Hippo pathway regulation, the link between F-actin and Yki activation is not as simple as Actin accumulation leads to Yki activation as the authors imply. I think this piece of data should be removed, or cannot be used as an argument for Hippo pathway inhibition.

4. Explanation of the scRNA-Seq.

Thank you, it is much clearer now.

5. Recruitment of haemocytes and the role of apoptosis

Authors answered satisfactorily. Thank you.

6. Toll expression on RasV12 and the increased sensitivity of Rasv12 cells to spz

Authors answered satisfactorily. Thank you.

7. The specificity of M6 mutant cells

The added data is very interesting, showing that it is not just dying cells next to RasV12 cells that is triggering overgrowth. The fact that other TCJ proteins (and scribble cells) behave the same is intriguing and is discussed in the manuscript. But a parallel with the scrib/RasV12 juxtaposition model could be discussed further: Wu et al., 2010 PMID: 20072127

8. Ets21C - Pvf1 axis

Authors answered satisfactorily to the Ets21C and Pvf1 comments I had made. Thank you.

However, the new experiments provided on the PvrDN expressed in the haemocytes to formally test a direct effect of Pvf1/Pvr on the recruitment of haemocytes is not really convincing based on the pictures provided (Fig. 7I - the size difference seems to fall within the variability of discs shown along the paper). It should be quantified and tested for significance (of course on similarly aged discs- see comment 1). Based on the single cell data, Pvr is actually strongly expressed in GFP+ tumour cells, and it remains possible that the effect of Pvf1 is actually autocrine to promote tumour cell growth/survival, and that another yet to be identified signal is responsible for the recruitment of haemocytes.

The experiment of transplantation in Fig EV8 using the PvrDN looks more convincing. However, it should be noted that it has been reported that the immune response is different on transplanted tumours than on endogenous tumours in brain tumour models, undermining the relevance of this assay to study "tumour micro-environment formation" - Voutyraki et al., 2023 PMID: 37549261

Minor comments

1) RNAi lines

Authors answered satisfactorily. Thank you.

2) Tumour "heterogeneity"

I appreciate the open response of the authors and agree with many of their points, and I am satisfied with the text. I was indeed referring to classical subclonal evolution driven by branching mutations, which is typically reported in cancer in patients.

3) Discussion

Thank you for providing a developed discussion bringing interesting perspectives.

Referee #3:

The authors fully addresses my previous comments. The manuscript improved tremendously. I was very impressed by authors' meticulous responses.

Response to reviewer's comments (MS# EMBOJ-2025-120470R):

Dear Reviewers and Editor,

Thank you very much for taking the time to review our revised manuscript. We have carefully addressed all editorial points and made the necessary revisions to respond to the additional concerns raised. Below, you will find a point-by-point response to the reviewers' comments. For your convenience, all changes in the revised manuscript are highlighted in red font. We sincerely hope that the revised manuscript meets your expectations and is now suitable for publication.

Referee #1

The authors have completely addressed my review comments, even conducting more experiments than I asked for.

I also particularly appreciate the longer discussion.

The paper is highly rigorous and comprehensive and provide exciting new findings on the importance of hemocytes in promoting tumor overgrowth in conditions of tumor heterogeneity. I believe it should be accepted.

Response: We sincerely thank you for the thoughtful and encouraging comments. We are pleased that the additional experiments and expanded discussion have adequately addressed your concerns. We greatly appreciate your recognition of the rigor and significance of our work, as well as your positive recommendation for acceptance. Your support is invaluable to us.

Referee #2:

The authors have performed a lot of text changes and added a lot of new data. Regarding the comments I raised during the review, there are two main points that I feel have not been completely addressed:

- The developmental timing

- The Hippo pathway regulation

For these two aspects, changes in the text and figure legends/presentations would answer most of my concerns. Below is a detailed review of the different points initially raised and how the authors have answered.

Response: We sincerely appreciate your thorough evaluation of our revised manuscript and the insightful feedback you have provided. We are pleased to address, in a point-by-point manner, the two main concerns you have highlighted. For each, we have provided our response along with the corresponding revisions made to the text, figure legends, and data presentation.

*Major comments**1. Developmental timing.*

While I do understand that Ras^{V12}//M6 animals are delayed and that one cannot really compare Ras^{V12}//WT with Ras^{V12}//M6 at same age for all phenotypes or assays, I still would like to see the exact time of the discs shown in the different figure panels. The authors acknowledge that the growth control of disc is primarily intrinsic, and so comparing discs of different growth potentials which have been allowed to grow for different periods could introduce clear bias. The authors have not really answered to my comment, and I think the text added in the M&M actually adds confusion as it is not clear how the 50% pupariation is estimated, and implies that each genotype

is dissected at different time/interval.

To be clear, I am not asking that all discs are shown at the same time point across the whole study. But I would like to see discs of similar age when rescue experiments are being made and conclusions are being drawn on difference of GFP extent.

So what I would like to see is the age of discs indicated on the figures: Fig 1E and 1G, Fig 2D, 2E, 2G, 2I, Fig 3I, 3J, Fig 4A, 4C, 4F, 4H, 4J, Fig 5B, 5D, 5E, 5H, Fig 6B, 6E, Fig 7H, 7I.

Furthermore, since rescues are being claimed based on GFP extent, I would like to be sure that discs shown and measured are of similar age in Fig 2G, 2I, Fig 3J, Fig 4C, 4F, 4H (on the M6//Ras^{V12} panels), 4J, Fig 5B, 5D, Fig 6B (for the two panels of the M6//Ras^{V12}), Fig 7H, 7I. Similar details also apply to the relevant EV figures.

Response: Thank you very much for your thoughtful and detailed comments regarding the developmental timing and staging of the discs in our study. We fully appreciate your concerns and agree that precise staging is critical, particularly when comparing genotypes with different growth kinetics and when drawing conclusions based on differences in GFP extent in rescue experiments.

In response to your request, we have carefully reviewed all the relevant figures and now include the exact developmental stage (day n After Egg Laying, AEL-n) in each panel wherever applicable. For the rescue experiments, we have confirmed that the discs presented and quantified in the corresponding figures are of comparable developmental stages. This was done to minimize variability and ensure the validity of our comparisons. Additionally, we have removed the section titled “The determination of the tumor dissection time” from the Methods and Protocols to avoid potential confusion.

2. Magnification of discs

The authors have answered satisfactorily to my comment. Thank you.

Response: Thank you very much for your sincere suggestions and response.

3. The Hippo pathway regulation

I think there is a bit of a misunderstanding. I know that many other studies have looked at a variety of putative Hippo (Yki) target genes for monitoring Yki activity; but my point is that many of them (wg, Diap1, myc, CycE...) are not specific to Hippo/Yki (depending on target considered, they can be targets of Notch, EGFR, or other pathways). The authors acknowledge that Diap1 protein level (and thus staining in discs) is under the control of rpr, and might not thus reflect directly Hippo pathway or Yki activities. Expanded is a more specific target for Yki and I would put more emphasis on that target rather than on others. If authors want to use Diap1, they should use the Diap1-LacZ element specific of the Sd/Yki sites made by Wu et al., 2008 PMID: 18258486. Thus, claims that rely only on wg, Diap1 or CycE staining should be either nuanced in the text or back-up with ex-LacZ or Diap-LacZ (similarly to what is shown in 3H&I): Fig 4A, EV4A, EV4B, EV5B, EV6E.

To be completely clear, I think authors should either show the ex-LacZ for the critical experiments stated, or alternatively be more cautious in the text. While I am convinced that increased yki-mediated transcription is occurring in the M6//RasV12 eye disc tumour paradigm (ex-LacZ), and that it participates to tumour overgrowth, I think authors should be more cautious in the text. In the text and in the schematics shown, it looks like the authors imply that the Tl/dl-Dif pathway regulates the JNK and Hippo pathways. Since the mechanism for this "regulation" is not identified,

I would ask the authors to be more nuanced in the text. Indeed, one possible model is that the Tl/dl-Dif pathway, rather than inhibiting the Hippo pathway, actually sensitises the wing discs cells to unchanged Yki levels (e.g. increased chromatin accessibility, cooperation between transcription factors at the level of gene promoters/enhancers...). Actually such a model would fit with the observation that Tl/dl in normal discs does not promote Diap1 and CycE expression. Here Tl/dl action is context-dependent and sensitises the discs to the oncomodule put in place by the Ras^{V12} and JNK overactivity in eye discs Atkins et al., 2016 PMID: 27476594.

Finally, on the Hippo pathway regulation, the link between F-actin and Yki activation is not as simple as Actin accumulation leads to Yki activation as the authors imply. I think this piece of data should be removed, or cannot be used as an argument for Hippo pathway inhibition.

Response: Thank you for your insightful and detailed comments regarding the interpretation of Hippo pathway activity in our study. We sincerely apologize for the potential misunderstanding. We fully agree that many commonly used Yki target genes, such as *wg*, *Diap1*, and *CycE*, are not strictly specific to the Hippo pathway and can be influenced by other signaling pathways. As you pointed out, *Diap1* expression can be regulated by the pro-apoptotic gene *rpr*, and therefore may not always reflect Yki activity directly. Indeed, we have considered using *Diap1-lacZ* for validation; however, since **both *Diap1-lacZ* and *M6⁺* are located on chromosome 3L**, it was not technically feasible to simultaneously monitor *Diap1-lacZ* expression in both GFP-positive and GFP-negative clones. In line with your suggestion, we selected *ex-lacZ* as a representative Yki target gene for analysis during the first round of revision, unless technical constraints—such as the requirement for multiple genetic elements to be assembled on the same chromosome arm—made its use impractical. In cases where *ex-lacZ* could not be utilized, we therefore validated as many other Yki target genes as possible and further supported our conclusions by analyzing their mRNA expression levels via qPCR.

We truly appreciate your rigorous caution regarding the interpretation of the Tl/dl-Dif pathway's influence on Hippo and JNK signaling. We agree that the current data do not allow us to definitively conclude that Tl/dl-Dif directly inhibits the Hippo pathway. In response, we have adopted a more cautious tone when drawing conclusions, explicitly stating that the interaction between Toll and Hippo/Yki signaling is context-dependent and warrants further in-depth and comprehensive investigation.

We also share your perspective on F-actin, particularly the view that the relationship between F-actin and Yki activation is not simply linear. As a result, we have removed the related data from the manuscript. We hope that this will convey more accurate and clearly defined conclusions to the readers.

Thank you again for your careful evaluation and insightful comments on our work.

4. Explanation of the scRNA-Seq.

Thank you, it is much clearer now.

Response: Thank you for your feedback.

5. Recruitment of hemocytes and the role of apoptosis

Authors answered satisfactorily. Thank you.

Response: Thank you for your comments. Your question has also enabled us to discover more interesting phenomena.

6. Toll expression on Ras^{V12} and the increased sensitivity of Ras^{V12} cells to spz

Authors answered satisfactorily. Thank you.

Response: Thank you very much for your suggestions and response.

7. The specificity of M6 mutant cells

The added data is very interesting, showing that it is not just dying cells next to Ras^{V12} cells that is triggering overgrowth. The fact that other TCJ proteins (and scribble cells) behave the same is intriguing and is discussed in the manuscript. But a parallel with the scrib/Ras^{V12} juxtaposition model could be discussed further: Wu et al., 2010 PMID: 20072127

Response: We sincerely appreciate your thoughtful feedback and suggestions. In the revised manuscript, we have included additional discussion between the Ras^{V12}//M6^{-/-} tumor model and the scrib^{-/-}//Ras^{V12} model, aiming to provide readers with deeper insights into the underlying mechanisms.

“Interestingly, the cell polarity protein Scrib, which localizes to bicellular septate junctions (Byri et al, 2015), promotes Ras^{V12} clone malignancy when mutated in neighboring cells (Wu et al., 2010), further highlighting the critical role of junctional integrity in tumor initiation and progression. Both Ras^{V12}//scrib^{-/-} and Ras^{V12}//M6^{-/-} models illustrate cooperative oncogenesis in *Drosophila*, where Ras^{V12} alone creates benign tumors but requires additional mutations in surrounding cells to drive malignancy. Both models exploit the genetic tractability of *Drosophila* to uncover fundamental principles of intercellular communication in tumor progression, demonstrating how distinct genetic alterations in neighboring cells can synergize to promote malignant transformation. Notably, the Ras^{V12}//scrib^{-/-} model primarily underscores the activation of JNK pathway, whereas our Ras^{V12}//M6^{-/-} model reveals a more complex signaling cascade involving multiple pathways and cell types. Further investigation into the mechanistic similarities and differences between these two different heterogeneous tumor models would be of great interest.”

8. Ets21C - Pvf1 axis

Authors answered satisfactorily to the Ets21C and Pvf1 comments I had made. Thank you.

However, the new experiments provided on the Pvr^{DN} expressed in the haemocytes to formally test the direct effect of Pvf1/Pvr on the recruitment of haemocytes is not really convincing based on the pictures provided (Fig. 7I - the size difference seems to fall within the variability of discs shown along the paper). It should be quantified and tested for significance (of course on similarly aged discs- see comment 1). Based on the single cell data, Pvr is actually strongly expressed in GFP+ tumour cells, and it remains possible that the effect of Pvf1 is actually autocrine to promote tumour cell growth/survival, and that another yet to be identified signal is responsible for the recruitment of haemocytes.

The experiment of transplantation in Fig EV8 using the Pvr^{DN} looks more convincing. However, it should be noted that it has been reported that the immune response is different on transplanted tumours than on endogenous tumours in brain tumour models, undermining the relevance of this assay to study "tumour micro-environment formation" - Voutyraki et al., 2023 PMID: 37549261

Response: Thank you for your valuable feedback. We sincerely apologize for the inconvenience caused by our oversight regarding Figure 7I. Due to space constraints in the main figure panel, the

corresponding statistical analysis for Figure 7I was included in Figure EV8A and EV8B. Our results show that expression of Pvr^{DN} in hemocytes significantly suppresses the growth of $QRas^{V12}/M6^{-/-}$ tumors and reduces hemocytes recruitment (Figs. EV8A,B), highlighting the critical role of the Pvr receptor on hemocytes in tumor progression through the promotion of hemocyte infiltration.

Our single-cell data further confirm that Pvr expression is upregulated in GFP^{+} cells. We agree with your perspective that Pvf1 may promote tumor growth/survival via autocrine signaling. Moreover, although overexpression of Pvr^{DN} in hemocytes significantly inhibits $QRas^{V12}/M6^{-/-}$ tumor size and decreases hemocyte recruitment, the number of recruited hemocytes in this context remains significantly higher than in $QRas^{V12}/WT$ benign tumors. Therefore, we fully agree that additional, yet unidentified signals likely contribute to hemocyte recruitment. As suggested, we have added a sentence highlighting the potential involvement of other factors that might contribute to this process. We sincerely appreciate your constructive feedback on our research.

Furthermore, in contrast to the transplantation experiments conducted by Voutyraki *et al.*, in which dissociated tumor cells were injected into the abdomens of adult flies, we transplanted early-stage, intact tumor tissues directly into the abdomens of adult flies without prior tissue dissociation. Therefore, we hypothesize that this methodological difference may account for the observed variations in the extent of hemocyte infiltration. A detailed comparison of these two approaches would be of great interest; however, it falls beyond the scope of the current study.

Minor comments

1) RNAi lines

Authors answered satisfactorily. Thank you.

Response: Thank you for your feedback.

2) Tumour "heterogeneity"

I appreciate the open response of the authors and agree with many of their points, and I am satisfied with the text. I was indeed referring to classical subclonal evolution driven by branching mutations, which is typically reported in cancer in patients.

Response: We appreciate your acknowledgment of the points we addressed regarding tumour heterogeneity.

3) Discussion

Thank you for providing a developed discussion bringing interesting perspectives.

Response: Thank you very much for your valuable feedback.

Referee #3:

The author fully addresses my previous comments. The manuscript improved tremendously. I was very impressed by authors' meticulous responses.

Response: Thank you very much for your kind and encouraging comments. We are pleased to hear that our revisions have significantly improved the manuscript and that you found our responses thorough and satisfactory.

Dear Xianjue,

Thank you for addressing the final editorial points. I apologise for the delay in the processing of your manuscript due to the holiday period. I am now pleased to inform you that your manuscript has been accepted for publication in the EMBO Journal. Congratulations with a nice study!

Before we forward your manuscript to our publishers, we would like to propose some minor edits in the manuscript title, abstract and synopsis (please see below and in the attached file). I have also written a short blurb that will accompany the title of your manuscript in our online table of contents. Please let me know if any corrections or adjustments are needed.

Title:
Hemocytes facilitate interclonal cooperation-induced tumor malignancy by hijacking the innate immune system in *Drosophila*

Blurb:
Loss of the tricellular junction protein M6 in the cells surrounding benign epithelial tumors induces hemocyte recruitment, which promote tumor progression via Toll pathway activation.

Synopsis:
The crosstalk between *Drosophila* hemocytes and epithelial tumors remains incompletely understood. This study reveals that hemocytes promote malignant transformation through a bidirectional communication loop with tumor cells: tumor cells recruit hemocytes, which, in turn, accelerate tumor progression.

- Mutation of the tricellular junction protein M6 in cells surrounding RasV12 benign tumors induces malignant transformation.
- M6 loss in the neighboring cells promotes overgrowth of RasV12 tumors via the Toll-JNK-Hippo signaling cascade.
- Hemocyte-derived Spz activates the Toll pathway in RasV12 clones within the RasV12//M6^{-/-} heterogeneous tumors.
- RasV12//M6^{-/-} heterogeneous tumors secrete Pvf1 ligand, activating Pvr receptor on hemocytes and facilitating their recruitment to the tumor.

If you have any questions, please do not hesitate to contact the Editorial Office. Thank you for this interesting contribution to The EMBO Journal!

With best wishes,

Ieva

Ieva Gailite, PhD
Senior Scientific Editor
The EMBO Journal
Meyerohofstrasse 1
D-69117 Heidelberg
Tel: +4962218891309
i.gailite@embojournal.org
